# HOW DOES SEMI-SUPERVISED LEARNING WITH PSEUDO-LABELERS WORK? A CASE STUDY

**Yiwen Kou**[1], **Zixiang Chen**[1], **Yuan Cao**[2,3], **Quanquan Gu**[1]
[1]Department of Computer Science, University of California, Los Angeles
[2]Department of Statistics and Actuarial Science, The University of Hong Kong
[3]Department of Mathematics, The University of Hong Kong
`evankou@ucla.edu`, `chenzx19@cs.ucla.edu`, `yuancao@hku.hk`,
`qgu@cs.ucla.edu`

## ABSTRACT

Semi-supervised learning is a popular machine learning paradigm that utilizes a large amount of unlabeled data as well as a small amount of labeled data to facilitate learning tasks. While semi-supervised learning has achieved great success in training neural networks, its theoretical understanding remains largely open. In this paper, we aim to theoretically understand a semi-supervised learning approach based on pre-training and linear probing. In particular, the semi-supervised learning approach we consider first trains a two-layer neural network based on the unlabeled data with the help of pseudo-labelers. Then it linearly probes the pre-trained network on a small amount of labeled data. We prove that, under a certain toy data generation model and two-layer convolutional neural network, the semi-supervised learning approach can achieve nearly zero test loss, while a neural network directly trained by supervised learning on the same amount of labeled data can only achieve constant test loss. Through this case study, we demonstrate a separation between semi-supervised learning and supervised learning in terms of test loss provided the same amount of labeled data.

## 1 INTRODUCTION

With the help of human-annotated labels, supervised learning has achieved remarkable success in several computer vision tasks (Girshick et al., 2014; Long et al., 2015; Krizhevsky et al., 2012; Tran et al., 2015). However, annotating large-scale datasets (e.g., video datasets with temporal dimensions) is time-consuming and costly. In order to reduce the number of labels used for training while maintaining a good prediction performance, a variety of methods have been proposed. Among these methods, *semi-supervised learning* (Scudder, 1965; Fralick, 1967; Agrawala, 1970), which leverages both a small amount of labeled data and a large amount of unlabeled data to improve learning performance, is one of the most widely used approaches. It has been shown to achieve promising performance for a wide variety of tasks, including image classification (Rasmus et al., 2015; Springenberg, 2015; Laine & Aila, 2016), image generation (Kingma et al., 2014; Odena, 2016; Salimans et al., 2016), domain adaptation (Saito et al., 2017; Shu et al., 2018; Lee et al., 2019), and word embedding (Turian et al., 2010; Peters et al., 2017).

One of the popular semi-supervised learning approaches is *pseudo-labeling* (Lee et al., 2013), which generates pseudo-labels of unlabeled data for pre-training. This approach has been remarkably successful in improving performance on many tasks. For example, in image classification, one can first train a teacher network on a small labeled dataset and use it as a *pseudo-labeler* to generate *pseudo-labels* for large unlabeled datasets. Then one can train a student network on the combination of labeled and pseudo-labeled images (Xie et al., 2020; Pham et al., 2021b; Rizve et al., 2021). In order to theoretically understand semi-supervised learning with *pseudo-labelers*, Oymak & Gulcu (2021) considered learning a linear classifier in the Gaussian mixture model setting. They are able to show that in the high dimensional limit, the predictors found by semi-supervised learning are correlated with the Bayes-optimal predictor. Frei et al. (2022c) further proved that the semi-supervised learning algorithm can provably converge to the Bayes-optimal predictor for mixture models. However, their analyses are limited to linear classifiers, and cannot explain the success of semi-supervised learning with neural networks.

In this paper, we attempt to theoretically explain the success of semi-supervised learning with pseudo-labelers in training neural networks. Specifically, we focus on a toy data model that contains both signal patches and noise patches, where the signal patch is correlated to the label while the noise patch is not. We consider semi-supervised learning with pre-training and linear probing. In the pre-training state, we train a two-layer convolutional neural network (CNN) on an unlabeled dataset with pseudo-labels. We then fine-tune the pre-trained model using linear probing on a small amount of labeled data. We provide a comprehensive analysis of the learning process in both pre-training and linear probing stages.

The contributions of our work are summarized as follows.

- We theoretically show that with the help of pseudo-labelers, CNN can learn the feature representation during the pre-training stage. Moreover, the learned feature is highly correlated with the true labels of the data, even though the true labels are unknown and not used during the pre-training stage.

- Based on our analysis of the pre-training process, we further show that when linear-probing the pre-trained model in the downstream task, the final classifier can achieve near-zero test loss and test error. Notably, these guarantees of small test loss and error only require a very small number of labeled training data.

- As a comparison, we show that standard supervised learning cannot learn a good classifier under the same setting. Specifically, we show that, even when the training process converges to a global minimum of the training loss, the learned two-layer CNN can only achieve constant level test loss. This, together with the aforementioned results for semi-supervised learning, demonstrates the advantage of semi-supervised learning over standard supervised learning.

**Notation.** We use lower case letters, lower case bold face letters, and upper case bold face letters to denote scalars, vectors, and matrices respectively. For a scalar $x$, we use $[x]_+$ to denote $\max\{x, 0\}$. For a vector $\mathbf{v} = (v_1, \cdots, v_d)^\top$, we denote by $\|\mathbf{v}\|_2 := \left( \sum_{i=1}^d v_i^2 \right)^{\frac{1}{2}}$ its $\ell_2$ norm, and use $\mathrm{supp}(\mathbf{v}) := \{j : v_j \neq 0\}$ to denote its support. For two sequences $\{a_k\}$ and $\{b_k\}$, we denote $a_k = O(b_k)$ if $|a_k| \leq C|b_k|$ for some absolute constant $C$, denote $a_k = \Omega(b_k)$ if $b_k = O(a_k)$, and denote $a_k = \Theta(b_k)$ if $|a_k| \leq C|b_k|$ and $a_k = \Omega(b_k)$. We also denote $a_k = o(b_k)$ if $\lim |a_k/b_k| = 0$. Finally, we use $\widetilde{\Theta}(\cdot)$, $\widetilde{O}(\cdot)$ and $\widetilde{\Omega}(\cdot)$ to omit logarithmic terms in the notations.

## 2 RELATED WORK

**Semi-supervised learning methods in practice.** Since the invention of semi-supervised learning in Scudder (1965); Fralick (1967); Agrawala (1970), a wide range of semi-supervised learning approaches have been proposed, including generative models (Miller & Uyar, 1996; Nigam et al., 2000), semi-supervised support vector machines (Bennett & Demiriz, 1998; Xu et al., 2007; 2009), graph-based methods (Zhu et al., 2003; Belkin et al., 2006; Zhou et al., 2003), and co-training (Blum & Mitchell, 1998), etc. For a comprehensive review of classical semi-supervised learning methods, please refer to Chapelle et al. (2010); Zhu & Goldberg (2009). In the past years, a number of deep semi-supervised learning approaches have been proposed, such as generative methods (Odena, 2016; Li et al., 2019), consistency regularization methods (Sajjadi et al., 2016; Laine & Aila, 2016; Rasmus et al., 2015; Tarvainen & Valpola, 2017) and pseudo-labeling methods (Lee et al., 2013; Zhai et al., 2019; Xie et al., 2020; Pham et al., 2021a). In this work, we will focus on pseudo-labeling methods.

**Theory of semi-supervised learning.** To understand semi-supervised learning, Castelli & Cover (1995; 1996) studied the relative value of labeled data over unlabeled data under a parametric assumption on the marginal distribution of input features. Later, a series of works proved that semi-supervised learning can possess better sample complexity or generalization performance than supervised learning under certain assumptions on the marginal distribution (Niyogi, 2013; Globerson et al., 2017) or the ratio of labeled and unlabeled samples (Singh et al., 2008; Darnstädt, 2015), while Balcan & Blum (2010) provided a unified PAC framework able to analyze both sample-complexity and algorithmic issues. Oymak & Gulcu (2021); Frei et al. (2022c) considered semi-supervised learning with pseudo-labers by learning a linear classifier for mixture models and convergence to Bayes-optimal predictor.

**Self-supervised learning in practice.** A closely related learning paradigm to semi-supervised learning is called self-supervised learning, which creates human-designed supervised learning problems to leverage natural structures and learn representations from unlabeled data. Representative self-supervised learning approaches include *contrastive learning* and *pretext-based self-supervised learning*. Contrastive learning (Caron et al., 2020; He et al., 2020; Chen et al., 2020) aims to group similar examples closer and dissimilar examples far from each other by utilizing a similarity metric, while pretext-based self-supervised tries to learn a good representation from *pretext tasks* generated from the unlabeled data to facilitate *downstream learning tasks*. In practice, various pretext tasks have been proposed, which include (1) generation-based ones such as colorizing grayscale images (Zhang et al., 2016), image inpainting (Pathak et al., 2016), image and video generation with GAN (Goodfellow et al., 2014; Brock et al., 2018; Karras et al., 2019; Vondrick et al., 2016; Tulyakov et al., 2018); and (2) context-based ones such as image jigsaw puzzle (Noroozi & Favaro, 2016), geometric transformation (Gidaris et al., 2018; Jing et al., 2018), frame order verification and recognition (Lee et al., 2017; Misra et al., 2016; Wei et al., 2018). The semi-supervised learning approach with pseudo-labelers studied in this paper is related to pretext-based self-supervised learning because the unlabeled data with pseudo-labels can be seen as a particular pretext task.

**Theory of self-supervised learning.** In order to understand self-supervised learning, there is a line of work towards understanding *contrastive learning* (Saunshi et al., 2019; Tsai et al., 2020; Mitrovic et al., 2020; Tian et al., 2020; Wang & Isola, 2020; Tosh et al., 2021b;a; HaoChen et al., 2021; Wen & Li, 2021; Saunshi et al., 2022), which is one of the most used self-supervised learning approaches based on data augmentation. Unlike contrastive learning, the theoretical understanding of pretext-based self-supervised learning is still rather limited. The only notable works are Lee et al. (2020) and Wei et al. (2020). Lee et al. (2020) proved generalization guarantees for self-supervised algorithms using empirical risk minimization on the pretext task under certain conditional independence assumptions. Wei et al. (2020) proved that under an "expansion" assumption, the minimizer of the population loss based on self-training and input-consistency regularization will achieve high prediction accuracy. Since semi-supervised learning with pseudo-labelers can be seen as a special case of pretext-based self-supervised learning (the pretext task is generated by the pseudo-labelers), we believe the case study in the current paper and its theoretical understanding can shed light on pretext-based self-supervised learning as well.

**Feature learning by neural networks.** Our work is also closely related to several recent works that study how neural networks learn the features. Allen-Zhu & Li (2020a) showed that adversarial training purifies the learned features by removing certain "dense mixtures" in the hidden layer weights of the network. Allen-Zhu & Li (2020b) studied how ensemble and knowledge distillation work in deep learning when the data have "multi-view" features. Zou et al. (2021) studied an aspect of feature learning by Adam and GD and showed that GD can learn the sparse features while Adam may fail even with proper regularization. Notably, there are two concurrent works studying the benign overfitting phenomenon in learning neural networks: Frei et al. (2022a) established theoretical guarantees for benign overfitting of two-layer fully connected neural networks with zero training error and test error close to the Bayes-optimal error, while Cao et al. (2022) studied the benign overfitting phenomenon in training a two-layer convolutional neural network (CNN), achieving arbitrarily small training and test loss. Our work studies a different aspect of feature learning afforded by semi-supervised learning versus supervised learning: given a small amount of labeled data, semi-supervised learning can learn the features with the help of pseudo-labelers, while supervised learning fails to learn the features and tends to overfit the noise in the training data.

## 3 PROBLEM SETUP AND PRELIMINARIES

In this section, we first give a brief overview of the semi-supervised learning pipeline using pseudo-labelers. Then we will introduce our data model, the convolutional neural network, and the detail of the training algorithms considered in this paper.

### 3.1 SEMI-SUPERVISED LEARNING PIPELINE WITH PSEUDO-LABELERS

In this paper, we consider a kind of semi-supervised learning (Xie et al., 2020; Pham et al., 2021b; Rizve et al., 2021), which leverages pseudo-labelers for pre-training. Such a semi-supervised learning method is related to a special kind of pretext-based self-supervised learning, whose pretext task

Figure 1: The general pipeline of semi-supervised learning with pre-training and linear probing.

is designed by generating pseudo-labels for unlabeled data with the help of pseudo-labelers (Zhai et al., 2019). The typical pipeline of this kind of semi-supervised learning is shown in Figure 1. Moreover, the case study we carry out is shown in Figure 2. The pretext task trains a two-layer convolutional neural network with the help of pseudo-labelers, and the downstream task trains a linear probe using the pre-trained models.

## 3.2 DATA DISTRIBUTION

Inspired by recent work (Allen-Zhu & Li, 2020b; Zou et al., 2021; Shen et al., 2022; Cao et al., 2022), we consider a toy data model where each data input $\mathbf{x}$ consists of two patches $\mathbf{x}^{(1)}$ and $\mathbf{x}^{(2)}$, where each patch has $d$ dimensions. We focus on binary classification task, and present our data distribution $\mathcal{D}$ in the following definition.

**Definition 3.1.** Each data point $(\mathbf{x}, y)$ with $\mathbf{x} = [\mathbf{x}^{(1)\top}, \mathbf{x}^{(2)\top}]^\top \in \mathbb{R}^{2d}$ and $y \in \{-1, +1\}$ is generated as follows: the label $y$ is generated as a Rademacher random variable; one of $\mathbf{x}^{(1)}, \mathbf{x}^{(2)}$ is given by the feature vector $y \cdot \mathbf{v}$, the other is given by a noise vector $\boldsymbol{\xi}$ that is generated from a $d$-dimensional Gaussian distribution $\mathcal{N}(\mathbf{0}, \sigma_p^2(\mathbf{I} - \mathbf{v}\mathbf{v}^\top / \|\mathbf{v}\|_2^2))$. We denote by $\mathcal{D}$ the joint distribution of $(\mathbf{x}, y)$, and denote by $\mathcal{D}_{\mathbf{x}}$ the marginal distribution of $\mathbf{x}$.

The most natural way to think of our data model is to treat patches $\mathbf{x}^{(1)}$ and $\mathbf{x}^{(2)}$ as the embedding of the image data: one of them is a signal which is label-dependent, and the other one is the noise that is label-independent. For simplicity, we assume that the noise patch is generated from the Gaussian distribution $N(\mathbf{0}, \sigma_p^2 \cdot (\mathbf{I} - \mathbf{v}\mathbf{v}^\top \cdot \|\mathbf{v}\|_2^{-2}))$ to ensure that the noise vector is orthogonal to the signal vector $\mathbf{v}$, and only consider the case where the data consists of one signal patch and one noise patch. However, our results and proof techniques can be easily extended to cover the setting with non-orthogonal signal/noise and multiple signal/noise patches. With this simple data model, in this case study we aim to show the effectiveness of semi-supervised learning and explain the mechanism behind semi-supervised learning with neural networks.

Since the positions of signal and noise are not specified in Definition 3.1. It is natural to use a classifier with a convolutional structure that applies the same function to each patch. More specifically, we consider learning a CNN with $n_l$ labeled examples $S' = \{(\mathbf{x}_i', y_i')\}_{i=1}^{n_l}$ generated from the distribution $\mathcal{D}$ and $n_u$ unlabeled examples $S = \{\mathbf{x}_i\}_{i=1}^{n_u}$ generated from the marginal distribution $\mathcal{D}_{\mathbf{x}}$, where $n_l$ is significantly smaller than the dimension $d$. If we only use the labeled data, the CNN can easily overfit the training dataset by memorizing the noise patches $\boldsymbol{\xi}_i$. Consequently, the CNN will perform badly on the fresh test data. Therefore, our case is hard to learn without using unlabeled examples.

## 3.3 SUPERVISED LEARNING MODELS

For supervised learning, we consider a two-layer CNN whose filters are applied to the patches $\mathbf{x}^{(1)}$ and $\mathbf{x}^{(2)}$ respectively and parameters in the second layers are set to be $\pm 1$. Then the CNN can be written as $f_{\mathbf{W}}(\mathbf{x}) = f_{\mathbf{W}}^{+1}(\mathbf{x}) - f_{\mathbf{W}}^{-1}(\mathbf{x})$ where $f_{\mathbf{W}}(\mathbf{x})^{+1}, f_{\mathbf{W}}(\mathbf{x})^{-1}$ are formulated as

$$f_{\mathbf{W}}^{+1}(\mathbf{x}) = \sum_{j=1}^{m} \left[ \sigma(\langle \mathbf{w}_j, \mathbf{x}^{(1)} \rangle) + \sigma(\langle \mathbf{w}_j, \mathbf{x}^{(2)} \rangle) \right],$$

$$f_{\mathbf{W}}^{-1}(\mathbf{x}) = \sum_{j=m+1}^{2m} \left[ \sigma(\langle \mathbf{w}_j, \mathbf{x}^{(1)} \rangle) + \sigma(\langle \mathbf{w}_j, \mathbf{x}^{(2)} \rangle) \right].$$

(3.1)

Figure 2: Illustration of our model. The left figure characterizes semi-supervised pre-train schema: NN is trained by minimizing errors between pseudo-labels $\widehat{y}$ and predictions $f_{\mathbf{W}}(\mathbf{x})$. After semi-supervised pre-training finished, the learned parameters $\{\mathbf{W}_k^*\}_{k=1}^K$ serve as pre-trained models and are adapted to a downstream task using linear probing, as shown in the right figure.

Here $\sigma$ is activation function $\text{ReLU}^q(\cdot) = [\cdot]_+^q (q > 2)$, $m$ is the width of the network, $\mathbf{w}_j \in \mathbb{R}^d$ denotes the $j$-th filter, and $\mathbf{W}$ is the collection of all filters $\{\mathbf{w}_j\}_{j=1}^{2m}$. Given labeled training dataset $S' = \{(\mathbf{x}_i', y_i')\}_{i=1}^{n_1}$, we train the CNN model by minimizing the empirical cross-entropy loss

$$L_{S'}(\mathbf{W}) = \frac{1}{n_1} \sum_{i=1}^{n_1} L_i(\mathbf{W}),$$

where $L_i(\mathbf{W}) = \ell(y_i' \cdot f_{\mathbf{W}}(\mathbf{x}_i'))$ with $\ell(z) = \log(1 + \exp(-z))$ denotes the individual loss for the training example $(\mathbf{x}_i, y_i)$. We minimize the empirical function $L_{S'}(\mathbf{W})$ with gradient descent as follows

$$\mathbf{w}_j^{(t+1)} = \mathbf{w}_j^{(t)} - \eta \cdot \nabla_{\mathbf{w}_j} L_{S'}(\mathbf{W}^{(t)}), \quad \mathbf{w}_j^{(0)} \sim \mathcal{N}(\mathbf{0}, \sigma_0^2 \mathbf{I}), \quad j \in [2m],$$

where $\eta > 0$ is the learning rate and $\sigma_0$ defines the scale of random initialization.

### 3.4 SEMI-SUPERVISED LEARNING MODELS

For semi-supervised pre-training, we assume that we have access to $K$ pseudo-labelers $\{f_k^{\text{w}}\}_{k=1}^K$. The accuracy of $k$-th pseudo-labeler is $p_k \in (1/2, 1)$. Then we use $K$ pseudo-labelers to generate $K$ pseudo-labeled dataset $\{S_k\}_{k=1}^K$, where $S_k := \left\{ (\mathbf{x}_i, \widehat{y}_{k,i}) \,\middle|\, \widehat{y}_{k,i} = f_k^{\text{w}}(\mathbf{x}_i) \right\}_{i=1}^{n_{\text{u}}}$. Next we solve $K$ pre-training tasks with two-layer CNN models $\{f_{\mathbf{W}_k}\}_{k=1}^K$ defined in (3.1) using $\{S_k\}_{k=1}^K$ respectively. Note that our result can cover $K = 1$ as a special case, where there is only one pseudo-labeler.

We consider learning the model parameter $\mathbf{W}_k$ by optimizing the empirical loss of both pseudo-labeled dataset $S_k$ and labeled dataset $S' = \{(\mathbf{x}_i', y_i')\}_{i=1}^{n_1}$ with weight decay regularization

$$L_{S_k \cup S'}(\mathbf{W}_k) = \frac{1}{n_{\text{u}} + n_1} \left( \sum_{i=1}^{n_{\text{u}}} L_i(\mathbf{W}_k) + \sum_{i'=1}^{n_1} L_{i'}(\mathbf{W}_k) \right) + \frac{\lambda}{2} \|\mathbf{W}_k\|_F^2,$$

where $\lambda \geq 0$ is the regularization parameter, $L_i(\mathbf{W}_k) = \ell(\widehat{y}_{k,i} \cdot f_{\mathbf{W}_k}(\mathbf{x}_i))$ denotes the individual loss for the pseudo-labeled data $(\mathbf{x}_i, \widehat{y}_{k,i})$, $L_{i'}(\mathbf{W}_k) = \ell(y_i' \cdot f_{\mathbf{W}_k}(\mathbf{x}_i'))$ denotes the individual loss for the labeled data $(\mathbf{x}_i', y_i')$. Our result can cover $n_1 = 0$, which corresponds to the case that there is no labeled data during pre-training. In light of this, our semi-supervised learning framework will reduce to a special kind of pretext-based self-supervised learning, where the pretext tasks are generated by pseudo-labelers.

We use gradient descent to minimize the regularized loss function $L_{S_k \cup S'}(\mathbf{W}_k)$. Starting from initial $\mathbf{W}_k^{(0)} := \{\mathbf{w}_{k,j}^{(0)}, j \in [2m]\}$, gradient descent update rule is as follows

$$\mathbf{w}_{k,j}^{(t+1)} = \mathbf{w}_{k,j}^{(t)} - \eta \cdot \nabla_{\mathbf{w}_{k,j}} L_{S_k \cup S'}(\mathbf{W}_k^{(t)}), \quad \mathbf{w}_{k,j}^{(0)} \sim \mathcal{N}(\mathbf{0}, \sigma_0^2 \mathbf{I}_d), \quad j \in [2m], k \in [K]$$

where $\eta > 0$ is the learning rate and $\sigma_0$ defines the scale of random initialization.

● **Downstream Task: Linear Model.** The semi-supervised pre-training gives us $K$ CNN models with parameters $\{\mathbf{W}_k^*\}_{k=1}^K$. Based on them, for the downstream task, we consider a linear model

$$g_{\mathbf{a}}(\mathbf{x}) = \sum_{k=1}^K a_k f_{\mathbf{W}_k^*}(\mathbf{x}),$$

where $a_k \in \mathbb{R}$ denotes the trainable weight for the $k$-th pre-trained model. Then, given $\{f_{\mathbf{W}_k^*}\}_{k=1}^K$ and labeled training data $S' = \{(\mathbf{x}_i', y_i')\}_{i=1}^n$, we consider learning the downstream linear model parameter $\mathbf{a}$ by optimizing the following empirical loss

$$L_{S'}(\mathbf{a}) = \frac{1}{n} \sum_{i=1}^n \ell(y_i' \cdot g_{\mathbf{a}}(\mathbf{x}_i')).$$

We initialize $\mathbf{a}$ as an all zero vector and optimize the empirical loss by gradient descent with learning rate $\eta$, i.e.,

$$\mathbf{a}^{(t+1)} = \mathbf{a}^{(t)} - \eta \cdot \nabla_{\mathbf{a}} L_{S'}(\mathbf{a}^{(t)}), \ \mathbf{a}^{(0)} = \mathbf{0}.$$

## 4 MAIN RESULTS

In this section, we present the main theoretical results in this paper. We start with a condition that is required by our analysis.

**Condition 4.1.** The strength of the signal is $\|\mathbf{v}\|_2^2 = \Theta(d)$, the noise variance is $\sigma_p = \Theta(d^\epsilon)$, where $0 < \epsilon < 1/8$ is a small constant, and the width of the network satisfies $m = \text{polylog}(d)$. We also assume that the size of the unlabeled dataset $n_u = \Omega(d^{4\epsilon})$, and labeled data $n_l = \widetilde{\Theta}(1)$. For both supervise learning and semi-supervised learning settings, we initialize the weight with $\sigma_0 = \Theta(d^{-3/4})$. For semi-supervised learning, we require $\lambda = o(d^{3/4})$ and assume that there exists a constant $C$ such that for all pseudo-labelers, their test accuracy $p_k > 1/2 + C$.

Since we generate the noise patch from the Gaussian distribution, the strength of the noise patch is $\|\boldsymbol{\xi}\|_2^2 \approx d^{1+\epsilon}$ by standard concentration inequalities, which is larger than the strength of the signal patch $\|\mathbf{v}\|_2^2 = \Theta(d)$. Therefore, Condition 4.1 defines a setting with large noises. The condition of $d \gg n_u \gg n_l$ further ensures that learning is in a sufficiently over-parameterized setting. Here we only require the neural network width $m$ to be polylogarithmic in the dimension $d$ and require the psudolablers to perform better than a random guess.

**Theorem 4.2** (Semi-supervised Learning: Pre-training). Let $k \in [K]$ and consider the semi-supervised pre-training of $f_{\mathbf{W}_k}(\mathbf{x})$. For any test data point $(\mathbf{x}, y)$, denote $\widehat{y} = f_k^{\mathrm{w}}(\mathbf{x})$. Then under Condition 4.1, after $T_0 = \widetilde{\Theta}(d^{-\frac{3}{4}} \eta^{-1})$ training iterations with learning rate $\eta = O(d^{-1.1})$, the trained neural network can achieve nearly 0 test error on the distribution $\mathcal{D}$: $\mathbb{P}_{(\mathbf{x}, y) \sim \mathcal{D}}[y \cdot f_{\mathbf{W}_k^{(T_0)}}(\mathbf{x}) \leq 0] = o(1)$.

Theorem 4.2 characterizes the prediction power of the feature representation learned in the pre-trained models using unlabeled data. For any test data point $(\mathbf{x}, y)$, the sign of $y$ can be predicted based on $f_{\mathbf{W}^{(T_0)}}(\mathbf{x})$ with high probability.

**Theorem 4.3** (Semi-supervised Learning: Downstream). Let $\left\{f_{\mathbf{W}_k^{(T_0^k)}}\right\}_{k=1}^d$ be the neural networks trained according to the $K$ pre-training tasks, and consider the learning of the downstream task based in $\left\{f_{\mathbf{W}_k^{(T_0^k)}}\right\}_{k=1}^d$. Under Condition 4.1, after $T' = \Theta(d^{0.1}/\eta)$ iterations with learning rate $\eta = \Theta(1)$, with probability $1 - o(1)$, the obtained $\mathbf{a}^{(T')}$ satisfies:

- Training error is 0: $\frac{1}{n} \sum_{i=1}^n \mathbb{1}[y_i \cdot g_{\mathbf{a}^{(T')}}(\mathbf{x}_i) \leq 0] = 0$.

- Test error and loss are nearly 0: $\mathbb{P}_{(\mathbf{x}, y) \sim \mathcal{D}}[y \cdot g_{\mathbf{a}^{(T')}}(\mathbf{x}) \leq 0] = o(1)$, $L_{\mathcal{D}}(\mathbf{a}^{(T')}) = o(1)$.

Theorem 4.3 shows that the feature representation learned based on the semi-supervised pre-training can ensure small training and test errors for the supervised downstream task. Notably, this result holds even though we assume that there are only a constant number of labeled data. This shows that semi-supervised learning can significantly reduce the need for a large labeled training dataset. For comparison, we also have the following guarantees on the performance of standard supervised learning of CNNs.

**Theorem 4.4** (Supervised Learning). Under supervised learning setting, after gradient descent for $T = \widetilde{\Theta}(d^{(1/4-\epsilon)q-3/2} \eta^{-1})$ iterations with learning rate $\eta = O(d^{-1-2\epsilon})$, then there exists $t \leq T$ such that with probability $1 - o(1)$ the CNNs defined in (3.1) with parameter $\mathbf{W}^{(t)}$ satisfies:

- Training loss is nearly zero: $L_{S'}(\mathbf{W}^{(t)}) = o(1)$.

- Test loss is high: $L_{\mathcal{D}}(\mathbf{W}^{(t)}) = \Theta(1)$.

Theorem 4.4 shows that although standard supervised learning can train a CNN model with nearly zero training loss, the obtained CNN model generalizes poorly to test data. Comparing Theorem 4.4 with Theorem 4.3 shows that the generalization of semi-supervised learning and supervised learning are largely different. The reason behind this difference is that, the pre-training, with a relatively large number of unlabeled training data, helps learn a feature representation that captures the feature $\mathbf{v}$ in our data model, while direct application of supervised learning can only memorize the noises $\boldsymbol{\xi}_i'$, $i \in [n_l]$ in the training dataset, which is independent of the labels of the data.

A recent line of work (Oymak & Gulcu, 2021; Frei et al., 2022c) studies the semi-supervised learning methods with pseudo-labelers. Our results are different from theirs in several aspects: (i) we are considering learning with CNNs rather than a linear model, so the problem is highly non-convex with various local minima, which makes the optimization analysis more challenging; (ii) the Bayesian optimal predictor is no longer unique for CNNs. Therefore, we measure the quality of the learned features via downstream task instead of making a comparison with the Bayesian optimal predictor; (iii) They can only deal with the case where the teacher network (pseudo-labeler) is the same as the student network (Frei et al., 2022c) or the case where the teacher network (pseudo-labeler) is at least as complex as the student network (Oymak & Gulcu, 2021). However, our teacher network (pseudo-labeler) is not specified and can be any structure, such as a linear network. Therefore we can handle the case where the student network is more complex than the teacher network, one of the most natural settings for semi-supervised learning with pseudo-labeler (Xie et al., 2020).

## 5 PROOF SKETCH

In this section, we present the proof sketch for the semi-supervised learning setting. And the proof sketch of the supervised learning setting is given in the appendix.

**Semi-supervised Pre-training.** We consider learning $K$ functions $f_{\mathbf{W}_k}(\mathbf{x})$, $k \in [K]$ based on the pre-training. Since the learning process of these $K$ functions can be analyzed in exactly the same way, here we only focus on the learning of one of these functions. For simplicity of notation, we drop the subscript $k$ in the following proof sketch.

Our study of the pre-training focuses on two aspects of the training process: *feature learning* and *noise memorization*. Specifically, we aim to monitor how the filters in the CNN model learn the feature vector $\mathbf{v}$ and the noise vectors $\boldsymbol{\xi}_i$'s. Therefore, we introduce the following notations.

$$
\begin{aligned}
\widehat{\Lambda}_1^{(t)} &:= \max_{1 \leq j \leq m} \langle \mathbf{w}_j^{(t)}, \mathbf{v} \rangle, \ \bar{\Lambda}_1^{(t)} := \max_{1 \leq j \leq m} -\langle \mathbf{w}_j^{(t)}, \mathbf{v} \rangle, \\
\widehat{\Lambda}_{-1}^{(t)} &:= \max_{m+1 \leq j \leq 2m} -\langle \mathbf{w}_j^{(t)}, \mathbf{v} \rangle, \ \bar{\Lambda}_{-1}^{(t)} := \max_{m+1 \leq j \leq 2m} \langle \mathbf{w}_j^{(t)}, \mathbf{v} \rangle, \\
\Gamma_i^{(t)} &:= \max_{1 \leq j \leq 2m} \langle \mathbf{w}_j^{(t)}, \boldsymbol{\xi}_i \rangle, \ \Gamma_i'^{(t)} := \max_{1 \leq j \leq 2m} \langle \mathbf{w}_j^{(t)}, \boldsymbol{\xi}_i' \rangle, \ \Gamma^{(t)} = \max \left\{ \max_{i \in [n_u]} \Gamma_i^{(t)}, \max_{i \in [n_l]} \Gamma_i'^{(t)}, \right\}.
\end{aligned}
$$
(5.1)

Based on the above definitions for $r \in \{\pm 1\}$, a larger $\widehat{\Lambda}_r^{(t)}$ implies better feature learning along the positive feature direction $\mathbf{v}$, while a larger $\bar{\Lambda}_r^{(t)}$ implies better feature learning along the negative feature direction $-\mathbf{v}$. Moreover, a larger $\Gamma^{(t)}$ implies a higher level of noise memorization.

Based on the update rule of gradient descent, for the inner products $\langle \mathbf{w}_j^{(t)}, \mathbf{v} \rangle$ and $\langle \mathbf{w}_j^{(t)}, \boldsymbol{\xi}_l \rangle$, for $j \in [2m]$, $l \in [n_u]$, we can obtain iterative equations in (A.1). With the help of the iterative equations and definitions in (5.1), we can further show the following lemma.

**Lemma 5.1.** Assume we use both unlabeled data with pseudo-labels generated by the pseudo-labeler and labeled data for the training of our CNN model. Then for $r \in \{\pm 1\}$, let $T_r$ be the first iteration that $\widehat{\Lambda}_r^{(t)}$ reaches $\Theta(1/m)$, then for $t \in [0, T_r]$, we have

$$
\begin{aligned}
\widehat{\Lambda}_r^{(t+1)} &\geq (1 - \eta\lambda) \cdot \widehat{\Lambda}_r^{(t)} + \eta \cdot C \cdot \Theta(d) \cdot (\widehat{\Lambda}_r^{(t)})^{q-1}, r \in \{\pm 1\}, \\
\bar{\Lambda}_r^{(t+1)} &\leq (1 - \eta\lambda) \cdot \bar{\Lambda}_r^{(t)}, r \in \{\pm 1\}, \\
\Gamma^{(t+1)} &\leq (1 - \eta\lambda) \cdot \Gamma^{(t)} + \eta \cdot \widetilde{\Theta}(d^{1-2\epsilon}) \cdot (\Gamma^{(t)})^{q-1},
\end{aligned}
$$

where $C$ is defined in Condition 4.1.

**Lemma 5.2.** Assume we use only labeled data for the training of our CNN model. Then for $i \in [n_l]$, let $T_i'$ be the first iteration that $\Gamma_i'^{(t)}$ reaches $\Theta(1/m)$, then we have

$$\widehat{\Lambda}_r^{(t+1)} \leq (1 - \eta\lambda) \cdot \widehat{\Lambda}_r^{(t)} + \eta \cdot \Theta(d) \cdot \big((\widehat{\Lambda}_r^{(t)})^{q-1} + (\bar{\Lambda}_r^{(t)})^{q-1}\big), r \in \{\pm 1\},$$

$$\bar{\Lambda}_r^{(t+1)} \leq (1 - \eta\lambda) \cdot \bar{\Lambda}_r^{(t)}, r \in \{\pm 1\},$$

$$\Gamma_i'^{(t+1)} \geq (1 - \eta\lambda) \cdot \Gamma_i'^{(t)} + \eta \cdot \widetilde{\Theta}(d^{1+2\epsilon}) \cdot (\Gamma_i'^{(t)})^{q-1}, i \in [n_l], \text{ for } t \in [0, T_i'].$$

Based on the results in Lemma 5.1, we can observe that if both pseudo-labeled and labeled data are used for training, the CNN will learn the positive direction of the feature vector $\mathbf{v}$, while barely tending to fit the negative direction of the feature vector or memorize the noise. And if only labeled data are used, the CNN will fit noise faster than a feature, which can be seen from Lemma 5.2. Leveraging Lemmas 5.1 and 5.2, we can obtain following Lemmas 5.3 and 5.4, which characterize the magnitude of feature learning and noise memorization.

**Lemma 5.3.** If both pseudo-labeled and labeled data are used to train CNN, for $r \in \{\pm 1\}$, let $T_r$ be the first iteration that $\widehat{\Lambda}_r^{(t)}$ reaches $\Theta(1/m)$ respectively. Let $T_0 = \max_{r \in \{\pm 1\}} \{T_r\}$. Then, it holds that $\widehat{\Lambda}_r^{(T_0)} = \widetilde{\Theta}(1)$, $\bar{\Lambda}_r^{(t)} = \widetilde{O}(d^{-\frac{1}{4}})$ and $\Gamma^{(t)} = \widetilde{O}(d^{-\frac{1}{4}+\epsilon})$ for all $t \in [0, T_0]$.

**Lemma 5.4.** If only labeled data are used to train CNN, for $i \in [n_l]$, let $T_i'$ be the first iteration that $\Gamma_i'^{(t)}$ reaches $\Theta(1/m)$. Let $T_0' = \max_{i \in [n_l]} T_i'$. Then, it holds that $\widehat{\Lambda}_r = \widetilde{O}(d^{-\frac{1}{4}})$, $\bar{\Lambda}_r = \widetilde{O}(d^{-\frac{1}{4}})$ for $r \in \{\pm 1\}$ and $\Gamma_i'^{(t)} = \widetilde{\Theta}(1)$ for $i \in [n_l]$.

The above results indicate the deviation between the two settings. The reason is that assume we consider a sequence $\{x_t\}$ with iterative equation $x_{t+1} = x_t + \eta \cdot C_t x_t^{q-1}$. If we only use labeled data, as shown in Lemma 5.2, $\Gamma_i'^{(t)}$ has $C_t = \widetilde{\Theta}(d^{1+2\epsilon})$ while $\widehat{\Lambda}_r^{(t)}$ has $C_t = \Theta(d)$, therefore $\Gamma_i'^{(t)}$ increases faster than $\widehat{\Lambda}_r^{(t)}$. In contrast, if we use both labeled data and pseudo-labeled data, $C_t$ will be $\widetilde{\Theta}(d^{1-2\epsilon})$ for $\Gamma_i'^{(t)}$ and $\Theta(d)$ for $\widehat{\Lambda}_r^{(t)}$, leading to a slower increasing speed of $\Gamma_i'^{(t)}$.

**Downstream task.** After the pre-training, we have obtained $K$ CNN classifiers $\big\{ f_{\mathbf{W}_k^{(T_0^k)}} \big\}_{k=1}^K$. Now we train the second-layer parameters $\mathbf{a}$ with the training data whose true labels are available. The following lemma shows that the $l_1$-norm of $\mathbf{a}$ will increase with a logarithmic order.

**Lemma 5.5.** For any learning rate $\eta = \Theta(1)$, we have $\big\|\mathbf{a}^{(t)}\big\|_1 = \log(t)/\widetilde{\Theta}(1)$. For any labeled data $(\mathbf{x}_i', y_i') \in S'$, we have with high probability that $y_i' \cdot f_{\mathbf{W}^{(t)}}(\mathbf{x}_i') = \big\|\mathbf{a}^{(t)}\big\|_1 \cdot \widetilde{\Theta}(1)$. For any newly generated data $(\mathbf{x}, y) \sim \mathcal{D}$, we also have with high probability that $y \cdot f_{\mathbf{W}^{(t)}}(\mathbf{x}) = \big\|\mathbf{a}^{(t)}\big\|_1 \cdot \widetilde{\Theta}(1)$.

With the help of the above lemma and note that training error and test error are related to $y \cdot f_{\mathbf{W}^{(T_0)}}(\mathbf{x})$ and test loss is related to $\|\mathbf{a}^{(T_0)}\|_1$, we can prove that after $T = \Theta(d^{0.1}/\eta)$ iterations with learning rate $\eta = \Theta(1)$, the model can achieve nearly zero training error, test error, training loss and test loss.

## 6 EXPERIMENTS

In this section, we perform numerical experiments on synthetic datasets, generated according to Definition 3.1, to verify our main theoretical results. The code and data for our experiments can be found on Github [1]. In particular, we set the problem dimension $d = 10000$, labeled training sample size $n_l = 20$ (10 positive samples and 10 negative samples), pseudo-labeled training sample size $n_u = 20000$ (10000 positive samples and 10000 negative samples), feature vector $\mathbf{v}$ sampled from distribution $\mathcal{N}(\mathbf{0}, \mathbf{I})$ and noise vector sampled from distribution $\mathcal{N}(\mathbf{0}, \sigma_p^2 \mathbf{I})$ where $\sigma_p = 10d^{0.01}$.

For semi-supervised learning task, we have a linear pseudo-labeler with test error $0.196 \pm 0.044$. Then, we use this classifier to generate pseudo-labels for $n_u = 20000$ unlabeled samples in order to help semi-supervised learning. After that, for pre-training, we use these pseudo-labeled samples and $n_l$ labeled samples together to train a CNN with network width $m = 20$, activation function $\sigma(z) = [z]_+^3$, regularization parameter $\lambda = 0.1$ and learning rate $\eta = 1 \times 10^{-4}$. Besides, we

---

[1]https://github.com/uclaml/SSL_Pseudo_Labeler

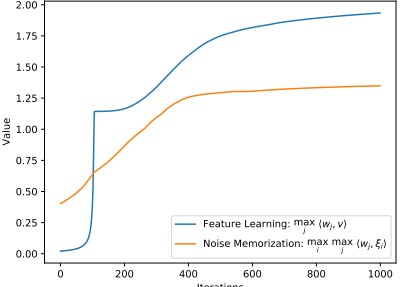 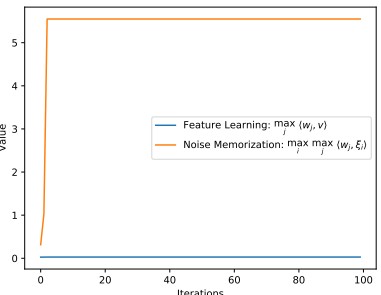

Figure 3: Visualization of the feature learning and noise memorization in the training process. (Left: Semi-supervised, Right: Supervised)

| | Semi-supervised | | Supervised |
|---|---|---|---|
| | Pre-train | Downstream | |
| Training error | $0.1753 \pm 0.0259$ | 0 | 0 |
| Test error | 0 | 0 | $0.4982 \pm 0.0208$ |
| Training loss | $0.4155 \pm 0.0418$ | $0.0150 \pm 0.0022$ | $(6.473 \pm 5.031) \times 10^{-7}$ |
| Test loss | $0.2200 \pm 0.0886$ | $0.0182 \pm 0.0021$ | $0.6931 \pm 0.0005$ |

Table 1: Training error and loss, test error and loss for semi-supervised and supervised learning.

initialize CNN parameters from $\mathcal{N}(0, \sigma_0^2)$, where $\sigma_0 = 0.1 \times d^{-3/4}$. After 200 iterations, we can obtain a CNN model with training error close to the error of pseudo-labeler and zero test error, according to Table 6. And for a downstream task, we use $n_l$ labeled samples to train a linear probe. By applying learning rate $\eta = 0.1$ and after $T = 100$ iterations, we can obtain a final model with low training and test loss as well as $100\%$ training accuracy and test accuracy.

For supervised learning task, we directly use $n_l$ labeled data to train a CNN with network width $m = 20$, activation function $\sigma(z) = [z]_+^3$, learning rate $\eta = 1 \times 10^{-4}$. After 200 iterations, we obtain a CNN with 0 training error and small training loss, about $0.5$ test error and high test loss, which indicates supervised learning will give a model that behaves badly and even no better than a random guess.

Moreover, for synthetic data experiments, we also calculate the inner products $\max_{j \in [m]} \langle \mathbf{w}_j^{(t)}, \mathbf{v} \rangle$ and $\max_{j \in [2m]} \left\{ \max_{i \in [n_u]} \langle \mathbf{w}_j^{(t)}, \boldsymbol{\xi}_i \rangle, \max_{i \in [n_l]} \langle \mathbf{w}_j^{(t)}, \boldsymbol{\xi}_i' \rangle \right\}$, i.e. $\widehat{\Lambda}_1^{(t)}$ and $\Gamma^{(t)}$, representing feature learning and noise memorization respectively, to verify our key lemmas. The results are reported in Figure 3. It can be seen from Figure 3 that under semi-supervised learning setting the algorithm will the feature learning will dominate the noise memorization though the noise patch has a larger norm than the signal patch, while under the supervised learning setting, the algorithm will entirely forget the feature but fit noise. This verifies Lemmas 5.3 and 5.4.

# 7 CONCLUSION AND FUTURE WORK

In this paper, we study semi-supervised learning with pseudo-labelers and provide a theoretical understanding of the success of semi-supervised learning. We show the advantage of semi-supervised learning over supervised learning through a case study. By considering a simple data model and two-layer CNN, we present a comprehensive analysis of the training procedure from a beyond-NTK feature learning perspective. We prove that the final classifier of a semi-supervised learning scenario can achieve near-zero test loss and error with only a small number of labeled training data, while its supervised-learned counterpart fails to achieve the same performance with the same data complexity.

In the current paper, we only focus on the simplest possible data and neural network models to study semi-supervised learning. For example, the second layer of the CNN is fixed during the training. What if the second layer is trainable? In addition, the stride is the same as the filter size in the current CNN, and it is reasonable to have the stride be smaller than the filter size. On the other hand, it would be interesting to consider linearly non-separable data (Shi et al., 2022; Frei et al., 2022b; Damian et al., 2022) and ReLU activation function with the help of pre-activation noise (Allen-Zhu & Li, 2020a). We leave these extensions as future works.

## ACKNOWLEDGEMENTS

We thank the anonymous reviewers for their helpful comments. YK, ZC and QG are supported in part by the National Science Foundation IIS-2008981 and the Sloan Research Fellowship.

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

## A  PROOF FOR SEMI-SUPERVISED LEARNING SETTING

We consider learning $K$ functions $f_{\mathbf{W}_k}(\mathbf{x})$, $k \in [K]$ based on the pre-training. Since the learning process of these $K$ functions can be analyzed in exactly the same way, here we only focus on the learning of one of these functions. For simplicity of notation, we drop the subscript $k$ in the following proof for Sections A.1, A.2, A.3, A.4, A.5, A.6 and A.7.

### A.1  GRADIENT CALCULATION

**Lemma A.1** (Gradient Calculation). The gradient of loss function $L_S(\mathbf{W})$ with respect to weight parameters $\mathbf{w}_j$ is

$$\nabla_{\mathbf{w}_j} L_{S \cup S'}(\mathbf{W}) = -\frac{q}{n_l + n_u} \bigg( \sum_{i=1}^{n_u} c_i \widehat{y}_i \big( [\langle \mathbf{w}_j, y_i \cdot \mathbf{v} \rangle]_+^{q-1} \cdot y_i \cdot \mathbf{v} + [\langle \mathbf{w}_j, \boldsymbol{\xi}_i \rangle]_+^{q-1} \cdot \boldsymbol{\xi}_i \big)$$
$$+ \sum_{i=1}^{n_l} b_i y_i' \big( [\langle \mathbf{w}_j, y_i' \cdot \mathbf{v} \rangle]_+^{q-1} \cdot y_i' \cdot \mathbf{v} + [\langle \mathbf{w}_j, \boldsymbol{\xi}_i' \rangle]_+^{q-1} \cdot \boldsymbol{\xi}_i' \big) \bigg) + \lambda \cdot \mathbf{w}_j,$$

for $1 \leq j \leq m$; and

$$\nabla_{\mathbf{w}_j} L_{S \cup S'}(\mathbf{W}) = \frac{q}{n_l + n_u} \bigg( \sum_{i=1}^{n_u} c_i \widehat{y}_i \big( [\langle \mathbf{w}_j, y_i \cdot \mathbf{v} \rangle]_+^{q-1} \cdot y_i \cdot \mathbf{v} + [\langle \mathbf{w}_j, \boldsymbol{\xi}_i \rangle]_+^{q-1} \cdot \boldsymbol{\xi}_i \big)$$
$$+ \sum_{i=1}^{n_l} b_i y_i' \big( [\langle \mathbf{w}_j, y_i' \cdot \mathbf{v} \rangle]_+^{q-1} \cdot y_i' \cdot \mathbf{v} + [\langle \mathbf{w}_j, \boldsymbol{\xi}_i' \rangle]_+^{q-1} \cdot \boldsymbol{\xi}_i' \big) \bigg) + \lambda \cdot \mathbf{w}_j,$$

for $m + 1 \leq j \leq 2m$, where $-\ell'(\widehat{y}_i \cdot f_{\mathbf{W}}(\mathbf{x}_i)) = \exp[-\widehat{y}_i \cdot f_{\mathbf{W}}(\mathbf{x}_i)]/(1 + \exp[-\widehat{y}_i \cdot f_{\mathbf{W}}(\mathbf{x}_i)])$ is denoted by $c_i$ and $-\ell'(y_i' \cdot f_{\mathbf{W}}(\mathbf{x}_i')) = \exp[-y_i' \cdot f_{\mathbf{W}}(\mathbf{x}_i')]/(1 + \exp[-y_i' \cdot f_{\mathbf{W}}(\mathbf{x}_i')])$ is denoted by $b_i$.

*Proof of Lemma A.1.* When $1 \leq j \leq m$,

$$\nabla_{\mathbf{w}_j} \ell(\widehat{y}_i \cdot f_{\mathbf{W}}(\mathbf{x}_i)) = \ell'(\widehat{y}_i \cdot f_{\mathbf{W}}(\mathbf{x}_i)) \cdot \widehat{y}_i \cdot \nabla_{\mathbf{w}_j} f_{\mathbf{W}}(\mathbf{x}_i)$$
$$= -c_i \cdot \widehat{y}_i \cdot \nabla_{\mathbf{w}_j} f_{\mathbf{W}}(\mathbf{x}_i)$$
$$= -c_i \widehat{y}_i \cdot \big( \sigma'(\langle \mathbf{w}_j, y_i \cdot \mathbf{v} \rangle) \cdot y_i \cdot \mathbf{v} + \sigma'(\langle \mathbf{w}_j, \boldsymbol{\xi}_i \rangle) \cdot \boldsymbol{\xi}_i \big)$$
$$= -q c_i \widehat{y}_i \big( [\langle \mathbf{w}_j, y_i \cdot \mathbf{v} \rangle]_+^{q-1} \cdot y_i \cdot \mathbf{v} + [\langle \mathbf{w}_j, \boldsymbol{\xi}_i \rangle]_+^{q-1} \cdot \boldsymbol{\xi}_i \big)$$

$$\nabla_{\mathbf{w}_j} \ell(y_i' \cdot f_{\mathbf{W}}(\mathbf{x}_i')) = \ell'(y_i' \cdot f_{\mathbf{W}}(\mathbf{x}_i')) \cdot y_i' \cdot \nabla_{\mathbf{w}_j} f_{\mathbf{W}}(\mathbf{x}_i')$$
$$= -b_i \cdot y_i' \cdot \nabla_{\mathbf{w}_j} f_{\mathbf{W}}(\mathbf{x}_i')$$
$$= -b_i y_i' \cdot \big( \sigma'(\langle \mathbf{w}_j, y_i' \cdot \mathbf{v} \rangle) \cdot y_i' \cdot \mathbf{v} + \sigma'(\langle \mathbf{w}_j, \boldsymbol{\xi}_i' \rangle) \cdot \boldsymbol{\xi}_i' \big)$$
$$= -q b_i y_i' \cdot \big( [\langle \mathbf{w}_j, y_i' \cdot \mathbf{v} \rangle]_+^{q-1} \cdot y_i' \cdot \mathbf{v} + [\langle \mathbf{w}_j, \boldsymbol{\xi}_i' \rangle]_+^{q-1} \cdot \boldsymbol{\xi}_i' \big)$$

and when $m + 1 \leq j \leq 2m$,

$$\nabla_{\mathbf{w}_j} \ell(\widehat{y}_i \cdot f_{\mathbf{W}}(\mathbf{x}_i)) = q c_i \widehat{y}_i \big( [\langle \mathbf{w}_j, y_i \cdot \mathbf{v} \rangle]_+^{q-1} \cdot y_i \cdot \mathbf{v} + [\langle \mathbf{w}_j, \boldsymbol{\xi}_i \rangle]_+^{q-1} \cdot \boldsymbol{\xi}_i \big)$$
$$\nabla_{\mathbf{w}_j} \ell(y_i' \cdot f_{\mathbf{W}}(\mathbf{x}_i')) = q b_i y_i' \cdot \big( [\langle \mathbf{w}_j, y_i' \cdot \mathbf{v} \rangle]_+^{q-1} \cdot y_i' \cdot \mathbf{v} + [\langle \mathbf{w}_j, \boldsymbol{\xi}_i' \rangle]_+^{q-1} \cdot \boldsymbol{\xi}_i' \big)$$

Note that $\nabla_{\mathbf{w}_j} L_{S \cup S'}(\mathbf{W}) = \big( \sum_{i=1}^{n_u} \nabla_{\mathbf{w}_j} \ell(\widehat{y}_i \cdot f_{\mathbf{W}}(\mathbf{x}_i)) + \sum_{i=1}^{n_l} \nabla_{\mathbf{w}_j} \ell(y_i' \cdot f_{\mathbf{W}}(\mathbf{x}_i')) \big)/(n_l + n_u) + \lambda \cdot \mathbf{w}_j$, we have proved the lemma. $\square$

### A.2  INNER PRODUCT UPDATE RULE CALCULATION

When the model is trained by gradient descent, the update rule can be formulated by

$$\mathbf{w}_j^{(t+1)} = \mathbf{w}_j^{(t)} - \eta \cdot \nabla_{\mathbf{w}_j} L_S(\mathbf{W}^{(t)}), \quad j \in [2m]. \tag{A.1}$$

We study the performance of entire training process from two perspective: feature learning and noise memorization. Mathematically, we will focus on two quantities: $\langle \mathbf{w}_j^{(t)}, \mathbf{v} \rangle$ and $\langle \mathbf{w}_j^{(t)}, \boldsymbol{\xi}_l \rangle$. And then we have following lemma for the inner product update rule.

**Lemma A.2** (Inner Product Update Rule). The feature learning and noise memorization performance of gradient descent can be formulated by

$$\langle \mathbf{w}_j^{(t+1)}, \mathbf{v} \rangle = (1 - \eta\lambda) \cdot \langle \mathbf{w}_j^{(t)}, \mathbf{v} \rangle + \frac{q\eta u_j}{n_{\mathrm{l}} + n_{\mathrm{u}}} \bigg( \sum_{i=1}^{n_{\mathrm{u}}} y_i \widehat{y}_i c_i^{(t)} [\langle \mathbf{w}_j^{(t)}, y_i \cdot \mathbf{v} \rangle]_+^{q-1} \|\mathbf{v}\|_2^2$$

$$+ \sum_{i=1}^{n_{\mathrm{l}}} b_i^{(t)} [\langle \mathbf{w}_j^{(t)}, y_i' \cdot \mathbf{v} \rangle]_+^{q-1} \|\mathbf{v}\|_2^2 \bigg),$$

$$\langle \mathbf{w}_j^{(t+1)}, \boldsymbol{\xi}_l \rangle = (1 - \eta\lambda) \cdot \langle \mathbf{w}_j^{(t)}, \boldsymbol{\xi}_l \rangle + \frac{q\eta u_j}{n_{\mathrm{l}} + n_{\mathrm{u}}} \bigg( \sum_{i=1}^{n_{\mathrm{u}}} \widehat{y}_i c_i^{(t)} [\langle \mathbf{w}_j^{(t)}, \boldsymbol{\xi}_i \rangle]_+^{q-1} \langle \boldsymbol{\xi}_i, \boldsymbol{\xi}_l \rangle$$

$$+ \sum_{i=1}^{n_{\mathrm{l}}} y_i' b_i^{(t)} [\langle \mathbf{w}_j^{(t)}, \boldsymbol{\xi}_i' \rangle]_+^{q-1} \langle \boldsymbol{\xi}_i', \boldsymbol{\xi}_l \rangle \bigg),$$

$$\langle \mathbf{w}_j^{(t+1)}, \boldsymbol{\xi}_l' \rangle = (1 - \eta\lambda) \cdot \langle \mathbf{w}_j^{(t)}, \boldsymbol{\xi}_l' \rangle + \frac{q\eta u_j}{n_{\mathrm{l}} + n_{\mathrm{u}}} \bigg( \sum_{i=1}^{n_{\mathrm{u}}} \widehat{y}_i c_i^{(t)} [\langle \mathbf{w}_j^{(t)}, \boldsymbol{\xi}_i \rangle]_+^{q-1} \langle \boldsymbol{\xi}_i, \boldsymbol{\xi}_l' \rangle$$

$$+ \sum_{i=1}^{n_{\mathrm{l}}} y_i' b_i^{(t)} [\langle \mathbf{w}_j^{(t)}, \boldsymbol{\xi}_i' \rangle]_+^{q-1} \langle \boldsymbol{\xi}_i', \boldsymbol{\xi}_l' \rangle \bigg),$$

where $j \in [2m]$, $l \in [n_{\mathrm{u}}]$ and $u_j := \mathbb{1}_{[1 \le j \le m]} - \mathbb{1}_{[m+1 \le j \le 2m]}$.

*Proof of Lemma A.2.* According to Lemma A.1 and gradient descent update rule (A.1), we have

$$\mathbf{w}_j^{(t+1)} = (1 - \eta\lambda) \cdot \mathbf{w}_j^{(t)} + \frac{q\eta u_j}{n_{\mathrm{l}} + n_{\mathrm{u}}} \cdot \bigg( \sum_{i=1}^{n_{\mathrm{u}}} c_i \widehat{y}_i \big( [\langle \mathbf{w}_j, y_i \cdot \mathbf{v} \rangle]_+^{q-1} \cdot y_i \cdot \mathbf{v} + [\langle \mathbf{w}_j, \boldsymbol{\xi}_i \rangle]_+^{q-1} \cdot \boldsymbol{\xi}_i \big)$$

$$+ \sum_{i=1}^{n_{\mathrm{l}}} b_i y_i' \big( [\langle \mathbf{w}_j, y_i' \cdot \mathbf{v} \rangle]_+^{q-1} \cdot y_i' \cdot \mathbf{v} + [\langle \mathbf{w}_j, \boldsymbol{\xi}_i' \rangle]_+^{q-1} \cdot \boldsymbol{\xi}_i' \big) \bigg)$$

Taking inner product with feature vector $\mathbf{v}$ and noise patch $\boldsymbol{\xi}_l$ and note that $\mathbf{v}$ is orthogonal to $\boldsymbol{\xi}_l$ according to the data model, we have

$$\langle \mathbf{w}_j^{(t+1)}, \mathbf{v} \rangle = (1 - \eta\lambda) \cdot \langle \mathbf{w}_j^{(t)}, \mathbf{v} \rangle + \frac{q\eta u_j}{n_{\mathrm{l}} + n_{\mathrm{u}}} \bigg( \sum_{i=1}^{n_{\mathrm{u}}} c_i^{(t)} \widehat{y}_i \big( [\langle \mathbf{w}_j, y_i \cdot \mathbf{v} \rangle]_+^{q-1} y_i \|\mathbf{v}\|_2^2 + [\langle \mathbf{w}_j, \boldsymbol{\xi}_i \rangle]_+^{q-1} \langle \boldsymbol{\xi}_i, \mathbf{v} \rangle \big)$$

$$+ \sum_{i=1}^{n_{\mathrm{l}}} b_i^{(t)} y_i' \big( [\langle \mathbf{w}_j, y_i' \cdot \mathbf{v} \rangle]_+^{q-1} y_i' \|\mathbf{v}\|_2^2 + [\langle \mathbf{w}_j, \boldsymbol{\xi}_i' \rangle]_+^{q-1} \langle \boldsymbol{\xi}_i', \mathbf{v} \rangle \big) \bigg)$$

$$= (1 - \eta\lambda) \cdot \langle \mathbf{w}_j^{(t)}, \mathbf{v} \rangle + \frac{q\eta u_j}{n_{\mathrm{l}} + n_{\mathrm{u}}} \bigg( \sum_{i=1}^{n_{\mathrm{u}}} y_i \widehat{y}_i c_i^{(t)} [\langle \mathbf{w}_j^{(t)}, y_i \cdot \mathbf{v} \rangle]_+^{q-1} \|\mathbf{v}\|_2^2$$

$$+ \sum_{i=1}^{n_{\mathrm{l}}} b_i^{(t)} [\langle \mathbf{w}_j^{(t)}, y_i' \cdot \mathbf{v} \rangle]_+^{q-1} \|\mathbf{v}\|_2^2 \bigg),$$

$$\langle \mathbf{w}_j^{(t+1)}, \boldsymbol{\xi}_l \rangle = (1 - \eta\lambda) \cdot \langle \mathbf{w}_j^{(t)}, \boldsymbol{\xi}_l \rangle + \frac{q\eta u_j}{n_{\mathrm{l}} + n_{\mathrm{u}}} \bigg( \sum_{i=1}^{n_{\mathrm{u}}} c_i^{(t)} \widehat{y}_i \big( [\langle \mathbf{w}_j, y_i \cdot \mathbf{v} \rangle]_+^{q-1} y_i \langle \mathbf{v}, \boldsymbol{\xi}_l \rangle + [\langle \mathbf{w}_j, \boldsymbol{\xi}_i \rangle]_+^{q-1} \langle \boldsymbol{\xi}_i, \boldsymbol{\xi}_l \rangle \big)$$

$$+ \sum_{i=1}^{n_{\mathrm{l}}} b_i^{(t)} y_i' \big( [\langle \mathbf{w}_j, y_i' \cdot \mathbf{v} \rangle]_+^{q-1} y_i' \langle \mathbf{v}, \boldsymbol{\xi}_l \rangle + [\langle \mathbf{w}_j, \boldsymbol{\xi}_i' \rangle]_+^{q-1} \langle \boldsymbol{\xi}_i', \boldsymbol{\xi}_l \rangle \big) \bigg)$$

$$= (1 - \eta\lambda) \cdot \langle \mathbf{w}_j^{(t)}, \boldsymbol{\xi}_l \rangle + \frac{q\eta u_j}{n_{\mathrm{l}} + n_{\mathrm{u}}} \bigg( \sum_{i=1}^{n_{\mathrm{u}}} \widehat{y}_i c_i^{(t)} [\langle \mathbf{w}_j^{(t)}, \boldsymbol{\xi}_i \rangle]_+^{q-1} \langle \boldsymbol{\xi}_i, \boldsymbol{\xi}_l \rangle$$

$$+ \sum_{i=1}^{n_{\mathrm{l}}} y_i' b_i^{(t)} [\langle \mathbf{w}_j^{(t)}, \boldsymbol{\xi}_i' \rangle]_+^{q-1} \langle \boldsymbol{\xi}_i', \boldsymbol{\xi}_l \rangle \bigg),$$

and

$$\langle \mathbf{w}_j^{(t+1)}, \boldsymbol{\xi}_l' \rangle = (1 - \eta\lambda) \cdot \langle \mathbf{w}_j^{(t)}, \boldsymbol{\xi}_l' \rangle + \frac{q\eta u_j}{n_\mathrm{l} + n_\mathrm{u}} \bigg( \sum_{i=1}^{n_\mathrm{u}} \widehat{y}_i c_i^{(t)} [\langle \mathbf{w}_j^{(t)}, \boldsymbol{\xi}_i \rangle]_+^{q-1} \langle \boldsymbol{\xi}_i, \boldsymbol{\xi}_l' \rangle$$

$$+ \sum_{i=1}^{n_\mathrm{l}} y_i' b_i^{(t)} [\langle \mathbf{w}_j^{(t)}, \boldsymbol{\xi}_i' \rangle]_+^{q-1} \langle \boldsymbol{\xi}_i', \boldsymbol{\xi}_l' \rangle \bigg),$$

which completes the proof. $\qquad \square$

### A.3 ESTIMATE $\widehat{\Lambda}_r^{(0)}, \bar{\Lambda}_r^{(0)}, \Gamma_i^{(0)}, \Gamma_i'^{(0)}$

Let $\widehat{\Lambda}_1^{(t)} = \max_{1 \le j \le m} \langle \mathbf{w}_j^{(t)}, \mathbf{v} \rangle$, $\widehat{\Lambda}_{-1}^{(t)} = \max_{m+1 \le j \le 2m} -\langle \mathbf{w}_j^{(t)}, \mathbf{v} \rangle$, $\bar{\Lambda}_1^{(t)} = \max_{m+1 \le j \le 2m} \langle \mathbf{w}_j^{(t)}, \mathbf{v} \rangle$, $\bar{\Lambda}_{-1}^{(t)} = \max_{1 \le j \le m} -\langle \mathbf{w}_j^{(t)}, \mathbf{v} \rangle$, which characterize the *feature learning* aspect of training process. An easy way to distinguish between $\widehat{\Lambda}_r^{(t)}$ and $\bar{\Lambda}_r^{(t)}$ is that $\widehat{\Lambda}_r^{(t)}$ should be large while $\bar{\Lambda}_r^{(t)}$ should be small.

Let $\Gamma_i^{(t)} = \max_{1 \le j \le 2m} \langle \mathbf{w}_j, \boldsymbol{\xi}_i \rangle, i \in [n_\mathrm{u}]$, $\Gamma_i'^{(t)} = \max_{1 \le j \le 2m} \langle \mathbf{w}_j, \boldsymbol{\xi}_i' \rangle, i \in [n_\mathrm{l}]$, which characterize the *noise memorization* aspect of training process with respect to a particular sample.

Let $\Gamma^{(t)} = \max \big\{ \max_{i \in [n_\mathrm{u}]} \Gamma_i^{(t)}, \max_{i \in [n_\mathrm{l}]} \Gamma_i'^{(t)} \big\}$, which characterize the *noise memorization* aspect of training process regardless of the sample index.

We first provide the concentration inequality for $\widehat{\Lambda}_r^{(0)}$ and $\bar{\Lambda}_r^{(0)}$ in the following lemma.

**Lemma A.3.** With probability at least $1 - 4\delta$ with respect to the randomness of initialization of $\mathbf{w}$, we have

$$\big| \widehat{\Lambda}_r^{(0)} - \mathbb{E}[\widehat{\Lambda}_r^{(0)}] \big| < \sqrt{8 \log\Big(\frac{1}{\delta}\Big)} \sigma_0 \|\mathbf{v}\|_2,$$

$$\big| \bar{\Lambda}_r^{(0)} - \mathbb{E}[\bar{\Lambda}_r^{(0)}] \big| < \sqrt{8 \log\Big(\frac{1}{\delta}\Big)} \sigma_0 \|\mathbf{v}\|_2,$$

and

$$\mathbb{E}[\widehat{\Lambda}_r^{(0)}] \asymp \sqrt{\log(m)} \sigma_0 \|\mathbf{v}\|_2, \mathbb{E}[\bar{\Lambda}_r^{(0)}] \asymp \sqrt{\log(m)} \sigma_0 \|\mathbf{v}\|_2, r \in \{\pm 1\}.$$

*Proof of Lemma A.3.* Note that $\widehat{\Lambda}_1^{(0)} = \max_{1 \le j \le m} \langle \mathbf{w}_j^{(0)}, \mathbf{v} \rangle$, $\widehat{\Lambda}_{-1}^{(0)} = \max_{m+1 \le j \le 2m} -\langle \mathbf{w}_j^{(0)}, \mathbf{v} \rangle$, $\bar{\Lambda}_1^{(0)} = \max_{m+1 \le j \le 2m} \langle \mathbf{w}_j^{(0)}, \mathbf{v} \rangle$ and $\bar{\Lambda}_{-1}^{(0)} = \max_{m+1 \le j \le 2m} -\langle \mathbf{w}_j^{(0)}, \mathbf{v} \rangle$, $\mathbf{w}_j^{(0)} \sim \mathcal{N}(\mathbf{0}, \sigma_0^2 \mathbf{I})$ and $\mathbf{v}$ is a fixed vector. Therefore, $\langle \mathbf{w}_j^{(0)}, \mathbf{v} \rangle \sim \mathcal{N}(0, \sigma_0^2 \|\mathbf{v}\|_2^2)$, $-\langle \mathbf{w}_j^{(0)}, \mathbf{v} \rangle \sim \mathcal{N}(0, \sigma_0^2 \|\mathbf{v}\|_2^2)$ for all $1 \le j \le 2m$ and $\widehat{\Lambda}_r^{(0)}, \bar{\Lambda}_r^{(0)}, r \in \{\pm 1\}$ are identically distributed. Therefore, without loss of generality, we only need to discuss the concentration of $\widehat{\Lambda}_1^{(0)}$. By applying Lemma C.1, we have

$$\mathbb{P}\Big( \big| \widehat{\Lambda}_1^{(0)} - \mathbb{E}[\widehat{\Lambda}_1^{(0)}] \big| > t \Big) \le 2e^{-\frac{t^2}{2\sigma_0^2 \|\mathbf{v}\|_2^2}}.$$

By applying Lemma C.2, we have

$$\mathbb{E}[\widehat{\Lambda}_1^{(0)}] \asymp \sqrt{\log(m)} \sigma_0 \|\mathbf{v}\|_2,$$

which completes the proof. $\qquad \square$

Then we provide concentration inequality for $\Gamma_i^{(0)}$ in the following lemma.

**Lemma A.4.** Suppose that $d \ge \Omega(\log(m(n_\mathrm{u} + n_\mathrm{l})/\delta))$, $m = \Omega(\log(1/\delta))$. Then with probability at least $1 - \delta$,

$$\frac{\sigma_0 \sigma_p \sqrt{d}}{4} \le \Gamma_i^{(0)} \le 2\sqrt{\log(16m(n_\mathrm{u} + n_\mathrm{l})/\delta)} \cdot \sigma_0 \sigma_p \sqrt{d}, \text{ for all } i \in [n_\mathrm{u}],$$

$$\frac{\sigma_0 \sigma_p \sqrt{d}}{4} \le \Gamma_i'^{(0)} \le 2\sqrt{\log(16m(n_\mathrm{u} + n_\mathrm{l})/\delta)} \cdot \sigma_0 \sigma_p \sqrt{d}, \text{ for all } i \in [n_\mathrm{l}].$$

*Proof of Lemma A.4.* By Lemma C.3, with probability at least $1 - \delta/4$,

$$
\begin{aligned}
\sigma_p \sqrt{d}/\sqrt{2} \leq \|\boldsymbol{\xi}_i\|_2 \leq \sqrt{3/2} \cdot \sigma_p \sqrt{d}, \text{ for } i \in [n_{\mathrm{u}}], \\
\sigma_p \sqrt{d}/\sqrt{2} \leq \|\boldsymbol{\xi}_i'\|_2 \leq \sqrt{3/2} \cdot \sigma_p \sqrt{d}, \text{ for } i \in [n_{\mathrm{l}}].
\end{aligned}
\tag{A.2}
$$

Therefore, by Gaussian tail bound and union bound, with probability at least $1 - \delta/4$,

$$
\begin{aligned}
\langle \mathbf{w}_j^{(0)}, \boldsymbol{\xi}_i \rangle \leq |\langle \mathbf{w}_j^{(0)}, \boldsymbol{\xi}_i \rangle| \leq \sqrt{2\log(8m/\delta)} \cdot \sigma_0 \|\boldsymbol{\xi}_i\|_2, \text{ for } i \in [n_{\mathrm{u}}], \\
\langle \mathbf{w}_j^{(0)}, \boldsymbol{\xi}_i' \rangle \leq |\langle \mathbf{w}_j^{(0)}, \boldsymbol{\xi}_i' \rangle| \leq \sqrt{2\log(8m/\delta)} \cdot \sigma_0 \|\boldsymbol{\xi}_i'\|_2, \text{ for } i \in [n_{\mathrm{l}}].
\end{aligned}
\tag{A.3}
$$

Note that $\mathbb{P}\big(\sigma_0 \sigma_p \sqrt{d}/4 > \langle \mathbf{w}_j^{(0)}, \boldsymbol{\xi}_i \rangle\big)$ is an absolute constant and therefore by the condition on $m$, we have

$$
\begin{aligned}
\mathbb{P}\left( \frac{\sigma_0 \sigma_p \sqrt{d}}{4} \leq \Gamma_i^{(t)} \right) &= \mathbb{P}\left( \frac{\sigma_0 \sigma_p \sqrt{d}}{4} \leq \max_{j \in [2m]} \langle \mathbf{w}_j^{(0)}, \boldsymbol{\xi}_i \rangle \right) \\
&= 1 - \mathbb{P}\left( \frac{\sigma_0 \sigma_p \sqrt{d}}{4} > \max_{j \in [2m]} \langle \mathbf{w}_j^{(0)}, \boldsymbol{\xi}_i \rangle \right) \\
&= 1 - \left( \mathbb{P}\left( \frac{\sigma_0 \sigma_p \sqrt{d}}{4} > \langle \mathbf{w}_j^{(0)}, \boldsymbol{\xi}_i \rangle \right) \right)^{2m} \\
&\geq 1 - \frac{\delta}{4},
\end{aligned}
$$

and

$$
\mathbb{P}\left( \frac{\sigma_0 \sigma_p \sqrt{d}}{4} \leq \Gamma_i'^{(t)} \right) \geq 1 - \frac{\delta}{4}.
$$

On the other hand, according to (A.2) and (A.3), we have

$$
\begin{aligned}
&\mathbb{P}\big( \Gamma_i^{(t)} \leq 2\sqrt{\log(16m(n_{\mathrm{u}} + n_{\mathrm{l}})/\delta)} \cdot \sigma_0 \sigma_p \sqrt{d} \big) \\
&= \mathbb{P}\Big( \max_{j \in [2m]} \langle \mathbf{w}_j^{(0)}, \boldsymbol{\xi}_i \rangle \leq 2\sqrt{\log(16m(n_{\mathrm{u}} + n_{\mathrm{l}})/\delta)} \cdot \sigma_0 \sigma_p \sqrt{d} \Big) \\
&\geq 1 - \frac{\delta}{4},
\end{aligned}
$$

and

$$
\mathbb{P}\big( \Gamma_i'^{(t)} \leq 2\sqrt{\log(16m(n_{\mathrm{u}} + n_{\mathrm{l}})/\delta)} \cdot \sigma_0 \sigma_p \sqrt{d} \big) \geq 1 - \frac{\delta}{4},
$$

which completes the proof. $\qquad\square$

### A.4 STAGE I OF GD: ON-DIAGONAL FEATURE LEARNING

In this stage, $\widehat{\Lambda}_1^{(t)}$ and $\widehat{\Lambda}_{-1}^{(t)}$ respectively increase to magnitude $\Theta(1/m)$ and $\bar{\Lambda}_1^{(t)}$, $\bar{\Lambda}_{-1}^{(t)}$ and $\Gamma_j^{(t)}$ remain small, the same magnitude as initialization. In order to characterize the behaviour of feature learning and noise memorization during Stage I, we decompose the analysis into following three parts:

1. First, in Lemma A.9, we provide a lower bound of the update rules of on-diagonal feature learning term of $\widehat{\Lambda}_1^{(t)}, \widehat{\Lambda}_{-1}^{(t)}$ to lower-bound their increasing speed, and an upper bound of off-diagonal feature learning term $\bar{\Lambda}_1^{(t)}, \bar{\Lambda}_{-1}^{(t)}$ to indicate their decrease.

2. Second, in Lemma A.11, we provide a upper bound of the update rules of noise memorization term $\Gamma^{(t)}$ to upper-bound its increasing speed.

3. Third, we provide a useful lemma, which is a derivation of Claim C.20 in Allen-Zhu & Li (2020b), which is called tensor power method. By applying tensor power method, we will prove that:
   - When $\widehat{\Lambda}_1^{(t)}$ reaches $\Theta(1/m)$ at $T_1$, $\bar{\Lambda}_1^{(t)}$ and $\Gamma^{(t)}$ remain a magnitude no more than initialization.
   - When $\widehat{\Lambda}_{-1}^{(t)}$ reaches $\Theta(1/m)$ at $T_{-1}$, $\bar{\Lambda}_{-1}$ and $\Gamma^{(t)}$ remain a magnitude no more than initialization.

### A.4.1 UPPER BOUND AND LOWER BOUND FOR $\widehat{\Lambda}_1^{(t)}, \widehat{\Lambda}_{-1}^{(t)}$ AND $\bar{\Lambda}_1^{(t)}, \bar{\Lambda}_{-1}^{(t)}$

We first consider Stage I of GD when $\max_{r\in\{\pm 1\}}\left\{\widehat{\Lambda}_r^{(t)}, \bar{\Lambda}_r^{(t)}\right\} \leq \Theta(m^{-1})$.

In this stage, we first prove following lemma:

**Lemma A.5.** As long as $\max_{r\in\{\pm 1\}}\left\{\widehat{\Lambda}_r^{(t)}, \bar{\Lambda}_r^{(t)}\right\} \leq \Theta(m^{-1})$, we have $c_i^{(t)} := -\ell'\left(\widehat{y}_i \cdot f_{\mathbf{W}^{(t)}}(\mathbf{x}_i)\right)$ and $b_i^{(t)} := -\ell'\left(y_i' \cdot f_{\mathbf{W}^{(t)}}(\mathbf{x}_i')\right)$ remains $1/2 \pm o(1)$.

*Proof of Lemma A.5.* Note that $\ell(z) = \log(1+\exp(-z))$ and $-\ell'(z) = \exp(-z)/\left(1+\exp(-z)\right)$, and without loss of generality assuming $\widehat{y}_i = y_i = 1$, we can express $c_i^{(t)}$ as follow:

$$c_i^{(t)} = -\ell'(f_{\mathbf{W}^{(t)}}(\mathbf{x}_i)) = \frac{e^{\sum_{j=m+1}^{2m}[\sigma(\langle\mathbf{w}_j^{(t)},\mathbf{v}\rangle)+\sigma(\langle\mathbf{w}_j^{(t)},\boldsymbol{\xi}_i\rangle)]}}{e^{\sum_{j=1}^m[\sigma(\langle\mathbf{w}_j^{(t)},\mathbf{v}\rangle)+\sigma(\langle\mathbf{w}_j^{(t)},\boldsymbol{\xi}_i\rangle)]} + e^{\sum_{j=m+1}^{2m}[\sigma(\langle\mathbf{w}_j^{(t)},\mathbf{v}\rangle)+\sigma(\langle\mathbf{w}_j^{(t)},\boldsymbol{\xi}_i\rangle)]}},$$

Since $\sigma(\langle\mathbf{w}_j^{(t)},\mathbf{v}\rangle)$ dominates $\sigma(\langle\mathbf{w}_j^{(t)},\boldsymbol{\xi}\rangle)$ for $j \in [m]$, which will be proved later by using *tensor power method*, we have

$$c_i^{(t)} = \frac{e^{\sum_{j=m+1}^{2m}[\sigma(\langle\mathbf{w}_j^{(t)},\mathbf{v}\rangle)+\sigma(\langle\mathbf{w}_j^{(t)},\boldsymbol{\xi}_i\rangle)]}}{e^{\sum_{j=1}^m \sigma(\langle\mathbf{w}_j^{(t)},\mathbf{v}\rangle)+\{\text{lower order term}\}} + e^{\sum_{j=m+1}^{2m}[\sigma(\langle\mathbf{w}_j^{(t)},\mathbf{v}\rangle)+\sigma(\langle\mathbf{w}_j^{(t)},\boldsymbol{\xi}_i\rangle)]}}.$$

On the one side,

$$c_i^{(t)} \geq \frac{1}{e^{\sum_{j=1}^m \sigma(\langle\mathbf{w}_j^{(t)},\mathbf{v}\rangle)+\{\text{lower order term}\}} + 1} \geq \frac{1}{e^{m(\widehat{\Lambda}_1^{(t)})^{q-1}} + 1} \geq \frac{1}{e^{\Theta(m^{-(q-1)})} + 1} = \frac{1}{2 + o(1)} = \frac{1}{2} - o(1).$$

On the other side, according to Lemma 5.3, we have $\bar{\Lambda}_1^{(t)} = \widetilde{O}(d^{-\frac{1}{4}})$ and $\Gamma^{(t)} = \widetilde{O}(d^{-\frac{1}{4}+\epsilon})$, it follows that

$$\begin{aligned}
c_i^{(t)} &\leq \frac{e^{m(\bar{\Lambda}_1^{(t)})^{q-1}+m(\Gamma^{(t)})^{q-1}}}{e^{\sum_{j=1}^m \sigma(\langle\mathbf{w}_j^{(t)},\mathbf{v}\rangle)+\{\text{lower order term}\}} + e^{m(\bar{\Lambda}_1^{(t)})^{q-1}+m(\Gamma^{(t)})^{q-1}}} \\
&= \frac{1+o(1)}{e^{\sum_{j=1}^m \sigma(\langle\mathbf{w}_j^{(t)},\mathbf{v}\rangle)+\{\text{lower order term}\}} + 1+o(1)} \\
&\leq \frac{1+o(1)}{1+1+o(1)} = \frac{1}{2}+o(1).
\end{aligned}$$

Therefore, we have $c_i^{(t)} = 1/2 \pm o(1)$ if $\widehat{y}_i = y_i = 1$ and other cases $(\widehat{y}_i = y_i = 1, \widehat{y}_i = -y_i, b_i^{(t)})$ can be proved in a similar way. $\quad\square$

By applying above lemma, we can obtain following lemma:

**Lemma A.6.** For any $\delta < 1/2$, with probability at least $1 - 2\delta$ over pseudo-labels generated by the pseudo-labeler, we have

$$\left|\frac{1}{n_{\mathrm{u}}}\sum_{i=1}^{n_{\mathrm{u}}}\widehat{y}_i y_i c_i^{(t)} - \left(p - \frac{1}{2}\right)\right| < \sqrt{\frac{1}{8n_{\mathrm{u}}}\log\frac{1}{\delta}} + o(1),$$

where $o(1)$ is with respect to $d$.

If we denote $\{(\mathbf{x}_i, y_i)|y_i = 1, i \in [n_{\mathrm{u}}]\}$ as $S_1$, $\{(\mathbf{x}_i, y_i)|y_i = -1, i \in [n_{\mathrm{u}}]\}$ as $S_{-1}$, $|S_1|$ as $n_1$ and $|S_{-1}|$ as $n_{-1}$, we have with probability at least $1 - 4\delta$ that

$$\left|\frac{1}{n_1}\sum_{i=1}^{n_1}\widehat{y}_i y_i c_i^{(t)} - \left(p - \frac{1}{2}\right)\right| < \sqrt{\frac{1}{8n_1}\log\frac{1}{\delta}} + o(1),$$

and

$$\left|\frac{1}{n_{-1}}\sum_{i=1}^{n_{-1}}\widehat{y}_i y_i c_i^{(t)} - \left(p - \frac{1}{2}\right)\right| < \sqrt{\frac{1}{8n_{-1}}\log\frac{1}{\delta}} + o(1).$$

*Proof of Lemma A.6.* First, according to Lemma A.5, we have

$$\frac{1}{n_{\mathrm{u}}}\sum_{i=1}^{n_{\mathrm{u}}}\widehat{y}_i y_i c_i^{(t)} = \frac{1}{n_{\mathrm{u}}}\sum_{i=1}^{n_{\mathrm{u}}}\widehat{y}_i y_i\left(c_i^{(t)}-\frac{1}{2}\right)+\frac{1}{2n_{\mathrm{u}}}\sum_{i=1}^{n_{\mathrm{u}}}\widehat{y}_i y_i = \frac{1}{2n_{\mathrm{u}}}\sum_{i=1}^{n_{\mathrm{u}}}\widehat{y}_i y_i \pm o(1) \tag{A.4}$$

Then, according to Hoeffding's inequality when $a_i = -1, b_i = 1$, we have

$$\mathbb{P}\left(\left|\frac{1}{n_{\mathrm{u}}}\sum_{i=1}^{n_{\mathrm{u}}}\widehat{y}_i y_i - \mathbb{E}\left[\frac{1}{n_{\mathrm{u}}}\sum_{i=1}^{n_{\mathrm{u}}}\widehat{y}_i y_i\right]\right| \geq t\right) \leq 2\exp\left(-\frac{2n_{\mathrm{u}}^2 t^2}{\sum_{i=1}^{n_{\mathrm{u}}}(a_i-b_i)^2}\right) = 2\exp\left(-2n_{\mathrm{u}}t^2\right).$$

Note that the pseudo-label $\widehat{y}_i$ generated by the pseudo-labeler takes $y_i$ with probability $p$ and $-y_i$ with probability $1-p$, we have $\mathbb{E}\left[\frac{1}{n_{\mathrm{u}}}\sum_{i=1}^{n_{\mathrm{u}}}\widehat{y}_i y_i\right] = \frac{1}{n_{\mathrm{u}}}\sum_{i=1}^{n_{\mathrm{u}}}\mathbb{E}\left[\widehat{y}_i y_i\right] = 2p - 1$. It follows that

$$\mathbb{P}\left(\left|\frac{1}{2n_{\mathrm{u}}}\sum_{i=1}^{n_{\mathrm{u}}}\widehat{y}_i y_i - \left(p-\frac{1}{2}\right)\right| \geq t\right) \leq 2\exp\left(-8n_{\mathrm{u}}t^2\right),$$

and therefore

$$\left|\frac{1}{2n_{\mathrm{u}}}\sum_{i=1}^{n_{\mathrm{u}}}\widehat{y}_i y_i - \left(p-\frac{1}{2}\right)\right| < \sqrt{\frac{1}{8n_{\mathrm{u}}}\log\frac{1}{\delta}} \tag{A.5}$$

holds with probability at least $1 - 2\delta$. According to (A.4) and (A.5), we have

$$\left|\frac{1}{2n_{\mathrm{u}}}\sum_{i=1}^{n_{\mathrm{u}}}\widehat{y}_i y_i - \left(p-\frac{1}{2}\right)\right| < \sqrt{\frac{1}{8n_{\mathrm{u}}}\log\frac{1}{\delta}} + o(1),$$

which verifies the first statement of the lemma. And the other part of the lemma can be proved in a similar way. □

According to above lemma and note that $n_{\mathrm{u}}, n_1, n_{-1} = \omega(1)$, we have further that

$$\left|\frac{1}{n_{\mathrm{u}}}\sum_{i=1}^{n_{\mathrm{u}}}\widehat{y}_i y_i c_i^{(t)} - \left(p-\frac{1}{2}\right)\right| = o(1), \quad \left|\frac{1}{n_r}\sum_{i=1}^{n_r}\widehat{y}_i y_i c_i^{(t)} - \left(p-\frac{1}{2}\right)\right| = o(1), r \in \{\pm1\}, \tag{A.6}$$

with high probability.

Besides, we also need an approximation about $n_1$ and $n_{-1}$, which is given as the following lemma:

**Lemma A.7.** For $r \in \{\pm1\}$, it holds with probability at least $1 - 2\delta$ that

$$\left|n_r - \frac{n_{\mathrm{u}}}{2}\right| < \sqrt{\frac{n_{\mathrm{u}}}{2}\log\frac{1}{\delta}},$$

where $n_r := |\{(\mathbf{x}_i, y_i)|y_i = r, i \in [n_{\mathrm{u}}]\}|$.

*Proof of Lemma A.7.* Note that $n_r = \sum_{i=1}^{n_{\mathrm{u}}}\mathbb{1}[X_i = r], r \in \{\pm1\}$ where $X_i$ takes label $+1$ or $-1$ with equal probability $1/2$, according to Hoeffding's inequality, we have

$$\mathbb{P}\left(\left|\sum_{i=1}^{n_{\mathrm{u}}}\mathbb{1}[X_i = r] - \mathbb{E}\left[\sum_{i=1}^{n_{\mathrm{u}}}\mathbb{1}[X_i = r]\right]\right| \geq t\right) \leq 2\exp\left(-\frac{2t^2}{n_{\mathrm{u}}}\right), r \in \{\pm1\},$$

and it follows that

$$\mathbb{P}\left(\left|n_r - \frac{n_{\mathrm{u}}}{2}\right| \geq t\right) \leq 2\exp\left(-\frac{2t^2}{n_{\mathrm{u}}}\right), r \in \{\pm1\},$$

leading to

$$\left|n_r - \frac{n_{\mathrm{u}}}{2}\right| < \sqrt{\frac{n_{\mathrm{u}}}{2}\log\frac{1}{\delta}},$$

with probability at least $1 - 2\delta$. □

For labeled dataset $S' = \{(\mathbf{x}_i', y_i')\}_{i=1}^{n_1}$, we also have

**Lemma A.8.** For $r \in \{\pm 1\}$, it holds with probability at least $1 - 2\delta$ that

$$\left| n'_r - \frac{n_l}{2} \right| < \sqrt{\frac{n_l}{2} \log \frac{1}{\delta}},$$

where $n'_r := |\{(\mathbf{x}'_i, y'_i) | y'_i = r, i \in [n_l]\}|$.

Then we are prepared to estimate a lower bound of increasing speed of $\widehat{\Lambda}^{(t)}$ and an upper bound of decreasing speed of $\bar{\Lambda}^{(t)}$ in the following lemma.

**Lemma A.9.** For $\widehat{\Lambda}_1^{(t)} := \max_{1 \le j \le m} \langle \mathbf{w}_j^{(t)}, \mathbf{v} \rangle$ and $\widehat{\Lambda}_{-1}^{(t)} := \max_{m+1 \le j \le 2m} \langle \mathbf{w}_j^{(t)}, -\mathbf{v} \rangle$, we have with high probability that

$$\widehat{\Lambda}_r^{(t+1)} \ge (1 - \eta\lambda) \cdot \widehat{\Lambda}_r^{(t)} + \eta \cdot \left( p - \frac{1}{2} \right) \cdot \Theta(d) \cdot (\widehat{\Lambda}_r^{(t)})^{q-1}, r \in \{\pm 1\}.$$

For $\bar{\Lambda}_1^{(t)} := \max_{m+1 \le j \le 2m} \langle \mathbf{w}_j^{(t)}, \mathbf{v} \rangle$ and $\bar{\Lambda}_1^{(t)} := \max_{1 \le j \le m} \langle \mathbf{w}_j^{(t)}, -\mathbf{v} \rangle$, we have with high probability that

$$\bar{\Lambda}_r^{(t+1)} \le (1 - \eta\lambda) \cdot \bar{\Lambda}_r^{(t)}, r \in \{\pm 1\}.$$

*Proof of Lemma A.9.* We first prove the former inequality. Let $j^* = \arg\max_{1 \le j \le m} \langle \mathbf{w}_j^{(t)}, \mathbf{v} \rangle$ and note that $u_{j^*} = \mathbb{1}_{[1 \le j \le m]} - \mathbb{1}_{[m+1 \le j \le 2m]} = 1$, then we have

$$\widehat{\Lambda}_1^{(t+1)} \ge \langle \mathbf{w}_{j^*}^{(t+1)}, \mathbf{v} \rangle$$

$$= (1 - \eta\lambda) \cdot \langle \mathbf{w}_{j^*}^{(t)}, \mathbf{v} \rangle + \frac{q\eta}{n_l + n_u} \Bigg( \underbrace{\sum_{i=1}^{n_u} y_i \widehat{y}_i c_i^{(t)} [\langle \mathbf{w}_{j^*}^{(t)}, y_i \cdot \mathbf{v} \rangle]_+^{q-1} \|\mathbf{v}\|_2^2}_{\clubsuit} + \underbrace{\sum_{i=1}^{n_l} b_i^{(t)} [\langle \mathbf{w}_{j^*}^{(t)}, y'_i \cdot \mathbf{v} \rangle]_+^{q-1} \|\mathbf{v}\|_2^2}_{\star} \Bigg)$$

Then we respectively estimate terms $\clubsuit$ and $\star$.

For $\clubsuit$, note the definition of $j^*$ that $\widehat{\Lambda}_1^{(t)} = \langle \mathbf{w}_{j^*}^{(t)}, \mathbf{v} \rangle$ and note the increasing property of $\widehat{\Lambda}_1^{(t)}$ and $\widehat{\Lambda}_1^{(0)} > 0$ with high probability, we have $\langle \mathbf{w}_{j^*}^{(t)}, \mathbf{v} \rangle > 0$. It follows that

$$\underbrace{\sum_{i=1}^{n_u} y_i \widehat{y}_i c_i^{(t)} [\langle \mathbf{w}_{j^*}^{(t)}, y_i \cdot \mathbf{v} \rangle]_+^{q-1} \|\mathbf{v}\|_2^2}_{\clubsuit} = \sum_{i \in S_1} y_i \widehat{y}_i c_i^{(t)} [\langle \mathbf{w}_{j^*}^{(t)}, \mathbf{v} \rangle]_+^{q-1} \|\mathbf{v}\|_2^2 + \sum_{i \in S_{-1}} y_i \widehat{y}_i c_i^{(t)} [-\langle \mathbf{w}_{j^*}^{(t)}, \mathbf{v} \rangle]_+^{q-1} \|\mathbf{v}\|_2^2$$

$$= \sum_{i \in S_1} y_i \widehat{y}_i c_i^{(t)} [\langle \mathbf{w}_{j^*}^{(t)}, \mathbf{v} \rangle]_+^{q-1} \|\mathbf{v}\|_2^2$$

$$= \left( \sum_{i \in S_1} y_i \widehat{y}_i c_i^{(t)} \right) \cdot \|\mathbf{v}\|_2^2 \cdot (\widehat{\Lambda}_1^{(t)})^{q-1}$$

$$= n_1 \cdot \left( p - \frac{1}{2} \pm o(1) \right) \cdot \|\mathbf{v}\|_2^2 \cdot (\widehat{\Lambda}_1^{(t)})^{q-1}, \tag{A.7}$$

where $S_1 := \{(\mathbf{x}_i, y_i) | y_i = 1, i \in [n_u]\}$, $S_{-1} := \{(\mathbf{x}_i, y_i) | y_i = -1, i \in [n_u]\}$, $n_1 = |S_1|$ and the last equality is due to (A.6).

For $\star$, similarly we have

$$\underbrace{\sum_{i=1}^{n_l} b_i^{(t)} [\langle \mathbf{w}_{j^*}^{(t)}, y'_i \cdot \mathbf{v} \rangle]_+^{q-1} \|\mathbf{v}\|_2^2}_{\star} = \sum_{i \in S'_1} b_i^{(t)} [\langle \mathbf{w}_{j^*}^{(t)}, \mathbf{v} \rangle]_+^{q-1} \|\mathbf{v}\|_2^2 + \sum_{i \in S'_{-1}} b_i^{(t)} [-\langle \mathbf{w}_{j^*}^{(t)}, \mathbf{v} \rangle]_+^{q-1} \|\mathbf{v}\|_2^2$$

$$= \sum_{i \in S'_1} b_i^{(t)} [\langle \mathbf{w}_{j^*}^{(t)}, \mathbf{v} \rangle]_+^{q-1} \|\mathbf{v}\|_2^2$$

$$= \left( \sum_{i \in S'_1} b_i^{(t)} \right) \cdot \|\mathbf{v}\|_2^2 \cdot (\widehat{\Lambda}_1^{(t)})^{q-1}$$

$$= n'_1 \cdot \left( \frac{1}{2} \pm o(1) \right) \cdot \|\mathbf{v}\|_2^2 \cdot (\widehat{\Lambda}_1^{(t)})^{q-1}, \tag{A.8}$$

where $S_1' = \{(\mathbf{x}_i', y_i') | y_i' = 1, i \in [n_l]\}$, $S_{-1}' = \{(\mathbf{x}_i', y_i') | y_i' = -1, i \in [n_l]\}$, $n_1' = |S_1'|$ and the last equality is due to Lemma A.5.

According to (A.7) and (A.8), we have

$\widehat{\Lambda}_1^{(t+1)}$

$$\geq (1 - \eta\lambda) \cdot \widehat{\Lambda}_1^{(t)} + \frac{q\eta}{n_l + n_u}\left(n_1 \cdot \left(p - \frac{1}{2} \pm o(1)\right) \cdot \|\mathbf{v}\|_2^2 \cdot \left(\widehat{\Lambda}_1^{(t)}\right)^{q-1} + n_1' \cdot \left(\frac{1}{2} \pm o(1)\right) \cdot \|\mathbf{v}\|_2^2 \cdot \left(\widehat{\Lambda}_1^{(t)}\right)^{q-1}\right)$$

$$= (1 - \eta\lambda) \cdot \widehat{\Lambda}_1^{(t)} + \frac{q\eta n_1}{n_l + n_u} \cdot \left(p - \frac{1}{2} \pm o(1)\right) \cdot \|\mathbf{v}\|_2^2 \cdot \left(\widehat{\Lambda}_1^{(t)}\right)^{q-1} + \frac{q\eta n_1'}{n_l + n_u} \cdot \left(\frac{1}{2} \pm o(1)\right) \cdot \|\mathbf{v}\|_2^2 \cdot \left(\widehat{\Lambda}_1^{(t)}\right)^{q-1}$$

$$= (1 - \eta\lambda) \cdot \widehat{\Lambda}_1^{(t)} + q\eta \cdot \left(\frac{n_1}{n_l + n_u} \cdot \left(p - \frac{1}{2} \pm o(1)\right) + \frac{n_1'}{n_l + n_u} \cdot \left(\frac{1}{2} \pm o(1)\right)\right) \cdot \|\mathbf{v}\|_2^2 \cdot \left(\widehat{\Lambda}_1^{(t)}\right)^{q-1}$$

$$= (1 - \eta\lambda) \cdot \widehat{\Lambda}_1^{(t)} + q\eta \cdot \left(\underbrace{\frac{n_1}{n_l + n_u} \cdot \left(p - \frac{1}{2}\right) + \frac{n_1'}{n_l + n_u} \cdot \frac{1}{2}}_{\spadesuit} \pm o(1)\right) \cdot \|\mathbf{v}\|_2^2 \cdot \left(\widehat{\Lambda}_1^{(t)}\right)^{q-1}. \quad \text{(A.9)}$$

According to Lemma A.7 and Lemma A.8, and note that $n_l = \widetilde{\Theta}(1)$, $n_u = \omega(d^{4\epsilon})$, we have for $\spadesuit$ that with probability at least $1 - 4\delta$

$$\left|\underbrace{\frac{n_1}{n_l + n_u} \cdot \left(p - \frac{1}{2}\right) + \frac{n_1'}{n_l + n_u} \cdot \frac{1}{2}}_{\spadesuit} - \frac{n_u}{2(n_l + n_u)} \cdot \left(p - \frac{1}{2}\right) - \frac{n_l}{2(n_l + n_u)} \cdot \frac{1}{2}\right|$$

$$\leq \frac{|n_1 - \frac{n_u}{2}|}{n_l + n_u} \cdot \left(p - \frac{1}{2}\right) + \frac{|n_1' - \frac{n_l}{2}|}{n_l + n_u} \cdot \frac{1}{2}$$

$$\leq \frac{\sqrt{\frac{n_u}{2} \log \frac{1}{\delta}}}{n_l + n_u} \cdot \left(p - \frac{1}{2}\right) + \frac{\sqrt{\frac{n_l}{2} \log \frac{1}{\delta}}}{n_l + n_u} \cdot \frac{1}{2}$$

$$= \Theta\left(\frac{1}{\sqrt{n_u}}\right)$$

$$= o(1)$$

Therefore, note that $n_u = \omega(n_l)$ and $n_u = \omega(1)$, we have

$$\underbrace{\frac{n_1}{n_l + n_u} \cdot \left(p - \frac{1}{2}\right) + \frac{n_1'}{n_l + n_u} \cdot \frac{1}{2}}_{\spadesuit} = \frac{n_u}{2(n_l + n_u)} \cdot \left(p - \frac{1}{2}\right) + \frac{n_l}{2(n_l + n_u)} \cdot \frac{1}{2} \pm o(1)$$

$$= \frac{1}{2} \cdot \left(p - \frac{1}{2}\right) \pm o(1) \quad \text{(A.10)}$$

Plugging (A.10) into (A.9), we have

$$\widehat{\Lambda}_1^{(t+1)} \geq (1 - \eta\lambda) \cdot \widehat{\Lambda}_1^{(t)} + q\eta \cdot \left(\frac{1}{2} \cdot \left(p - \frac{1}{2}\right) \pm o(1)\right) \cdot \|\mathbf{v}\|_2^2 \cdot \left(\widehat{\Lambda}_1^{(t)}\right)^{q-1}$$

$$= (1 - \eta\lambda) \cdot \widehat{\Lambda}_1^{(t)} + \eta \cdot \left(p - \frac{1}{2}\right) \cdot \Theta(d) \cdot \left(\widehat{\Lambda}_1^{(t)}\right)^{q-1}, \quad \text{(A.11)}$$

which verifies the first inequality of case $r = 1$ in the lemma.

Let $j^{**} = \operatorname{argmax}_{m+1 \leq j \leq 2m} \langle \mathbf{w}_j^{(t)}, -\mathbf{v} \rangle$ and note that $u_{j^{**}} = \mathbb{1}_{[1 \leq j \leq m]} - \mathbb{1}_{[m+1 \leq j \leq 2m]} = -1$, we have

$$\widehat{\Lambda}_{-1}^{(t+1)} \geq \langle \mathbf{w}_{j^*}^{(t+1)}, -\mathbf{v} \rangle$$

$$= (1 - \eta\lambda) \cdot \langle \mathbf{w}_{j^{**}}^{(t)}, -\mathbf{v} \rangle + \frac{q\eta}{n_l + n_u}\left(\underbrace{\sum_{i=1}^{n_u} y_i \widehat{y}_i c_i^{(t)} [\langle \mathbf{w}_{j^{**}}^{(t)}, y_i \cdot \mathbf{v} \rangle]_+^{q-1} \|\mathbf{v}\|_2^2}_{\clubsuit}\right.$$

$$\left. + \underbrace{\sum_{i=1}^{n_l} b_i^{(t)} [\langle \mathbf{w}_{j^{**}}^{(t)}, y_i' \cdot \mathbf{v} \rangle]_+^{q-1} \|\mathbf{v}\|_2^2}_{\star}\right)$$

For ♣, note the definition of $j^{**}$ that $\widehat{\Lambda}_{-1}^{(t)} = \langle \mathbf{w}_{j^{**}}^{(t)}, -\mathbf{v} \rangle$ and note the increasing property of $\widehat{\Lambda}_{-1}^{(t)}$ and $\widehat{\Lambda}_{-1}^{(0)} > 0$ with high probability, we have $\langle \mathbf{w}_{j^{**}}^{(t)}, -\mathbf{v} \rangle > 0$. According to (A.6), it follows that

$$\underbrace{\sum_{i=1}^{n_{\mathrm{u}}} y_i \widehat{y}_i c_i^{(t)} [\langle \mathbf{w}_{j^{**}}^{(t)}, y_i \cdot \mathbf{v} \rangle]_+^{q-1} \|\mathbf{v}\|_2^2}_{\clubsuit} = \sum_{i \in S_{-1}} y_i \widehat{y}_i c_i^{(t)} [\langle \mathbf{w}_{j^{**}}^{(t)}, -\mathbf{v} \rangle]_+^{q-1} \|\mathbf{v}\|_2^2$$

$$= n_{-1} \cdot \left( p - \frac{1}{2} \pm o(1) \right) \cdot \|\mathbf{v}\|_2^2 \cdot \left( \widehat{\Lambda}_{-1}^{(t)} \right)^{q-1}, \qquad (A.12)$$

where $S_{-1} := \{ (\mathbf{x}_i, y_i) | y_i = -1, i \in [n_{\mathrm{u}}] \}$, $n_{-1} = |S_{-1}|$.

For ★, according to Lemma A.5, similarly we have

$$\underbrace{\sum_{i=1}^{n_{\mathrm{l}}} b_i^{(t)} [\langle \mathbf{w}_{j^{**}}^{(t)}, y_i' \cdot \mathbf{v} \rangle]_+^{q-1} \|\mathbf{v}\|_2^2}_{\star} = \sum_{i \in S'_{-1}} b_i^{(t)} [\langle \mathbf{w}_{j^{**}}^{(t)}, -\mathbf{v} \rangle]_+^{q-1} \|\mathbf{v}\|_2^2 = n'_{-1} \cdot \left( \frac{1}{2} \pm o(1) \right) \cdot \|\mathbf{v}\|_2^2 \cdot \left( \widehat{\Lambda}_{-1}^{(t)} \right)^{q-1},$$

$$(A.13)$$

where $S'_{-1} = \{ (\mathbf{x}_i', y_i') | y_i' = -1, i \in [n_{\mathrm{l}}] \}$ and $n'_{-1} = |S'_{-1}|$.

According to (A.12) and (A.13), we have

$$\widehat{\Lambda}_{-1}^{(t+1)} \geq (1 - \eta\lambda) \cdot \widehat{\Lambda}_{-1}^{(t)} + q\eta \cdot \left( \underbrace{\frac{n_{-1}}{n_{\mathrm{l}} + n_{\mathrm{u}}} \cdot \left( p - \frac{1}{2} \right) + \frac{n'_{-1}}{n_{\mathrm{l}} + n_{\mathrm{u}}} \cdot \frac{1}{2}}_{\spadesuit} \pm o(1) \right) \cdot \|\mathbf{v}\|_2^2 \cdot \left( \widehat{\Lambda}_{-1}^{(t)} \right)^{q-1}.$$

$$(A.14)$$

According to Lemma A.7 and Lemma A.8, and note that $n_{\mathrm{l}} = \widetilde{\Theta}(1), n_{\mathrm{u}} = \omega(d^{4\epsilon})$, we have for ♠ that with probability at least $1 - 4\delta$

$$\left| \underbrace{\frac{n_{-1}}{n_{\mathrm{l}} + n_{\mathrm{u}}} \cdot \left( p - \frac{1}{2} \right) + \frac{n'_{-1}}{n_{\mathrm{l}} + n_{\mathrm{u}}} \cdot \frac{1}{2}}_{\spadesuit} - \frac{n_{\mathrm{u}}}{2(n_{\mathrm{l}} + n_{\mathrm{u}})} \cdot \left( p - \frac{1}{2} \right) - \frac{n_{\mathrm{l}}}{2(n_{\mathrm{l}} + n_{\mathrm{u}})} \cdot \frac{1}{2} \right|$$

$$\leq \frac{|n_{-1} - \frac{n_{\mathrm{u}}}{2}|}{n_{\mathrm{l}} + n_{\mathrm{u}}} \cdot \left( p - \frac{1}{2} \right) + \frac{|n'_{-1} - \frac{n_{\mathrm{l}}}{2}|}{n_{\mathrm{l}} + n_{\mathrm{u}}} \cdot \frac{1}{2}$$

$$\leq \frac{\sqrt{\frac{n_{\mathrm{u}}}{2} \log \frac{1}{\delta}}}{n_{\mathrm{l}} + n_{\mathrm{u}}} \cdot \left( p - \frac{1}{2} \right) + \frac{\sqrt{\frac{n_{\mathrm{l}}}{2} \log \frac{1}{\delta}}}{n_{\mathrm{l}} + n_{\mathrm{u}}} \cdot \frac{1}{2}$$

$$= \Theta\left( \frac{1}{\sqrt{n_{\mathrm{u}}}} \right)$$

$$= o(1).$$

Therefore, note that $n_{\mathrm{u}} = \omega(n_{\mathrm{l}})$ and $n_{\mathrm{u}} = \omega(1)$, we have

$$\underbrace{\frac{n_{-1}}{n_{\mathrm{l}} + n_{\mathrm{u}}} \cdot \left( p - \frac{1}{2} \right) + \frac{n'_{-1}}{n_{\mathrm{l}} + n_{\mathrm{u}}} \cdot \frac{1}{2}}_{\spadesuit} = \frac{n_{\mathrm{u}}}{2(n_{\mathrm{l}} + n_{\mathrm{u}})} \cdot \left( p - \frac{1}{2} \right) + \frac{n_{\mathrm{l}}}{2(n_{\mathrm{l}} + n_{\mathrm{u}})} \cdot \frac{1}{2} \pm o(1)$$

$$= \frac{1}{2} \cdot \left( p - \frac{1}{2} \right) \pm o(1) \qquad (A.15)$$

Plugging (A.15) into (A.14), we have

$$\widehat{\Lambda}_{-1}^{(t+1)} \geq (1 - \eta\lambda) \cdot \widehat{\Lambda}_{-1}^{(t)} + q\eta \cdot \left( \frac{1}{2} \cdot \left( p - \frac{1}{2} \right) \pm o(1) \right) \cdot \|\mathbf{v}\|_2^2 \cdot \left( \widehat{\Lambda}_{-1}^{(t)} \right)^{q-1}$$

$$= (1 - \eta\lambda) \cdot \widehat{\Lambda}_{-1}^{(t)} + \eta \cdot \left( p - \frac{1}{2} \right) \cdot \Theta(d) \cdot \left( \widehat{\Lambda}_{-1}^{(t)} \right)^{q-1}, \qquad (A.16)$$

which verifies the first inequality of case $r = -1$ in the lemma.

Next, we prove the latter part of the lemma. Let $j^\natural = \arg\max_{m+1 \leq j \leq 2m} \langle \mathbf{w}_j^{(t+1)}, \mathbf{v} \rangle$, then we have:

$$\bar{\Lambda}_1^{(t+1)} = \langle \mathbf{w}_{j^\natural}^{(t+1)}, \mathbf{v} \rangle$$

$$= (1 - \eta\lambda) \cdot \langle \mathbf{w}_{j^\natural}^{(t)}, \mathbf{v} \rangle - \frac{q\eta}{n_\mathrm{l} + n_\mathrm{u}} \bigg( \underbrace{\sum_{i=1}^{n_\mathrm{u}} y_i \widehat{y}_i c_i^{(t)} [\langle \mathbf{w}_{j^\natural}^{(t)}, y_i \cdot \mathbf{v} \rangle]_+^{q-1} \|\mathbf{v}\|_2^2}_{\clubsuit}$$

$$+ \underbrace{\sum_{i=1}^{n_\mathrm{l}} b_i^{(t)} [\langle \mathbf{w}_{j^\natural}^{(t)}, y_i' \cdot \mathbf{v} \rangle]_+^{q-1} \|\mathbf{v}\|_2^2}_{\bigstar} \bigg).$$

For $\clubsuit$, according to (A.6), we have

$$\underbrace{\sum_{i=1}^{n_\mathrm{u}} y_i \widehat{y}_i c_i^{(t)} [\langle \mathbf{w}_{j^\natural}^{(t)}, y_i \cdot \mathbf{v} \rangle]_+^{q-1} \|\mathbf{v}\|_2^2}_{\clubsuit}$$

$$= \sum_{i \in S_1} y_i \widehat{y}_i c_i^{(t)} [\langle \mathbf{w}_{j^\natural}^{(t)}, \mathbf{v} \rangle]_+^{q-1} \|\mathbf{v}\|_2^2 + \sum_{i \in S_{-1}} y_i \widehat{y}_i c_i^{(t)} [\langle \mathbf{w}_{j^\natural}^{(t)}, -\mathbf{v} \rangle]_+^{q-1} \|\mathbf{v}\|_2^2$$

$$= \bigg( \sum_{i \in S_1} y_i \widehat{y}_i c_i^{(t)} \bigg) \cdot [\langle \mathbf{w}_{j^\natural}^{(t)}, \mathbf{v} \rangle]_+^{q-1} \|\mathbf{v}\|_2^2 + \bigg( \sum_{i \in S_{-1}} y_i \widehat{y}_i c_i^{(t)} \bigg) \cdot [\langle \mathbf{w}_{j^\natural}^{(t)}, -\mathbf{v} \rangle]_+^{q-1} \|\mathbf{v}\|_2^2$$

$$= n_1 \cdot \bigg( p - \frac{1}{2} \pm o(1) \bigg) \cdot [\langle \mathbf{w}_{j^\natural}^{(t)}, \mathbf{v} \rangle]_+^{q-1} \|\mathbf{v}\|_2^2 + n_{-1} \cdot \bigg( p - \frac{1}{2} \pm o(1) \bigg) \cdot [\langle \mathbf{w}_{j^\natural}^{(t)}, -\mathbf{v} \rangle]_+^{q-1} \|\mathbf{v}\|_2^2 \geq 0,$$

and for $\bigstar$ it's obvious that

$$\underbrace{\sum_{i=1}^{n_\mathrm{l}} b_i^{(t)} [\langle \mathbf{w}_{j^\natural}^{(t)}, y_i' \cdot \mathbf{v} \rangle]_+^{q-1} \|\mathbf{v}\|_2^2 \geq 0.}_{\bigstar}$$

Therefore, it follows that

$$\bar{\Lambda}_1^{(t+1)} \leq (1 - \eta\lambda) \cdot \langle \mathbf{w}_{j^\natural}^{(t)}, \mathbf{v} \rangle \leq (1 - \eta\lambda) \bar{\Lambda}_1^{(t)}.$$

Let $j^{\natural\natural} = \arg\max_{1 \leq j \leq m} \langle \mathbf{w}_j^{(t+1)}, -\mathbf{v} \rangle$, then we have:

$$\bar{\Lambda}_{-1}^{(t+1)} = \langle \mathbf{w}_{j^{\natural\natural}}^{(t+1)}, -\mathbf{v} \rangle$$

$$= (1 - \eta\lambda) \cdot \langle \mathbf{w}_{j^{\natural\natural}}^{(t)}, -\mathbf{v} \rangle - \frac{q\eta}{n_\mathrm{l} + n_\mathrm{u}} \bigg( \sum_{i=1}^{n_\mathrm{u}} y_i \widehat{y}_i c_i^{(t)} [\langle \mathbf{w}_{j^{\natural\natural}}^{(t)}, y_i \cdot \mathbf{v} \rangle]_+^{q-1} \|\mathbf{v}\|_2^2$$

$$+ \sum_{i=1}^{n_\mathrm{l}} b_i^{(t)} [\langle \mathbf{w}_{j^{\natural\natural}}^{(t)}, y_i' \cdot \mathbf{v} \rangle]_+^{q-1} \|\mathbf{v}\|_2^2 \bigg)$$

$$\leq (1 - \eta\lambda) \cdot \langle \mathbf{w}_{j^{\natural\natural}}^{(t)}, -\mathbf{v} \rangle$$

$$\leq (1 - \eta\lambda) \cdot \bar{\Lambda}_{-1}^{(t)},$$

which verifies the second part of the lemma. $\qquad\square$

Although the accuracy of pseudo-labeler is larger than $1/2$, which is used as an assumption in the previous proof, we can also analyse the model with high label flipping probability and the accuracy of pseudo-labeler $p$ is smaller than $1/2$. In this case, the neural network for pre-training will turn to fit the opposite direction of feature vector, $\bar{\Lambda}_r^{(t)}$ will increase and $\widehat{\Lambda}_r^{(t)}$ will decrease, which is formulated as the following lemma.

**Lemma A.10.** For $\widehat{\Lambda}_1^{(t)} := \max_{1 \le j \le m} \langle \mathbf{w}_j^{(t)}, \mathbf{v} \rangle$ and $\widehat{\Lambda}_{-1}^{(t)} := \max_{m+1 \le j \le 2m} \langle \mathbf{w}_j^{(t)}, -\mathbf{v} \rangle$, we have with high probability that

$$\widehat{\Lambda}_r^{(t+1)} \le (1 - \eta\lambda) \cdot \widehat{\Lambda}_r^{(t)}, r \in \{\pm 1\}.$$

For $\bar{\Lambda}_1^{(t)} := \max_{m+1 \le j \le 2m} \langle \mathbf{w}_j^{(t)}, \mathbf{v} \rangle$ and $\bar{\Lambda}_1^{(t)} := \max_{1 \le j \le m} \langle \mathbf{w}_j^{(t)}, -\mathbf{v} \rangle$, we have with high probability that

$$\bar{\Lambda}_r^{(t+1)} \ge (1 - \eta\lambda) \cdot \bar{\Lambda}_r^{(t)} + \eta \cdot \left( \frac{1}{2} - p \right) \cdot \Theta(d) \cdot (\bar{\Lambda}_r^{(t)})^{q-1}, r \in \{\pm 1\}.$$

*Proof of Lemma A.10.* First, we prove the former part of this lemma. Let $j^* = \arg\max_{1 \le j \le m} \langle \mathbf{w}_j^{(t+1)}, \mathbf{v} \rangle$ and note that $u_{j^*} = \mathbb{1}_{[1 \le j \le m]} - \mathbb{1}_{[m+1 \le j \le 2m]} = 1$, then we have

$$\widehat{\Lambda}_1^{(t+1)} = \langle \mathbf{w}_{j^*}^{(t+1)}, \mathbf{v} \rangle$$

$$= (1 - \eta\lambda) \cdot \langle \mathbf{w}_{j^*}^{(t)}, \mathbf{v} \rangle + \frac{q\eta}{n_l + n_u} \bigg( \underbrace{\sum_{i=1}^{n_u} y_i \widehat{y}_i c_i^{(t)} [\langle \mathbf{w}_{j^*}^{(t)}, y_i \cdot \mathbf{v} \rangle]_+^{q-1} \|\mathbf{v}\|_2^2}_{\clubsuit}$$

$$+ \underbrace{\sum_{i=1}^{n_l} b_i^{(t)} [\langle \mathbf{w}_{j^*}^{(t)}, y_i' \cdot \mathbf{v} \rangle]_+^{q-1} \|\mathbf{v}\|_2^2}_{\bigstar} \bigg).$$

For $\clubsuit$, according to (A.6), we have

$$\underbrace{\sum_{i=1}^{n_u} y_i \widehat{y}_i c_i^{(t)} [\langle \mathbf{w}_{j^*}^{(t)}, y_i \cdot \mathbf{v} \rangle]_+^{q-1} \|\mathbf{v}\|_2^2}_{\clubsuit}$$

$$= \sum_{i \in S_1} y_i \widehat{y}_i c_i^{(t)} [\langle \mathbf{w}_{j^*}^{(t)}, \mathbf{v} \rangle]_+^{q-1} \|\mathbf{v}\|_2^2 + \sum_{i \in S_{-1}} y_i \widehat{y}_i c_i^{(t)} [\langle \mathbf{w}_{j^*}^{(t)}, -\mathbf{v} \rangle]_+^{q-1} \|\mathbf{v}\|_2^2$$

$$= \bigg( \sum_{i \in S_1} y_i \widehat{y}_i c_i^{(t)} \bigg) \cdot [\langle \mathbf{w}_{j^*}^{(t)}, \mathbf{v} \rangle]_+^{q-1} \|\mathbf{v}\|_2^2 + \bigg( \sum_{i \in S_{-1}} y_i \widehat{y}_i c_i^{(t)} \bigg) \cdot [\langle \mathbf{w}_{j^*}^{(t)}, -\mathbf{v} \rangle]_+^{q-1} \|\mathbf{v}\|_2^2$$

$$= n_1 \cdot \bigg( p - \frac{1}{2} \pm o(1) \bigg) \cdot [\langle \mathbf{w}_{j^*}^{(t)}, \mathbf{v} \rangle]_+^{q-1} \|\mathbf{v}\|_2^2 + n_{-1} \cdot \bigg( p - \frac{1}{2} \pm o(1) \bigg) \cdot [\langle \mathbf{w}_{j^*}^{(t)}, -\mathbf{v} \rangle]_+^{q-1} \|\mathbf{v}\|_2^2,$$

For $\bigstar$, according to (A.6), we have

$$\underbrace{\sum_{i=1}^{n_l} b_i^{(t)} [\langle \mathbf{w}_{j^*}^{(t)}, y_i' \cdot \mathbf{v} \rangle]_+^{q-1} \|\mathbf{v}\|_2^2}_{\bigstar}$$

$$= \sum_{i \in S_1'} b_i^{(t)} [\langle \mathbf{w}_{j^*}^{(t)}, \mathbf{v} \rangle]_+^{q-1} \|\mathbf{v}\|_2^2 + \sum_{i \in S_{-1}'} b_i^{(t)} [\langle \mathbf{w}_{j^*}^{(t)}, -\mathbf{v} \rangle]_+^{q-1} \|\mathbf{v}\|_2^2$$

$$= n_1' \cdot \bigg( \frac{1}{2} \pm o(1) \bigg) \cdot [\langle \mathbf{w}_{j^*}^{(t)}, \mathbf{v} \rangle]_+^{q-1} \|\mathbf{v}\|_2^2 + n_{-1}' \cdot \bigg( \frac{1}{2} \pm o(1) \bigg) \cdot [\langle \mathbf{w}_{j^*}^{(t)}, -\mathbf{v} \rangle]_+^{q-1} \|\mathbf{v}\|_2^2,$$

It follows that

$$\underbrace{\sum_{i=1}^{n_u} y_i \widehat{y}_i c_i^{(t)} [\langle \mathbf{w}_{j^*}^{(t)}, y_i \cdot \mathbf{v} \rangle]_+^{q-1} \|\mathbf{v}\|_2^2}_{\clubsuit} + \underbrace{\sum_{i=1}^{n_l} b_i^{(t)} [\langle \mathbf{w}_{j^*}^{(t)}, y_i' \cdot \mathbf{v} \rangle]_+^{q-1} \|\mathbf{v}\|_2^2}_{\bigstar}$$

$$= \bigg( n_1 \cdot \bigg( p - \frac{1}{2} \pm o(1) \bigg) + n_1' \cdot \bigg( \frac{1}{2} \pm o(1) \bigg) \bigg) \cdot [\langle \mathbf{w}_{j^*}^{(t)}, \mathbf{v} \rangle]_+^{q-1} \|\mathbf{v}\|_2^2$$

$$+ \bigg( n_{-1} \cdot \bigg( p - \frac{1}{2} \pm o(1) \bigg) + n_{-1}' \cdot \bigg( \frac{1}{2} \pm o(1) \bigg) \bigg) \cdot [\langle \mathbf{w}_{j^*}^{(t)}, \mathbf{v} \rangle]_+^{q-1} \|\mathbf{v}\|_2^2.$$

According to Lemma A.7 and note that $n_u = \omega(n_l)$, it holds with probability at least $1 - 8\delta$ that

$$n_1' \cdot \left(\frac{1}{2} \pm o(1)\right) \leq \left(\frac{n_l}{2} + \sqrt{\frac{n_l}{2}\log\frac{1}{\delta}}\right) \cdot \left(\frac{1}{2} \pm o(1)\right) = \Theta(n_l) = o(n_u)$$

$$\leq \left(\frac{n_u}{2} + \sqrt{\frac{n_u}{2}\log\frac{1}{\delta}}\right) \cdot \left(\frac{1}{2} - p \pm o(1)\right) \leq n_1 \cdot \left(\frac{1}{2} - p \pm o(1)\right),$$

$$n_{-1}' \cdot \left(\frac{1}{2} \pm o(1)\right) \leq \left(\frac{n_l}{2} + \sqrt{\frac{n_l}{2}\log\frac{1}{\delta}}\right) \cdot \left(\frac{1}{2} \pm o(1)\right) = \Theta(n_l) = o(n_u)$$

$$\leq \left(\frac{n_u}{2} + \sqrt{\frac{n_u}{2}\log\frac{1}{\delta}}\right) \cdot \left(\frac{1}{2} - p \pm o(1)\right) \leq n_{-1} \cdot \left(\frac{1}{2} - p \pm o(1)\right),$$

leading to $\clubsuit + \bigstar \leq 0$. Therefore,

$$\widehat{\Lambda}_1^{(t+1)} \leq (1 - \eta\lambda)\langle \mathbf{w}_{j^*}^{(t)}, \mathbf{v}\rangle \leq (1 - \eta\lambda) \cdot \widehat{\Lambda}_1^{(t)}.$$

And we can prove in a similar way that $\widehat{\Lambda}_{-1}^{(t+1)} \leq (1 - \eta\lambda) \cdot \widehat{\Lambda}_{-1}^{(t)}$.

Next, we prove the second part of the lemma. Let $j^\natural = \arg\max_{m+1 \leq j \leq 2m}\langle \mathbf{w}_j^{(t)}, \mathbf{v}\rangle$ and note that $u_{j^\natural} = \mathbb{1}_{[1 \leq j \leq m]} - \mathbb{1}_{[m+1 \leq j \leq 2m]} = -1$, then we have

$$\bar{\Lambda}_1^{(t+1)} \geq \langle \mathbf{w}_{j^\natural}^{(t+1)}, \mathbf{v}\rangle$$

$$= (1 - \eta\lambda) \cdot \langle \mathbf{w}_{j^\natural}^{(t)}, \mathbf{v}\rangle - \frac{q\eta}{n_l + n_u}\bigg(\underbrace{\sum_{i=1}^{n_u} y_i\widehat{y}_i c_i^{(t)}[\langle \mathbf{w}_{j^\natural}^{(t)}, y_i \cdot \mathbf{v}\rangle]_+^{q-1}\|\mathbf{v}\|_2^2}_{\clubsuit}$$

$$+ \underbrace{\sum_{i=1}^{n_l} b_i^{(t)}[\langle \mathbf{w}_{j^\natural}^{(t)}, y_i' \cdot \mathbf{v}\rangle]_+^{q-1}\|\mathbf{v}\|_2^2}_{\bigstar}\bigg).$$

For $\clubsuit$, note the definition of $j^\natural$ that $\bar{\Lambda}_1^{(t)} = \langle \mathbf{w}_{j^\natural}^{(t)}, \mathbf{v}\rangle$ and note the increasing property of $\bar{\Lambda}_1^{(t)}$ in this case and $\bar{\Lambda}_1^{(0)} > 0$ with high probability, we have $\langle \mathbf{w}_{j^\natural}^{(t)}, \mathbf{v}\rangle > 0$. It follows that

$$\underbrace{\sum_{i=1}^{n_u} y_i\widehat{y}_i c_i^{(t)}[\langle \mathbf{w}_{j^\natural}^{(t)}, y_i \cdot \mathbf{v}\rangle]_+^{q-1}\|\mathbf{v}\|_2^2}_{\clubsuit} = \sum_{i \in S_1} y_i\widehat{y}_i c_i^{(t)}[\langle \mathbf{w}_{j^\natural}^{(t)}, \mathbf{v}\rangle]_+^{q-1}\|\mathbf{v}\|_2^2 + \sum_{i \in S_{-1}} y_i\widehat{y}_i c_i^{(t)}[-\langle \mathbf{w}_{j^\natural}^{(t)}, \mathbf{v}\rangle]_+^{q-1}\|\mathbf{v}\|_2^2$$

$$= \sum_{i \in S_1} y_i\widehat{y}_i c_i^{(t)}[\langle \mathbf{w}_{j^\natural}^{(t)}, \mathbf{v}\rangle]_+^{q-1}\|\mathbf{v}\|_2^2$$

$$= \left(\sum_{i \in S_1} y_i\widehat{y}_i c_i^{(t)}\right) \cdot \|\mathbf{v}\|_2^2 \cdot (\bar{\Lambda}_1^{(t)})^{q-1}$$

$$= n_1 \cdot \left(p - \frac{1}{2} \pm o(1)\right) \cdot \|\mathbf{v}\|_2^2 \cdot (\bar{\Lambda}_1^{(t)})^{q-1}, \tag{A.17}$$

For $\bigstar$, similarly we have

$$\underbrace{\sum_{i=1}^{n_l} b_i^{(t)}[\langle \mathbf{w}_{j^\natural}^{(t)}, y_i' \cdot \mathbf{v}\rangle]_+^{q-1}\|\mathbf{v}\|_2^2}_{\bigstar} = \sum_{i \in S_1'} b_i^{(t)}[\langle \mathbf{w}_{j^\natural}^{(t)}, \mathbf{v}\rangle]_+^{q-1}\|\mathbf{v}\|_2^2 + \sum_{i \in S_{-1}'} b_i^{(t)}[-\langle \mathbf{w}_{j^\natural}^{(t)}, \mathbf{v}\rangle]_+^{q-1}\|\mathbf{v}\|_2^2$$

$$= \sum_{i \in S_1'} b_i^{(t)}[\langle \mathbf{w}_{j^\natural}^{(t)}, \mathbf{v}\rangle]_+^{q-1}\|\mathbf{v}\|_2^2$$

$$= \left(\sum_{i \in S_1'} b_i^{(t)}\right) \cdot \|\mathbf{v}\|_2^2 \cdot (\bar{\Lambda}_1^{(t)})^{q-1}$$

$$= n_1' \cdot \left(\frac{1}{2} \pm o(1)\right) \cdot \|\mathbf{v}\|_2^2 \cdot (\bar{\Lambda}_1^{(t)})^{q-1}. \tag{A.18}$$

According to Lemma A.7, (A.17) and (A.18), we have $n_1' = o(n_1)$ with high probability, therefore

$$\clubsuit + \bigstar = n_1 \cdot \left( p - \frac{1}{2} \pm o(1) \right) \cdot \|\mathbf{v}\|_2^2 \cdot (\bar{\Lambda}_1^{(t)})^{q-1},$$

leading to

$$\bar{\Lambda}_1^{(t+1)} \geq (1 - \eta\lambda) \cdot \langle \mathbf{w}_{j^\natural}^{(t)}, \mathbf{v} \rangle - \frac{q\eta n_1}{n_1 + n_{\mathrm{u}}} \cdot \left( p - \frac{1}{2} \pm o(1) \right) \cdot \|\mathbf{v}\|_2^2 \cdot (\bar{\Lambda}_1^{(t)})^{q-1}$$

$$= (1 - \eta\lambda) \cdot \bar{\Lambda}_1^{(t)} + \eta \cdot \left( \frac{1}{2} - p \right) \cdot \Theta(d) \cdot (\bar{\Lambda}_1^{(t)})^{q-1}.$$

And we can prove in a similar way that

$$\bar{\Lambda}_1^{(t+1)} \geq (1 - \eta\lambda) \cdot \bar{\Lambda}_1^{(t)} + \eta \cdot \left( \frac{1}{2} - p \right) \cdot \Theta(d) \cdot (\bar{\Lambda}_1^{(t)})^{q-1}.$$

$\square$

In this case ($p < 1/2$), given a small amount of labeled data, downstream task parameter $\mathbf{a}$ will learn the negative direction and the main theorems still hold.

### A.4.2 UNIFORM UPPER BOUND FOR $\Gamma^{(t)}$

The following lemma provides an upper bound for the increasing rate of $\Gamma^{(t)}$.

**Lemma A.11.** For $\Gamma_i^{(t)} := \max_{j \in [2m]} \langle \mathbf{w}_j, \boldsymbol{\xi}_i \rangle, i \in [n_{\mathrm{u}}]$, $\Gamma_i'^{(t)} := \max_{j \in [2m]} \langle \mathbf{w}_j, \boldsymbol{\xi}_i' \rangle, i \in [n_l]$, $\Gamma^{(t)} := \max\{\max_{i \in [n_{\mathrm{u}}]} \Gamma_i^{(t)}, \max_{i \in [n_l]} \Gamma_i'^{(t)}\}$, we have with high probability that

$$\Gamma_i^{(t+1)} \leq (1 - \eta\lambda) \cdot \Gamma_i^{(t)} + \eta \cdot \max \left\{ \widetilde{\Theta}(d^{\frac{1}{2}+2\epsilon}), \widetilde{\Theta}\left( \frac{d^{1+2\epsilon}}{n_{\mathrm{u}}} \right) \right\} \cdot (\Gamma^{(t)})^{q-1}, i \in [n_l],$$

$$\Gamma_i'^{(t+1)} \leq (1 - \eta\lambda) \cdot \Gamma_i'^{(t)} + \eta \cdot \max \left\{ \widetilde{\Theta}(d^{\frac{1}{2}+2\epsilon}), \widetilde{\Theta}\left( \frac{d^{1+2\epsilon}}{n_{\mathrm{u}}} \right) \right\} \cdot (\Gamma^{(t)})^{q-1}, i \in [n_l],$$

and

$$\Gamma^{(t+1)} \leq (1 - \eta\lambda) \cdot \Gamma^{(t)} + \eta \cdot \max \left\{ \widetilde{\Theta}(d^{\frac{1}{2}+2\epsilon}), \widetilde{\Theta}\left( \frac{d^{1+2\epsilon}}{n_{\mathrm{u}}} \right) \right\} \cdot (\Gamma^{(t)})^{q-1},$$

where $\epsilon < 1/8$.

*Proof of Lemma A.11.* We first prove the former inequality. Let $j^\star = \arg\max_{1 \leq j \leq 2m} \langle \mathbf{w}_j^{(t+1)}, \boldsymbol{\xi}_l \rangle$, where $l \in [n_{\mathrm{u}}]$ is fixed. According to Lemma A.2, we have
$\Gamma_l^{(t+1)} = \langle \mathbf{w}_{j^\star}^{(t+1)}, \boldsymbol{\xi}_l \rangle$

$$= (1 - \eta\lambda) \cdot \langle \mathbf{w}_{j^\star}^{(t)}, \boldsymbol{\xi}_l \rangle + \frac{q\eta u_{j^\star}}{n_1 + n_{\mathrm{u}}} \left( \sum_{i=1}^{n_{\mathrm{u}}} \widehat{y}_i c_i^{(t)} [\langle \mathbf{w}_{j^\star}^{(t)}, \boldsymbol{\xi}_i \rangle]_+^{q-1} \langle \boldsymbol{\xi}_i, \boldsymbol{\xi}_l \rangle + \sum_{i=1}^{n_1} y_i' b_i^{(t)} [\langle \mathbf{w}_{j^\star}^{(t)}, \boldsymbol{\xi}_i' \rangle]_+^{q-1} \langle \boldsymbol{\xi}_i', \boldsymbol{\xi}_l \rangle \right)$$

$$\leq (1 - \eta\lambda) \cdot \langle \mathbf{w}_{j^\star}^{(t)}, \boldsymbol{\xi}_l \rangle + \frac{q\eta}{n_1 + n_{\mathrm{u}}} \left( \underbrace{\sum_{i=1}^{n_{\mathrm{u}}} c_i^{(t)} [\langle \mathbf{w}_{j^\star}^{(t)}, \boldsymbol{\xi}_i \rangle]_+^{q-1} |\langle \boldsymbol{\xi}_i, \boldsymbol{\xi}_l \rangle|}_{\clubsuit} + \underbrace{\sum_{i=1}^{n_1} b_i^{(t)} [\langle \mathbf{w}_{j^\star}^{(t)}, \boldsymbol{\xi}_i' \rangle]_+^{q-1} |\langle \boldsymbol{\xi}_i', \boldsymbol{\xi}_l \rangle|}_{\bigstar} \right),$$

(A.19)

where the last inequality is due to triangle inequality.

For $\clubsuit$, note that $l \in [n_{\mathrm{u}}]$ and there exists an $i \in [n_{\mathrm{u}}]$ equivalent to $l$, it follows that

$$\underbrace{\sum_{i=1}^{n_{\mathrm{u}}} c_i^{(t)} [\langle \mathbf{w}_{j^\star}^{(t)}, \boldsymbol{\xi}_i \rangle]_+^{q-1} |\langle \boldsymbol{\xi}_i, \boldsymbol{\xi}_l \rangle|}_{\clubsuit}$$

$$= \sum_{i \in [n_{\mathrm{u}}], i \neq l} c_i^{(t)} [\langle \mathbf{w}_{j^\star}^{(t)}, \boldsymbol{\xi}_i \rangle]_+^{q-1} |\langle \boldsymbol{\xi}_i, \boldsymbol{\xi}_l \rangle| + c_l^{(t)} [\langle \mathbf{w}_{j^\star}^{(t)}, \boldsymbol{\xi}_l \rangle]_+^{q-1} \|\boldsymbol{\xi}_l\|_2^2$$

(A.20)

$$\leq (n_{\mathrm{u}} - 1) \cdot \left( \frac{1}{2} + o(1) \right) \cdot \widetilde{\Theta}(d^{\frac{1}{2}+2\epsilon}) \cdot (\Gamma^{(t)})^{q-1} + \left( \frac{1}{2} + o(1) \right) \cdot \widetilde{\Theta}(d^{1+2\epsilon}) \cdot (\Gamma^{(t)})^{q-1}$$

$$= (n_{\mathrm{u}} - 1) \cdot \widetilde{\Theta}(d^{\frac{1}{2}+2\epsilon}) \cdot (\Gamma^{(t)})^{q-1} + \widetilde{\Theta}(d^{1+2\epsilon}) \cdot (\Gamma^{(t)})^{q-1},$$

where the inequality is due to Lemma A.5, $\|\boldsymbol{\xi}_l\|_2^2 = \widetilde{\Theta}(d\sigma_p^2) = \widetilde{\Theta}(d^{1+2\epsilon})$, $|\langle \boldsymbol{\xi}_i, \boldsymbol{\xi}_l \rangle| = \widetilde{\Theta}(d^{\frac{1}{2}}\sigma_p^2) = \widetilde{\Theta}(d^{\frac{1}{2}+2\epsilon})$ according to Lemma C.3 and the definition of $\Gamma^{(t)}$.

For $\bigstar$, we have

$$\underbrace{\sum_{i=1}^{n_1} b_i^{(t)} [\langle \mathbf{w}_{j^\star}^{(t)}, \boldsymbol{\xi}_i' \rangle]_+^{q-1} |\langle \boldsymbol{\xi}_i', \boldsymbol{\xi}_l \rangle|}_{\bigstar} \leq n_1 \cdot \left( \frac{1}{2} + o(1) \right) \cdot \widetilde{\Theta}(d^{\frac{1}{2}+2\epsilon}) \cdot \left( \Gamma^{(t)} \right)^{q-1} = n_1 \cdot \widetilde{\Theta}(d^{\frac{1}{2}+2\epsilon}) \cdot \left( \Gamma^{(t)} \right)^{q-1},$$

$$\text{(A.21)}$$

Plugging (A.20) and (A.21) into (A.19), we have

$$\Gamma_l^{(t+1)} \leq (1 - \eta\lambda) \cdot \Gamma_l^{(t)} + \eta \cdot \left( \frac{q}{n_1 + n_u} \cdot \left( (n_u + n_1 - 1) \cdot \widetilde{\Theta}(d^{\frac{1}{2}+2\epsilon}) + \widetilde{\Theta}(d^{1+2\epsilon}) \right) \right) \cdot \left( \Gamma^{(t)} \right)^{q-1}$$

$$\leq (1 - \eta\lambda) \cdot \Gamma_l^{(t)} + \eta \cdot \max \left\{ \widetilde{\Theta}(d^{\frac{1}{2}+2\epsilon}), \widetilde{\Theta}\left( \frac{d^{1+2\epsilon}}{n_u} \right) \right\} \cdot \left( \Gamma^{(t)} \right)^{q-1},$$

which is the first part of this lemma.

Let $j^\star = \operatorname{argmax}_{1 \leq j \leq 2m} \langle \mathbf{w}_j^{(t+1)}, \boldsymbol{\xi}_l' \rangle$, where $l \in [n_1]$ is fixed. According to Lemma A.2, we have

$$\Gamma_l'^{(t+1)} = \langle \mathbf{w}_{j^\star}^{(t+1)}, \boldsymbol{\xi}_l' \rangle$$

$$= (1 - \eta\lambda) \cdot \langle \mathbf{w}_{j^\star}^{(t)}, \boldsymbol{\xi}_l' \rangle + \frac{q\eta u_{j^\star}}{n_1 + n_u} \left( \sum_{i=1}^{n_u} \widehat{y}_i c_i^{(t)} [\langle \mathbf{w}_{j^\star}^{(t)}, \boldsymbol{\xi}_i \rangle]_+^{q-1} \langle \boldsymbol{\xi}_i, \boldsymbol{\xi}_l' \rangle + \sum_{i=1}^{n_1} y_i' b_i^{(t)} [\langle \mathbf{w}_{j^\star}^{(t)}, \boldsymbol{\xi}_i' \rangle]_+^{q-1} \langle \boldsymbol{\xi}_i', \boldsymbol{\xi}_l' \rangle \right)$$

$$\leq (1 - \eta\lambda) \cdot \langle \mathbf{w}_{j^\star}^{(t)}, \boldsymbol{\xi}_l' \rangle + \frac{q\eta}{n_1 + n_u} \left( \underbrace{\sum_{i=1}^{n_u} c_i^{(t)} [\langle \mathbf{w}_{j^\star}^{(t)}, \boldsymbol{\xi}_i \rangle]_+^{q-1} |\langle \boldsymbol{\xi}_i, \boldsymbol{\xi}_l' \rangle|}_{\clubsuit} + \underbrace{\sum_{i=1}^{n_1} b_i^{(t)} [\langle \mathbf{w}_{j^\star}^{(t)}, \boldsymbol{\xi}_i' \rangle]_+^{q-1} |\langle \boldsymbol{\xi}_i', \boldsymbol{\xi}_l' \rangle|}_{\bigstar} \right),$$

$$\text{(A.22)}$$

For $\clubsuit$, we have

$$\underbrace{\sum_{i=1}^{n_u} c_i^{(t)} [\langle \mathbf{w}_{j^\star}^{(t)}, \boldsymbol{\xi}_i \rangle]_+^{q-1} |\langle \boldsymbol{\xi}_i, \boldsymbol{\xi}_l \rangle|}_{\clubsuit} \leq \sum_{i=1}^{n_u} \left( \frac{1}{2} \pm o(1) \right) \cdot \widetilde{\Theta}(d^{\frac{1}{2}+2\epsilon}) \cdot \left( \Gamma^{(t)} \right)^{q-1} = n_u \cdot \widetilde{\Theta}(d^{\frac{1}{2}+2\epsilon}) \cdot \left( \Gamma^{(t)} \right)^{q-1},$$

$$\text{(A.23)}$$

where the inequality is due to Lemma A.5, $|\langle \boldsymbol{\xi}_i, \boldsymbol{\xi}_l \rangle| = \widetilde{\Theta}(d^{\frac{1}{2}}\sigma_p^2) = \widetilde{\Theta}(d^{\frac{1}{2}+2\epsilon})$ and the definition of $\Gamma^{(t)}$.

For $\bigstar$, note that $l \in [n_1]$ and there exists an $i \in [n_1]$ equivalent to $l$, it follows that

$$\underbrace{\sum_{i=1}^{n_1} b_i^{(t)} [\langle \mathbf{w}_{j^\star}^{(t)}, \boldsymbol{\xi}_i' \rangle]_+^{q-1} |\langle \boldsymbol{\xi}_i', \boldsymbol{\xi}_l' \rangle|}_{\bigstar}$$

$$= \sum_{i \in [n_1], i \neq l} b_i^{(t)} [\langle \mathbf{w}_{j^\star}^{(t)}, \boldsymbol{\xi}_i' \rangle]_+^{q-1} |\langle \boldsymbol{\xi}_i', \boldsymbol{\xi}_l' \rangle| + b_l^{(t)} [\langle \mathbf{w}_{j^\star}^{(t)}, \boldsymbol{\xi}_l' \rangle]_+^{q-1} \|\boldsymbol{\xi}_l'\|_2^2 \quad \text{(A.24)}$$

$$\leq (n_1 - 1) \cdot \left( \frac{1}{2} + o(1) \right) \cdot \widetilde{\Theta}(d^{\frac{1}{2}+2\epsilon}) \cdot \left( \Gamma^{(t)} \right)^{q-1} + \left( \frac{1}{2} + o(1) \right) \cdot \widetilde{\Theta}(d^{1+2\epsilon}) \cdot \left( \Gamma^{(t)} \right)^{q-1}$$

$$= (n_1 - 1) \cdot \widetilde{\Theta}(d^{\frac{1}{2}+2\epsilon}) + \widetilde{\Theta}(d^{1+2\epsilon}) \cdot \left( \Gamma^{(t)} \right)^{q-1}$$

Plugging (A.23) and (A.24) into (A.22), we have

$$\Gamma_l'^{(t+1)} \leq (1 - \eta\lambda) \cdot \Gamma_l'^{(t+1)} + \eta \cdot \left( \frac{q}{n_1 + n_u} \cdot \left( (n_u + n_1 - 1) \cdot \widetilde{\Theta}(d^{\frac{1}{2}+2\epsilon}) + \widetilde{\Theta}(d^{1+2\epsilon}) \right) \right) \cdot \left( \Gamma^{(t)} \right)^{q-1}$$

$$\leq (1 - \eta\lambda) \cdot \Gamma_l'^{(t+1)} + \eta \cdot \max \left\{ \widetilde{\Theta}(d^{\frac{1}{2}+2\epsilon}), \widetilde{\Theta}\left( \frac{d^{1+2\epsilon}}{n_u} \right) \right\} \cdot \left( \Gamma^{(t)} \right)^{q-1},$$

which verifies the second inequality in this lemma.

Note that $\Gamma^{(t)} = \max\{\max_{l\in[n_\mathrm{u}]} \Gamma_l^{(t)}, \max_{l\in[n_\mathrm{l}]} \Gamma_l'^{(t)}\}$, without loss of generality, we assume $\Gamma^{(t)} = \max_{l\in[n_\mathrm{u}]} \Gamma_l^{(t)}$ and assume $l^* = \mathrm{argmax}_{l\in[n_\mathrm{u}]} \Gamma_l^{(t+1)}$, we have

$$\Gamma^{(t+1)} = \Gamma_{l^*}^{(t+1)} \leq (1 - \eta\lambda) \cdot \Gamma_{l^*}^{(t)} + \eta \cdot \max\left\{\widetilde{\Theta}(d^{\frac{1}{2}+2\epsilon}), \widetilde{\Theta}\left(\frac{d^{1+2\epsilon}}{n_\mathrm{u}}\right)\right\} \cdot (\Gamma^{(t)})^{q-1}$$

$$\leq (1 - \eta\lambda) \cdot \Gamma^{(t)} + \eta \cdot \max\left\{\widetilde{\Theta}(d^{\frac{1}{2}+2\epsilon}), \widetilde{\Theta}\left(\frac{d^{1+2\epsilon}}{n_\mathrm{u}}\right)\right\} \cdot (\Gamma^{(t)})^{q-1},$$

which verifies the third inequality in this lemma.

$\square$

### A.4.3 TENSOR POWER METHOD: PROVING $\Gamma^{(t)} = O(\Gamma^{(0)})$ DURING $[0, T_r]$ AND COMPUTING THE MAGNITUDE OF $T_r$

In this section, we first show that off-diagonal correlation ($\bar{\Lambda}_r^{(t)}$ for $p > 1/2$ and $\widehat{\Lambda}_r^{(t)}$ for $p < 1/2$) remains initialization magnitude during $[0, T_r]$. If the accuracy of pseudo-labeler $p > 1/2$, we have off-diagonal correlation $\bar{\Lambda}_r^{(t+1)} \leq (1 - \eta\lambda) \cdot \bar{\Lambda}_r^{(t)}$ for $r \in \{\pm 1\}$, therefore, $\bar{\Lambda}_r^{(t)} = O(\bar{\Lambda}_r^{(0)}) = \widetilde{O}(d^{-\frac{1}{4}})$. If $p < 1/2$, we have off-diagonal correlation $\widehat{\Lambda}_r^{(t+1)} \leq (1 - \eta\lambda) \cdot \widehat{\Lambda}_r^{(t)}$ for $r \in \{\pm 1\}$, therefore, $\widehat{\Lambda}_r^{(t)} = O(\widehat{\Lambda}_r^{(0)}) = \widetilde{O}(d^{-\frac{1}{4}})$. In this paper, we mainly focus on $p > 1/2$.

According to Sections A.4.1 and A.4.2, we have obtained following upper bounds and lower bounds for *feature learning* term $\widehat{\Lambda}_r^{(t)}, \bar{\Lambda}_r^{(t)}, r \in \{\pm 1\}$ and *noise memorization* term $\Gamma^{(t)}$: When $t \in [0, T_r]$, we have

$$\widehat{\Lambda}_r^{(t+1)} \geq \widehat{\Lambda}_r^{(t)} + \eta \cdot (2p - 1) \cdot \Theta(d) \cdot (\widehat{\Lambda}_r^{(t)})^{q-1} \text{ and } \bar{\Lambda}_r^{(t+1)} \leq (1 - \eta\lambda) \cdot \bar{\Lambda}_r^{(t)}, \text{ for } r \in \{\pm 1\};$$

$$\Gamma^{(t+1)} \leq (1 - \eta\lambda) \cdot \Gamma^{(t)} + \eta \cdot \max\left\{\widetilde{\Theta}(d^{\frac{1}{2}+2\epsilon}), \widetilde{\Theta}\left(\frac{d^{1+2\epsilon}}{n_\mathrm{u}}\right)\right\} \cdot (\Gamma^{(t)})^{q-1}.$$

$$(A.25)$$

According to Condition 4.1, assume $n_\mathrm{u} = \Omega(d^{4\epsilon})$ and note that $\epsilon < 1/8$, we have

$$\max\left\{\widetilde{\Theta}(d^{\frac{1}{2}+2\epsilon}), \widetilde{\Theta}\left(\frac{d^{1+2\epsilon}}{n_\mathrm{u}}\right)\right\} = \max\left\{\widetilde{\Theta}(d^{\frac{1}{2}+2\epsilon}), \widetilde{O}(d^{1-2\epsilon})\right\} = \widetilde{O}(d^{1-2\epsilon}),$$

leading to

$$\Gamma^{(t+1)} \leq (1 - \eta\lambda) \cdot \Gamma^{(t)} + \eta \cdot \widetilde{\Theta}(d^{1-2\epsilon}) \cdot (\Gamma^{(t)})^{q-1}.$$

By leveraging tensor power method introduced in Lemma C.4, we can prove following lemma about the magnitude of $\Gamma^{(t)}$:

**Lemma A.12.** $\Gamma^{(t)}$ remains initialization magnitude during $[0, \max_{r\in\{\pm 1\}}\{T_r\}]$.

*Proof of Lemma A.12.* Let $T_r^*$ be the first iteration $t$ in which $\widehat{\Lambda}_r^{(t)} \geq A$ for $r \in \{\pm 1\}$, let $T^*$ be the first iteration $t$ in which $\Gamma^{(t)} \geq A'$, then according to Lemma C.4, we know

$$\sum_{t\geq 0, x_t \leq A} \eta \leq \frac{\delta}{(1 - (1+\delta)^{-(q-2)})x_0 C_1} + \eta \cdot \frac{C_2}{C_1}(1+\delta)^{q-1}\left(1 + \frac{\log(A/x_0)}{\log(1+\delta)}\right),$$

$$\sum_{t\geq 0, x_t \leq A} \eta \geq \frac{\delta(1 - (x_0/A)^{q-2})}{(1+\delta)^{q-1}(1 - (1+\delta)^{-(q-2)})x_0 C_2} - \eta \cdot (1+\delta)^{-(q-1)}\left(1 + \frac{\log(A/x_0)}{\log(1+\delta)}\right).$$

And it follows that

$$\eta \cdot T_r^* \leq \frac{\delta}{(1 - (1+\delta)^{-(q-2)})\widehat{\Lambda}_r^{(0)} C_1} + \eta \cdot \frac{C_2}{C_1}(1+\delta)^{q-1}\left(1 + \frac{\log(A/\widehat{\Lambda}_r^{(0)})}{\log(1+\delta)}\right),$$

$$\eta \cdot T^* \geq \frac{\delta'(1 - (x_0/A')^{q-2})}{(1+\delta)^{q-1}(1 - (1+\delta)^{-(q-2)})\Gamma^{(0)} C_2'} - \eta \cdot (1+\delta')^{-(q-1)}\left(1 + \frac{\log(A'/\Gamma^{(0)})}{\log(1+\delta')}\right),$$

where $C_1, C_2 = (2p-1) \cdot \widetilde{\Theta}(d)$ and $C_1', C_2' = \widetilde{\Theta}(d^{1-2\epsilon})$ according to (A.25).

Taking $A = \Theta(1/m), A' = C \cdot \Gamma^{(t)}$ where $C$ is a large constant and $C = \Theta(1)$, $\delta = \delta' = \frac{1}{2}$ and note that $\widehat{\Lambda}_r^{(0)} = \widetilde{\Theta}(\sigma_0 d^{\frac{1}{2}}) = \widetilde{\Theta}(d^{-\frac{1}{4}}), \Gamma^{(0)} = \widetilde{\Theta}(\sigma_0 \sigma_p d^{\frac{1}{2}}) = \widetilde{\Theta}(d^{-\frac{1}{4}+\epsilon})$, we have

$$\eta \cdot T_r^* \leq \widetilde{\Theta}(d^{-\frac{3}{4}}) + \eta \cdot \widetilde{\Theta}(1) = \widetilde{\Theta}(d^{-\frac{3}{4}}), \tag{A.26}$$

and

$$\eta \cdot T^* \geq \widetilde{\Theta}(d^{-\frac{3}{4}+\epsilon}) - \eta \cdot \widetilde{\Theta}(1) = \widetilde{\Theta}(d^{-\frac{3}{4}+\epsilon}). \tag{A.27}$$

Therefore, combining (A.26) and (A.27), we have $\eta \cdot T^* \geq \widetilde{\Theta}(d^{-\frac{3}{4}+\epsilon}) > \widetilde{\Theta}(d^{-\frac{3}{4}}) \geq \eta \cdot T_r^*$, leading to $T^* > T_r^*$ for both $r \in \{-1. +1\}$. This indicates that when $\widehat{\Lambda}_1^{(t)}, \widehat{\Lambda}_{-1}^{(t)}$ reach $\Theta(1/m)$, $\Gamma^{(t)}$ remain the same magnitude as initialization. $\qquad\square$

By leveraging tensor power method, we can also estimate the length of Stage I, i.e. $T_1, T_{-1}$, by applying tensor power method. To use tensor power method, we need to upper-bound the increasing speed of $\widehat{\Lambda}_r^{(t)}$. We have the following lemma:

**Lemma A.13.** For $r \in \{\pm 1\}$, we have with high probability that

$$\widehat{\Lambda}_r^{(t+1)} \geq (1-\eta\lambda) \cdot \widehat{\Lambda}_r^{(t)} + \eta \cdot q\left(p - \frac{1}{2} - o(1)\right) \cdot \|\mathbf{v}\|_2^2 \cdot (\widehat{\Lambda}_r^{(t)})^{q-1},$$

$$\widehat{\Lambda}_r^{(t+1)} \leq (1-\eta\lambda) \cdot \widehat{\Lambda}_r^{(t)} + \eta \cdot q\left(p - \frac{1}{2} + o(1)\right) \cdot \|\mathbf{v}\|_2^2 \cdot (\widehat{\Lambda}_r^{(t)})^{q-1}.$$

*Proof of Lemma A.13.* Let $j^* = \arg\max_{1 \leq j \leq m}\langle \mathbf{w}_j^{(t+1)}, \mathbf{v}\rangle$ and note that $u_{j^*} = \mathbb{1}_{[1 \leq j \leq m]} = \mathbb{1}_{[m+1 \leq j \leq 2m]} = 1$, then we have

$$\widehat{\Lambda}_1^{(t+1)} = \langle \mathbf{w}_{j^*}^{(t+1)}, \mathbf{v}\rangle$$

$$= (1-\eta\lambda) \cdot \langle \mathbf{w}_{j^*}^{(t)}, \mathbf{v}\rangle + \frac{q\eta}{n_l + n_u}\left(\underbrace{\sum_{i \in S_1} y_i\widehat{y}_i c_i^{(t)}[\langle \mathbf{w}_{j^*}^{(t)}, \mathbf{v}\rangle]_+^{q-1}\|\mathbf{v}\|_2^2 + \sum_{i \in S_{-1}} y_i\widehat{y}_i c_i^{(t)}[-\langle \mathbf{w}_{j^*}^{(t)}, \mathbf{v}\rangle]_+^2\|\mathbf{v}\|_2^{q-1}}_{\clubsuit}\right)$$

$$+ \frac{q\eta}{n_l + n_u}\left(\underbrace{\sum_{i \in S_1'} b_i^{(t)}[\langle \mathbf{w}_{j^*}^{(t)}, \mathbf{v}\rangle]_+^{q-1}\|\mathbf{v}\|_2^2 + \sum_{i \in S_1'} b_i^{(t)}[-\langle \mathbf{w}_{j^*}^{(t)}, \mathbf{v}\rangle]_+^{q-1}\|\mathbf{v}\|_2^2}_{\star}\right). \tag{A.28}$$

For $\clubsuit$, according to Lemma A.6, we have

$$\underbrace{\sum_{i \in S_1} y_i\widehat{y}_i c_i^{(t)}[\langle \mathbf{w}_{j^*}^{(t)}, \mathbf{v}\rangle]_+^{q-1}\|\mathbf{v}\|_2^2 + \sum_{i \in S_{-1}} y_i\widehat{y}_i c_i^{(t)}[-\langle \mathbf{w}_{j^*}^{(t)}, \mathbf{v}\rangle]_+^{q-1}\|\mathbf{v}\|_2^2}_{\clubsuit}$$

$$= n_1 \cdot \left(p - \frac{1}{2} \pm o(1)\right) \cdot [\langle \mathbf{w}_{j^*}^{(t)}, \mathbf{v}\rangle]_+^{q-1}\|\mathbf{v}\|_2^2 + n_{-1} \cdot \left(p - \frac{1}{2} \pm o(1)\right) \cdot [-\langle \mathbf{w}_{j^*}^{(t)}, \mathbf{v}\rangle]_+^{q-1}\|\mathbf{v}\|_2^2$$

$$\leq n_1 \cdot \left(p - \frac{1}{2} \pm o(1)\right) \cdot \|\mathbf{v}\|_2^2 \cdot (\widehat{\Lambda}_1^{(t)})^{q-1} + n_{-1} \cdot \left(p - \frac{1}{2} \pm o(1)\right) \cdot \|\mathbf{v}\|_2^2 \cdot (\bar{\Lambda}_{-1}^{(t)})^{q-1}$$

$$= n_1 \cdot \left(p - \frac{1}{2} \pm o(1)\right) \cdot \|\mathbf{v}\|_2^2 \cdot (\widehat{\Lambda}_1^{(t)})^{q-1}, \tag{A.29}$$

where the last equality is due to $\widehat{\Lambda}_1^{(t)} = \omega(\bar{\Lambda}_{-1}^{(t)})$.

For ★, according to Lemma A.5, we have

$$\underbrace{\sum_{i \in S_1'} b_i^{(t)} [\langle \mathbf{w}_{j^*}^{(t)}, \mathbf{v} \rangle]_+^{q-1} \|\mathbf{v}\|_2^2 + \sum_{i \in S_1'} b_i^{(t)} [-\langle \mathbf{w}_{j^*}^{(t)}, \mathbf{v} \rangle]_+^{q-1} \|\mathbf{v}\|_2^2}_{\bigstar}$$

$$= n_1' \cdot \left( \frac{1}{2} \pm o(1) \right) \cdot [\langle \mathbf{w}_{j^*}^{(t)}, \mathbf{v} \rangle]_+^{q-1} \|\mathbf{v}\|_2^2 + n_{-1}' \cdot \left( \frac{1}{2} \pm o(1) \right) \cdot [-\langle \mathbf{w}_{j^*}^{(t)}, \mathbf{v} \rangle]_+^{q-1} \|\mathbf{v}\|_2^2 \quad \text{(A.30)}$$

$$\leq n_1' \cdot \left( \frac{1}{2} \pm o(1) \right) \cdot \|\mathbf{v}\|_2^2 \cdot \left( \widehat{\Lambda}_1^{(t)} \right)^{q-1} + n_{-1}' \cdot \left( \frac{1}{2} \pm o(1) \right) \cdot \|\mathbf{v}\|_2^2 \cdot \left( \bar{\Lambda}_{-1}^{(t)} \right)^{q-1}$$

$$= n_1' \cdot \left( \frac{1}{2} \pm o(1) \right) \cdot \|\mathbf{v}\|_2^2 \cdot \left( \widehat{\Lambda}_1^{(t)} \right)^{q-1},$$

where the last equality is due to $\widehat{\Lambda}_1^{(t)} = \omega(\bar{\Lambda}_{-1}^{(t)})$.

Plugging (A.29) and (A.30) into (A.28), we have

$$\widehat{\Lambda}_1^{(t+1)}$$

$$\leq (1 - \eta\lambda) \cdot \widehat{\Lambda}_1^{(t)} + \frac{q\eta}{n_l + n_u} \left( n_1 \cdot \left( p - \frac{1}{2} \pm o(1) \right) \cdot \|\mathbf{v}\|_2^2 \cdot \left( \widehat{\Lambda}_1^{(t)} \right)^{q-1} + n_1' \cdot \left( \frac{1}{2} \pm o(1) \right) \cdot \|\mathbf{v}\|_2^2 \cdot \left( \widehat{\Lambda}_1^{(t)} \right)^{q-1} \right)$$

$$= (1 - \eta\lambda) \cdot \widehat{\Lambda}_1^{(t)} + \frac{q\eta n_1}{n_l + n_u} \cdot \left( p - \frac{1}{2} \pm o(1) \right) \cdot \|\mathbf{v}\|_2^2 \cdot \left( \widehat{\Lambda}_1^{(t)} \right)^{q-1} + \frac{q\eta n_1'}{n_l + n_u} \cdot \left( \frac{1}{2} \pm o(1) \right) \cdot \|\mathbf{v}\|_2^2 \cdot \left( \widehat{\Lambda}_1^{(t)} \right)^{q-1}$$

$$= (1 - \eta\lambda) \cdot \widehat{\Lambda}_1^{(t)} + q\eta \cdot \left( \frac{n_1}{n_l + n_u} \cdot \left( p - \frac{1}{2} \pm o(1) \right) + \frac{n_1'}{n_l + n_u} \cdot \left( \frac{1}{2} \pm o(1) \right) \right) \cdot \|\mathbf{v}\|_2^2 \cdot \left( \widehat{\Lambda}_1^{(t)} \right)^{q-1}$$

$$= (1 - \eta\lambda) \cdot \widehat{\Lambda}_1^{(t)} + q\eta \cdot \left( \underbrace{\frac{n_1}{n_l + n_u} \cdot \left( p - \frac{1}{2} \right) + \frac{n_1'}{n_l + n_u} \cdot \frac{1}{2}}_{\spadesuit} \pm o(1) \right) \cdot \|\mathbf{v}\|_2^2 \cdot \left( \widehat{\Lambda}_1^{(t)} \right)^{q-1}.$$

$$\text{(A.31)}$$

Note that we have already proved in (A.9) that

$$\widehat{\Lambda}_1^{(t+1)} \leq (1 - \eta\lambda) \cdot \widehat{\Lambda}_1^{(t)} + q\eta \cdot \left( \underbrace{\frac{n_1}{n_l + n_u} \cdot \left( p - \frac{1}{2} \right) + \frac{n_1'}{n_l + n_u} \cdot \frac{1}{2}}_{\spadesuit} \pm o(1) \right) \cdot \|\mathbf{v}\|_2^2 \cdot \left( \widehat{\Lambda}_1^{(t)} \right)^{q-1}.$$

$$\text{(A.32)}$$

Note we have already prove in (A.10) that

$$\underbrace{\frac{n_1}{n_l + n_u} \cdot \left( p - \frac{1}{2} \right) + \frac{n_1'}{n_l + n_u} \cdot \frac{1}{2}}_{\spadesuit} = \frac{1}{2} \cdot \left( p - \frac{1}{2} \right) \pm o(1)$$

Therefore, we have

$$\widehat{\Lambda}_1^{(t+1)} \geq (1 - \eta\lambda) \cdot \widehat{\Lambda}_1^{(t)} + q\eta \cdot \left( p - \frac{1}{2} - o(1) \right) \cdot \|\mathbf{v}\|_2^2 \cdot \left( \widehat{\Lambda}_1^{(t)} \right)^{q-1},$$

$$\widehat{\Lambda}_1^{(t+1)} \leq (1 - \eta\lambda) \cdot \widehat{\Lambda}_1^{(t)} + q\eta \cdot \left( p - \frac{1}{2} + o(1) \right) \cdot \|\mathbf{v}\|_2^2 \cdot \left( \widehat{\Lambda}_1^{(t)} \right)^{q-1}.$$

In a similar way, we can prove that

$$\widehat{\Lambda}_{-1}^{(t+1)} \geq (1 - \eta\lambda) \cdot \widehat{\Lambda}_{-1}^{(t)} + q\eta \cdot \left( p - \frac{1}{2} - o(1) \right) \cdot \|\mathbf{v}\|_2^2 \cdot \left( \widehat{\Lambda}_{-1}^{(t)} \right)^{q-1},$$

$$\widehat{\Lambda}_{-1}^{(t+1)} \leq (1 - \eta\lambda) \cdot \widehat{\Lambda}_{-1}^{(t)} + q\eta \cdot \left( p - \frac{1}{2} + o(1) \right) \cdot \|\mathbf{v}\|_2^2 \cdot \left( \widehat{\Lambda}_{-1}^{(t)} \right)^{q-1},$$

which completes the proof of this lemma. $\qquad \square$

**Lemma A.14** (Length of pre-training). For $r \in \{\pm 1\}$, let $T_r$ be the first iteration that $\widehat{\Lambda}_r^{(t)}$ reaches $\Theta(1/m)$ respectively. Then $T_r = \widetilde{\Theta}(d^{-\frac{3}{4}})/\eta$ for all $r \in \{\pm 1\}$.

*Proof of Lemma A.14.* By leveraging tensor power method given in Lemma C.4,

$$\sum_{t\geq 0, x_t \leq A} \eta \leq \frac{\delta}{(1-(1+\delta)^{-(q-2)})x_0 C_1} + \eta \cdot \frac{C_2}{C_1}(1+\delta)^{q-1}\left(1+\frac{\log(A/x_0)}{\log(1+\delta)}\right),$$

$$\sum_{t\geq 0, x_t \leq A} \eta \geq \frac{\delta(1-(x_0/A)^{q-2})}{(1+\delta)^{q-1}(1-(1+\delta)^{-(q-2)})x_0 C_2} - \eta \cdot (1+\delta)^{-(q-1)}\left(1+\frac{\log(A/x_0)}{\log(1+\delta)}\right),$$

we have for $r \in \{\pm 1\}$ that

$$\eta \cdot T_r^* = \sum_{t\geq 0, \widehat{\Lambda}_r^{(t)} \leq A} \eta \leq \underbrace{\frac{\delta}{(1-(1+\delta)^{-(q-2)})\widehat{\Lambda}_r^{(0)}C_1}}_{(i)} + \underbrace{\eta \cdot \frac{C_2}{C_1}(1+\delta)^{q-1}\left(1+\frac{\log(A/\widehat{\Lambda}_r^{(0)})}{\log(1+\delta)}\right)}_{(ii)},$$

$$\eta \cdot T_r^* = \sum_{t\geq 0, \widehat{\Lambda}_r^{(t)} \leq A} \eta \geq \underbrace{\frac{\delta(1-(x_0/A)^{q-2})}{(1+\delta)^{q-1}(1-(1+\delta)^{-(q-2)})\widehat{\Lambda}_r^{(0)}C_2}}_{(iii)} - \underbrace{\eta \cdot (1+\delta)^{-(q-1)}\left(1+\frac{\log(A/\widehat{\Lambda}_r^{(0)})}{\log(1+\delta)}\right)}_{(iv)},$$

where $C_1$ is taken as $q(p-\frac{1}{2}-o(1)) \cdot \|\mathbf{v}\|_2^2$ and $C_2$ is taken as $q(p-\frac{1}{2}+o(1)) \cdot \|\mathbf{v}\|_2^2$ according to Lemma A.13. Taking $\delta = \frac{1}{k}$, $A = \Theta(1/m)$ and note that terms $(ii), (iv)$ are respectively dominated by terms $(i), (iii)$ when $\eta$ is sufficiently small and letting $k \to \infty$, we have

$$\frac{1}{\widehat{\Lambda}_r^{(0)}C_2} - \{\text{lower order terms}\} \leq \eta \cdot T_r^* \leq \frac{1}{\widehat{\Lambda}_r^{(0)}C_1} + \{\text{lower order terms}\},$$

for $r \in \{\pm 1\}$. And it follows that

$$\eta \cdot T_r^* = \frac{1}{q(p-\frac{1}{2})\|\mathbf{v}\|_2^2 \cdot \widehat{\Lambda}_r^{(0)}} \pm \{\text{lower order terms}\}. \tag{A.33}$$

And by Lemma A.3, we have $\eta \cdot T_r^* = \Theta(1/q(p-\frac{1}{2})\|\mathbf{v}\|_2^2 \cdot \sqrt{\log(m)}\sigma_0\|\mathbf{v}\|_2) = \widetilde{\Theta}(d^{-3/4})$, which completes the proof. $\square$

The discussion in this section verifies Lemma 5.3 and provides a clear understanding about how $\widehat{\Lambda}_r^{(t)}, \bar{\Lambda}_r^{(t)}$ varies within the iteration range $[0, T_r]$ for $r \in \{\pm 1\}$. Note that the iteration numbers when $\widehat{\Lambda}_1^{(t)}$ and $\widehat{\Lambda}_{-1}^{(t)}$ reaches $\Theta(1/m)$ ($T_1$ and $T_{-1}$) are different, however, since $T_{-1}$ and $T_1$ have the same magnitude, it remains clear that although $T_1 \neq T_{-1}$ (wlog, assume $T_1 < T_{-1}$), we still have $\widehat{\Lambda}_1^{(t)} = \widetilde{\Theta}(1)$ and $\bar{\Lambda}_1^{(t)} = \widetilde{O}(d^{-\frac{1}{4}})$ within the iteration range $[T_1, T_{-1}]$, since off-diagonal feature learning also costs time no less than order $\Theta(1/\eta\sigma_0\|\mathbf{v}\|_2^3\sqrt{\log m})$, which is higher order than $|T_1 - T_{-1}| = \Theta(1/\eta\sigma_0\|\mathbf{v}\|_2^3 \log m)$, according to (A.33) and Lemma A.3. Therefore, at time $T_0 := \max\{T_1, T_{-1}\}$, off-diagonal $\bar{\Lambda}_1^{(t)}, \bar{\Lambda}_{-1}^{(t)}$ still remain initialization magnitude $\widetilde{O}(d^{-\frac{1}{4}})$, $\Gamma_1^{(t)}, \Gamma_{-1}^{(t)}$ remain initialization magnitude $\widetilde{O}(d^{-\frac{1}{4}+\epsilon})$, while on-diagonal $\widehat{\Lambda}_1^{(t)}, \widehat{\Lambda}_{-1}^{(t)}$ reach and then remain $\widetilde{\Theta}(1)$.

## A.5 PROOF OF LEMMA 5.2

If we only use labeled data $S'$ for the optimization of CNN, according to Lemma B.1, we have

$$\mathbf{w}_j^{(t+1)} = \mathbf{w}_j^{(t)} - \nabla_{\mathbf{w}_j}L_{S'}(\mathbf{W})$$

$$= (1-\eta\lambda) \cdot \mathbf{w}_j^{(t)} + \frac{q\eta u_j}{n_1}\sum_{i=1}^{n_1} b_i^{(t)}y_i'\big([\langle\mathbf{w}_j, y_i' \cdot \mathbf{v}\rangle]_+^{q-1} \cdot y_i' \cdot \mathbf{v} + [\langle\mathbf{w}_j, \boldsymbol{\xi}_i'\rangle]_+^{q-1} \cdot \boldsymbol{\xi}_i'\big),$$

where $u_j := \mathbb{1}_{[1\leq j\leq m]} - \mathbb{1}_{[m+1\leq 2m]}$, $b_i^{(t)} = -\ell'(y_i' \cdot f_\mathbf{W}(\mathbf{x}_i')) = \exp[-y_i' \cdot f_\mathbf{W}(\mathbf{x}_i')]/(1+\exp[-y_i' \cdot f_\mathbf{W}(\mathbf{x}_i')])$.

Notice that $\mathbf{v}$ and $\boldsymbol{\xi}'_i$ are orthogonal to each other, we have

$$\langle \mathbf{w}_j^{(t+1)}, \mathbf{v} \rangle = (1 - \eta\lambda) \cdot \langle \mathbf{w}_j^{(t)}, \mathbf{v} \rangle + \frac{q\eta u_j}{n_1} \sum_{i=1}^{n_1} b_i^{(t)} \cdot [\langle \mathbf{w}_j^{(t)}, y_i' \cdot \mathbf{v} \rangle]_+^{q-1} \cdot \|\mathbf{v}\|_2^2,$$

$$\langle \mathbf{w}_j^{(t+1)}, \boldsymbol{\xi}'_l \rangle = (1 - \eta\lambda) \cdot \langle \mathbf{w}_j^{(t)}, \boldsymbol{\xi}_i \rangle + \frac{q\eta u_j}{n_1} \sum_{i=1}^{n_1} b_i^{(t)} y_i' \cdot [\langle \mathbf{w}_j^{(t)}, \boldsymbol{\xi}'_i \rangle]_+^{q-1} \cdot \langle \boldsymbol{\xi}'_i, \boldsymbol{\xi}'_l \rangle, i \in [n_1].$$

Let $T_i'$ be the first iteration that $\Gamma_i'^{(t)}$ reaches $\Theta(1/m)$, then we have following lemma:

**Lemma A.15.** As long as $\Gamma_i'^{(t)} \leq \Theta(1/m)$, $b_i^{(t)} := -\ell'(y_i' \cdot f_{\mathbf{W}^{(t)}}(\mathbf{x}_i'))$ will remain $1/2 \pm o(1)$.

*Proof of Lemma A.15.* Note that $\ell(z) = \log(1+\exp(-z))$ and $-\ell'(z) = \exp(-z)/(1+\exp(-z))$, and without loss of generality assuming $y_i' = 1$, we can express $b_i^{(t)}$ as follow:

$$b_i^{(t)} = -\ell'(f_{\mathbf{W}^{(t)}}(\mathbf{x}_i')) = \frac{e^{\sum_{j=m+1}^{2m}[\sigma(\langle \mathbf{w}_j^{(t)}, \mathbf{v} \rangle) + \sigma(\langle \mathbf{w}_j^{(t)}, \boldsymbol{\xi}'_i \rangle)]}}{e^{\sum_{j=1}^{m}[\sigma(\langle \mathbf{w}_j^{(t)}, \mathbf{v} \rangle) + \sigma(\langle \mathbf{w}_j^{(t)}, \boldsymbol{\xi}'_i \rangle)]} + e^{\sum_{j=m+1}^{2m}[\sigma(\langle \mathbf{w}_j^{(t)}, \mathbf{v} \rangle) + \sigma(\langle \mathbf{w}_j^{(t)}, \boldsymbol{\xi}'_i \rangle)]}},$$

Since $\sigma(\langle \mathbf{w}_j^{(t)}, \boldsymbol{\xi} \rangle)$ will dominate $\sigma(\langle \mathbf{w}_j^{(t)}, \mathbf{v} \rangle)$ , which will be proved later by using *tensor power method*, we have

$$b_i^{(t)} = -\ell'(f_{\mathbf{W}^{(t)}}(\mathbf{x}_i')) = \frac{e^{\sum_{j=m+1}^{2m}[\sigma(\langle \mathbf{w}_j^{(t)}, \mathbf{v} \rangle) + \sigma(\langle \mathbf{w}_j^{(t)}, \boldsymbol{\xi}'_i \rangle)]}}{e^{\sum_{j=1}^{m} \sigma(\langle \mathbf{w}_j^{(t)}, \boldsymbol{\xi}'_i \rangle) \{+\text{lower order term}\}} + e^{\sum_{j=m+1}^{2m}[\sigma(\langle \mathbf{w}_j^{(t)}, \mathbf{v} \rangle) + \sigma(\langle \mathbf{w}_j^{(t)}, \boldsymbol{\xi}'_i \rangle)]}},$$

On the one side,

$$b_i^{(t)} \geq \frac{1}{e^{\sum_{j=1}^{m} \sigma(\langle \mathbf{w}_j^{(t)}, \boldsymbol{\xi}'_i \rangle) \{+\text{lower order term}\}} + 1}$$

$$\geq \frac{1}{e^{m(\Gamma_i'^{(t)})^q \{+\text{lower order term}\}} + 1}$$

$$\geq \frac{1}{e^{\Theta(m^{-(q-1)})} + 1} = \frac{1}{2 + o(1)} = \frac{1}{2} - o(1).$$

On the other side, according to Lemma 5.4, we have $\bar{\Lambda}_1^{(t)} = \widetilde{O}(d^{-\frac{1}{4}})$, it follows that

$$b_i^{(t)} \leq \frac{e^{m(\bar{\Lambda}_1^{(t)})^q + o(1)}}{e^{\sum_{j=1}^{m} \sigma(\langle \mathbf{w}_j^{(t)}, \boldsymbol{\xi}'_i \rangle) + \{\text{lower order term}\}} + e^{m(\bar{\Lambda}_1^{(t)})^q + o(1)}}$$

$$= \frac{1 + o(1)}{e^{\sum_{j=1}^{m} \sigma(\langle \mathbf{w}_j^{(t)}, \boldsymbol{\xi}'_i \rangle) + \{\text{lower order term}\}} + 1 + o(1)}$$

$$\leq \frac{1 + o(1)}{1 + 1 + o(1)} = \frac{1}{2} + o(1).$$

Therefore, we have $b_i^{(t)} = 1/2 \pm o(1)$ and the other case of $y_i = -1$ can be proved in a similar way. $\quad\square$

With the help of above lemma, we are now ready to prove Lemma 5.2.

*Proof of Lemma 5.2.* Let $j^* = \arg\max_{1 \le j \le m}\langle \mathbf{w}_j^{(t+1)}, \mathbf{v}\rangle$ and note that $u_j = 1$, according to Lemma A.15, we have

$$
\begin{aligned}
\widehat{\Lambda}_1^{(t+1)} &= \langle \mathbf{w}_{j^*}^{(t+1)}, \mathbf{v}\rangle \\
&= (1-\eta\lambda) \cdot \langle \mathbf{w}_{j^*}^{(t)}, \mathbf{v}\rangle + \frac{q\eta}{n_1}\sum_{i=1}^{n_1} b_i^{(t)}[\langle \mathbf{w}_{j^*}^{(t)}, y_i' \cdot \mathbf{v}\rangle]_+^{q-1}\|\mathbf{v}\|_2^2 \\
&= (1-\eta\lambda) \cdot \langle \mathbf{w}_{j^*}^{(t)}, \mathbf{v}\rangle + \frac{q\eta}{n_1}\sum_{i \in S_1'} b_i^{(t)}[\langle \mathbf{w}_{j^*}^{(t)}, \mathbf{v}\rangle]_+^{q-1}\|\mathbf{v}\|_2^2 + \frac{q\eta}{n_1}\sum_{i \in S_{-1}'} b_i^{(t)}[\langle \mathbf{w}_{j^*}^{(t)}, -\mathbf{v}\rangle]_+^{q-1}\|\mathbf{v}\|_2^2 \\
&= (1-\eta\lambda) \cdot \langle \mathbf{w}_{j^*}^{(t)}, \mathbf{v}\rangle + \frac{q\eta}{n_1}\underbrace{\sum_{i \in S_1'} \left(\frac{1}{2} \pm o(1)\right)[\langle \mathbf{w}_{j^*}^{(t)}, \mathbf{v}\rangle]_+^{q-1}\|\mathbf{v}\|_2^2}_{\clubsuit} \\
&\quad + \frac{q\eta}{n_1}\underbrace{\sum_{i \in S_{-1}'} \left(\frac{1}{2} \pm o(1)\right)[\langle \mathbf{w}_{j^*}^{(t)}, -\mathbf{v}\rangle]_+^{q-1}\|\mathbf{v}\|_2^2}_{\star}
\end{aligned}
$$

$$\tag{A.34}$$

For $\clubsuit$, we have

$$
\underbrace{\sum_{i \in S_1'} \left(\frac{1}{2} \pm o(1)\right)[\langle \mathbf{w}_{j^*}^{(t)}, \mathbf{v}\rangle]_+^{q-1}\|\mathbf{v}\|_2^2}_{\clubsuit} = n_1' \cdot \left(\frac{1}{2} \pm o(1)\right) \cdot [\langle \mathbf{w}_{j^*}^{(t)}, \mathbf{v}\rangle]_+^{q-1}\|\mathbf{v}\|_2^2
$$

$$\tag{A.35}$$

$$
\le n_1' \cdot \left(\frac{1}{2} \pm o(1)\right) \cdot \|\mathbf{v}\|_2^2 \cdot (\widehat{\Lambda}_1^{(t)})^{q-1}.
$$

For $\star$, we have

$$
\underbrace{\sum_{i \in S_{-1}'} \left(\frac{1}{2} \pm o(1)\right)[\langle \mathbf{w}_{j^*}^{(t)}, -\mathbf{v}\rangle]_+^{q-1}\|\mathbf{v}\|_2^2}_{\star} = n_{-1}' \cdot \left(\frac{1}{2} \pm o(1)\right) \cdot [\langle \mathbf{w}_{j^*}^{(t)}, -\mathbf{v}\rangle]_+^{q-1}\|\mathbf{v}\|_2^2
$$

$$\tag{A.36}$$

$$
\le n_{-1}' \cdot \left(\frac{1}{2} \pm o(1)\right) \cdot \|\mathbf{v}\|_2^2 \cdot (\bar{\Lambda}_{-1}^{(t)})^{q-1}.
$$

By plugging (A.35) and (A.36) in (A.34), and according to Lemma A.8, we have with probability at least $1 - 4\delta$ that

$$
\begin{aligned}
\widehat{\Lambda}_1^{(t+1)} &\le (1-\eta\lambda) \cdot \widehat{\Lambda}_1^{(t)} + \frac{q\eta}{n_1}\left(n_1' \cdot \left(\frac{1}{2} \pm o(1)\right) \cdot \|\mathbf{v}\|_2^2 \cdot (\widehat{\Lambda}_1^{(t)})^{q-1} + n_{-1}' \cdot \left(\frac{1}{2} \pm o(1)\right) \cdot \|\mathbf{v}\|_2^2 \cdot (\bar{\Lambda}_{-1}^{(t)})^{q-1}\right) \\
&\le (1-\eta\lambda) \cdot \widehat{\Lambda}_1^{(t)} + \frac{q\eta}{n_1}\left(\left(\frac{n_1}{2} + \sqrt{\frac{n_1}{2}\log\frac{1}{\delta}}\right) \cdot \left(\frac{1}{2} \pm o(1)\right) \cdot \|\mathbf{v}\|_2^2 \cdot (\widehat{\Lambda}_1^{(t)})^{q-1}\right. \\
&\qquad\qquad\qquad \left. + \left(\frac{n_1}{2} + \sqrt{\frac{n_1}{2}\log\frac{1}{\delta}}\right) \cdot \left(\frac{1}{2} \pm o(1)\right) \cdot \|\mathbf{v}\|_2^2 \cdot (\bar{\Lambda}_{-1}^{(t)})^{q-1}\right) \\
&= (1-\eta\lambda) \cdot \widehat{\Lambda}_1^{(t)} + q\eta\left(\left(\frac{1}{4} \pm o(1)\right) \cdot \|\mathbf{v}\|_2^2 \cdot (\widehat{\Lambda}_1^{(t)})^{q-1} + \left(\frac{1}{4} \pm o(1)\right) \cdot \|\mathbf{v}\|_2^2 \cdot (\bar{\Lambda}_{-1}^{(t)})^{q-1}\right) \\
&= (1-\eta\lambda) \cdot \widehat{\Lambda}_1^{(t)} + \eta \cdot \Theta(d) \cdot \left((\widehat{\Lambda}_1^{(t)})^2 + (\bar{\Lambda}_{-1}^{(t)})^{q-1}\right).
\end{aligned}
$$

And we can prove in the same way that with probability at least $1 - 4\delta$ we have

$$
\widehat{\Lambda}_{-1}^{(t+1)} \le (1-\eta\lambda) \cdot \widehat{\Lambda}_{-1}^{(t)} + \eta \cdot \Theta(d) \cdot \left((\widehat{\Lambda}_{-1}^{(t)})^{q-1} + (\bar{\Lambda}_1^{(t)})^{q-1}\right).
$$

Let $j^\star = \arg\max_{m+1\leq j\leq 2m}\langle\mathbf{w}_j^{(t+1)},\mathbf{v}\rangle$ and note that $u_j = -1$, we have

$$
\begin{aligned}
\bar{\Lambda}_1^{(t+1)} &= \langle\mathbf{w}_{j^\star}^{(t+1)},\mathbf{v}\rangle \\
&= (1-\eta\lambda)\cdot\langle\mathbf{w}_{j^\star}^{(t)},\mathbf{v}\rangle - \frac{q\eta}{n_1}\sum_{i=1}^{n_1} b_i^{(t)}[\langle\mathbf{w}_{j^\star}^{(t)},y_i'\cdot\mathbf{v}\rangle]_+^{q-1}\|\mathbf{v}\|_2^2 \\
&\leq (1-\eta\lambda)\cdot\langle\mathbf{w}_{j^\star}^{(t)},\mathbf{v}\rangle \\
&\leq (1-\eta\lambda)\cdot\bar{\Lambda}_1^{(t)}.
\end{aligned}
\tag{A.37}
$$

And we can prove in the same way that $\bar{\Lambda}_{-1}^{(t+1)} \leq (1-\eta\lambda)\cdot\bar{\Lambda}_{-1}^{(t)}$.

Next, we consider the increasing rate of $\Gamma_l'^{(t)}$ where $l \in [n_1]$ is fixed. If $y_l = 1$, let $j^\natural = \arg\max_{1\leq j\leq m}\langle\mathbf{w}_j^{(t)},\boldsymbol{\xi}_l'\rangle$ and note that $u_j = 1$, we have

$$
\begin{aligned}
\Gamma_l'^{(t+1)} &\geq \langle\mathbf{w}_{j^\natural}^{(t+1)},\boldsymbol{\xi}_l'\rangle \\
&= (1-\eta\lambda)\cdot\langle\mathbf{w}_{j^\natural}^{(t)},\boldsymbol{\xi}_l'\rangle + \frac{q\eta}{n_1}\sum_{i=1}^{n_1} b_i^{(t)}y_i'\cdot[\langle\mathbf{w}_{j^\natural}^{(t)},\boldsymbol{\xi}_i'\rangle]_+^{q-1}\cdot\langle\boldsymbol{\xi}_i',\boldsymbol{\xi}_l'\rangle \\
&= (1-\eta\lambda)\cdot\langle\mathbf{w}_{j^\natural}^{(t)},\boldsymbol{\xi}_l'\rangle + \frac{q\eta}{n_1}b_l^{(t)}[\langle\mathbf{w}_{j^\natural}^{(t)},\boldsymbol{\xi}_l'\rangle]_+^{q-1}\|\boldsymbol{\xi}_l'\|_2^2 + \frac{q\eta}{n_1}\sum_{i\in[n_1],i\neq l} b_i^{(t)}y_i'[\langle\mathbf{w}_{j^\natural}^{(t)},\boldsymbol{\xi}_i'\rangle]_+^{q-1}\langle\boldsymbol{\xi}_i',\boldsymbol{\xi}_l'\rangle \\
&= (1-\eta\lambda)\cdot\langle\mathbf{w}_{j^\natural}^{(t)},\boldsymbol{\xi}_l'\rangle + \frac{q\eta}{n_1}b_l^{(t)}[\langle\mathbf{w}_{j^\natural}^{(t)},\boldsymbol{\xi}_l'\rangle]_+^{q-1}\|\boldsymbol{\xi}_l'\|_2^2\{\pm\text{ lower order terms}\} \\
&\geq (1-\eta\lambda)\cdot\Gamma_l'^{(t)} + \frac{q\eta}{n_1}\cdot\left(\frac{1}{2} - o(1)\right)\cdot\|\boldsymbol{\xi}_l'\|_2^2\cdot(\Gamma_l'^{(t)})^{q-1} \\
&= (1-\eta\lambda)\cdot\Gamma_l'^{(t)} + \eta\cdot\widetilde{\Theta}(d^{1+2\epsilon})\cdot(\Gamma_l'^{(t)})^{q-1},
\end{aligned}
\tag{A.38}
$$

where the third equality holds if we properly choose the order of $\lambda$.

If $y_l = -1$, let $j^\sharp = \arg\max_{m+1\leq j\leq 2m}\langle\mathbf{w}_j^{(t)},\boldsymbol{\xi}_l'\rangle$ and note that $u_j = -1$, we have

$$
\begin{aligned}
\Gamma_l'^{(t+1)} &\geq \langle\mathbf{w}_{j^\natural}^{(t+1)},\boldsymbol{\xi}_l'\rangle \\
&= (1-\eta\lambda)\cdot\langle\mathbf{w}_{j^\natural}^{(t)},\boldsymbol{\xi}_l'\rangle - \frac{q\eta}{n_1}\sum_{i=1}^{n_1} b_i^{(t)}y_i'\cdot[\langle\mathbf{w}_{j^\natural}^{(t)},\boldsymbol{\xi}_i'\rangle]_+^{q-1}\cdot\langle\boldsymbol{\xi}_i',\boldsymbol{\xi}_l'\rangle \\
&= (1-\eta\lambda)\cdot\langle\mathbf{w}_{j^\natural}^{(t)},\boldsymbol{\xi}_l'\rangle + \frac{q\eta}{n_1}b_l^{(t)}[\langle\mathbf{w}_{j^\natural}^{(t)},\boldsymbol{\xi}_l'\rangle]_+^{q-1}\|\boldsymbol{\xi}_l'\|_2^2 - \frac{q\eta}{n_1}\sum_{i\in[n_1],i\neq l} b_i^{(t)}y_i'[\langle\mathbf{w}_{j^\natural}^{(t)},\boldsymbol{\xi}_i'\rangle]_+^{q-1}\langle\boldsymbol{\xi}_i',\boldsymbol{\xi}_l'\rangle \\
&= (1-\eta\lambda)\cdot\langle\mathbf{w}_{j^\natural}^{(t)},\boldsymbol{\xi}_l'\rangle + \frac{q\eta}{n_1}b_l^{(t)}[\langle\mathbf{w}_{j^\natural}^{(t)},\boldsymbol{\xi}_l'\rangle]_+^{q-1}\|\boldsymbol{\xi}_l'\|_2^2\{\pm\text{ lower order terms}\} \\
&\geq (1-\eta\lambda)\cdot\Gamma_l'^{(t)} + \frac{q\eta}{n_1}\cdot\left(\frac{1}{2} - o(1)\right)\cdot\|\boldsymbol{\xi}_l'\|_2^2\cdot(\Gamma_l'^{(t)})^{q-1} \\
&= (1-\eta\lambda)\cdot\Gamma_l'^{(t)} + \eta\cdot\widetilde{\Theta}(d^{1+2\epsilon})\cdot(\Gamma_l'^{(t)})^{q-1},
\end{aligned}
\tag{A.39}
$$

where the third equality holds if we properly choose the order of $\lambda$.

According to (A.38) and (A.39), we always have

$$
\Gamma_l'^{(t+1)} \geq (1-\eta\lambda)\cdot\Gamma_l'^{(t)} + \eta\cdot\widetilde{\Theta}(d^{1+2\epsilon})\cdot(\Gamma_l'^{(t)})^{q-1}.
$$

$\square$

## A.6  PROOF OF LEMMA 5.4

By applying Lemma C.4 to $\Gamma_i^{(t)}$ and taking $C_1 = \widetilde{\Theta}(d^{1+2\epsilon})$, $\delta = 1/2$, $A = \Theta(1/m)$, we have

$$
\sum_{t\geq 0,\Gamma_i^{(t)}\leq A}\eta \leq \Theta(1/C_1\Gamma_i^{(t)}) = \widetilde{\Theta}(d^{-\frac{3}{4}-3\epsilon}).
$$

And note the definition of $T_i'$, we have

$$\eta \cdot T_i' = \widetilde{\Theta}(d^{-\frac{3}{4}-3\epsilon}). \tag{A.40}$$

In Lemma 5.2, we have already prove that

$$\widehat{\Lambda}_r^{(t+1)} \leq (1-\eta\lambda) \cdot \widehat{\Lambda}_r^{(t)} + \eta \cdot \Theta(d) \cdot \left( (\widehat{\Lambda}_r^{(t)})^{q-1} + (\bar{\Lambda}_{-r}^{(t)})^{q-1} \right),$$

$$\widehat{\Lambda}_r^{(t+1)} \leq (1-\eta\lambda) \cdot \widehat{\Lambda}_r^{(t+1)}, r \in \{\pm 1\}. \tag{A.41}$$

Define $\Lambda^{(t)} := \max_{r\in\{\pm 1\}}\{\widehat{\Lambda}_r^{(t)}, \bar{\Lambda}_r^{(t)}\}$, according to (A.41), we have

$$\Lambda^{(t+1)} \leq (1-\eta\lambda) \cdot \Lambda^{(t)} + \eta \cdot \Theta(d) \cdot (\Lambda^{(t)})^{q-1}.$$

By applying Lemma C.4 to $\Lambda^{(t)}$, and taking $C_1 = \Theta(d), \delta = 1/2, A = C \cdot \Lambda^{(0)}$, where $A$ is a large constant, we have

$$\sum_{t\geq 0, \Lambda^{(t)}\leq A} \eta \geq \Theta(1/C_1\Lambda^{(0)}) = \widetilde{\Theta}(d^{\frac{1}{4}}).$$

Let $T'$ be the first iteration that $\Lambda^{(t)}$ reaches $C \cdot \Lambda^{(0)}$, then we have

$$\eta \cdot T' = \widetilde{\Theta}(d^{-\frac{3}{4}}). \tag{A.42}$$

According to (A.40) and (A.42), we have $T' = \omega(T_i')$, which indicates that when $\Gamma_i^{(t)}$ reaches $\Theta(1/m)$, $\Lambda^{(t)}$ remains initialization magnitude $\widetilde{\Theta}(d^{-\frac{1}{4}})$.

## A.7 EMPIRICAL, TEST ERROR AND LOSS FOR EARLY STOPPED CLASSIFIER

Assume the accuracy of pseudo-labeler $p$ is larger than $1/2$. We first estimate the empirical loss for early stopped classifier $f_{\mathbf{W}^{(T_0)}}$, where $T_0 = \max_{r\in\{\pm 1\}}\{T_r\}$ and $T_r$ is defined as the first iteration that $\widehat{\Lambda}_r^{(t)}$ reaches $\Theta(1/m)$. According to Section A.4.3 and Lemma A.12, we have $\widehat{\Lambda}_r^{(T_0)} = \widetilde{\Theta}(1), \bar{\Lambda}_r^{(T_0)} = \widetilde{O}(d^{-\frac{1}{4}}), \Gamma^{(t)} = \widetilde{O}(d^{-\frac{1}{4}+\epsilon})$, for $r \in \{\pm 1\}$. We have the following lemma:

**Lemma A.16.** Early stopped classifier $f_{\mathbf{W}^{(T_0)}}(\mathbf{x})$ possesses following properties:

1. Training error of early stopped classifier $f_{\mathbf{W}^{(T_0)}}(\mathbf{x})$ is asymptotically $1-p$: $\frac{1}{n_{\mathrm{u}}+n_{\mathrm{l}}}\left(\sum_{i=1}^{n_{\mathrm{u}}}\mathbb{1}[\widehat{y}_i \cdot f_{\mathbf{W}^{(T_0)}}(\mathbf{x}_i) \leq 0] + \sum_{i=1}^{n_{\mathrm{l}}}\mathbb{1}[y_i' \cdot f_{\mathbf{W}^{(T_0)}}(\mathbf{x}_i') \leq 0]\right) = 1 - p \pm o(1)$.

2. Test error is nearly $1-p$, if we use pseudo-label $\widehat{y}$ generated by pseudo-labeler as target: $\mathbb{P}_{(\mathbf{x},y)\sim\mathcal{D},\widehat{y}\sim y\cdot\mathcal{B}(p)}[\widehat{y} \cdot f_{\mathbf{W}^{(T_0)}}(\mathbf{x}) \leq 0] = 1 - p \pm o(1)$.

3. Test error is nearly $0$, if we use true label $y$ as target: $\mathbb{P}_{(\mathbf{x},y)\sim\mathcal{D}}[y \cdot f_{\mathbf{W}^{(T_0)}}(\mathbf{x}) \leq 0] = o(1)$ and hence $\operatorname{sign} f_{\mathbf{W}^{(T_0)}}(\mathbf{x}) = \operatorname{sign}(y)$ with high probability,

where $p$ is the accuracy of the pseudo-labeler. We can regard $p$ as the probability that $\mathbf{x}_i$ is paired with true label $y_i$, $1-p$ is the probability that $\mathbf{x}_i$ is paired with wrong label $-y_i$.

*Proof of Lemma A.16.* Recall the definition of $f_{\mathbf{W}}$ in (3.1) that

$$f_{\mathbf{W}^{(T_0)}}(\mathbf{x}_i) = \sum_{j=1}^m \left[ \sigma\left(\langle \mathbf{w}_j^{(T_0)}, y_i \cdot \mathbf{v}\rangle\right) + \sigma\left(\langle \mathbf{w}_j^{(T_0)}, \boldsymbol{\xi}_i\rangle\right)\right]$$

$$- \sum_{j=m+1}^{2m} \left[ \sigma\left(\langle \mathbf{w}_j^{(T_0)}, y_i \cdot \mathbf{v}\rangle\right) + \sigma\left(\langle \mathbf{w}_j^{(T_0)}, \boldsymbol{\xi}_i\rangle\right)\right].$$

According to Section A.4.3 and Lemma A.12, we have $\widehat{\Lambda}_r^{(T_0)} = \widetilde{\Theta}(1), \bar{\Lambda}_r^{(T_0)} = \widetilde{O}(d^{-\frac{1}{4}}), \Gamma^{(t)} = \max\left\{\max_{i\in[n_{\mathrm{u}}]}\Gamma_i^{(t)}, \max_{i\in[n_{\mathrm{l}}]}\Gamma_i'^{(t)}\right\} = \widetilde{O}(d^{-\frac{1}{4}+\epsilon})$, for $r \in \{\pm 1\}$. If $y_i = 1$, we have following lower bound for $f_{\mathbf{W}^{(T_0)}}(\mathbf{x}_i)$

$$f_{\mathbf{W}^{(T_0)}}(\mathbf{x}_i) = \sum_{j=1}^m \left[ \sigma\left(\langle \mathbf{w}_j^{(T_0)}, \mathbf{v}\rangle\right) + \sigma\left(\langle \mathbf{w}_j^{(T_0)}, \boldsymbol{\xi}_i\rangle\right)\right] - \sum_{j=m+1}^{2m} \left[ \sigma\left(\langle \mathbf{w}_j^{(T_0)}, \mathbf{v}\rangle\right) + \sigma\left(\langle \mathbf{w}_j^{(T_0)}, \boldsymbol{\xi}_i\rangle\right)\right]$$

$$\geq (\widehat{\Lambda}_1^{(T_0)})^q + (\Gamma_i^{(T_0)})^q - m(\bar{\Lambda}_1^{(T_0)})^q - m(\Gamma_i^{(T_0)})^q$$

$$\geq (\widehat{\Lambda}_1^{(T_0)})^q \{-\text{ lower order terms}\},$$

and following upper bound for $f_{\mathbf{W}^{(T_0)}}(\mathbf{x}_i)$:

$$
\begin{aligned}
f_{\mathbf{W}^{(T_0)}}(\mathbf{x}_i) &= \sum_{j=1}^{m} \left[ \sigma\big(\langle \mathbf{w}_j^{(T_0)}, \mathbf{v}\rangle\big) + \sigma\big(\langle \mathbf{w}_j^{(T_0)}, \boldsymbol{\xi}_i\rangle\big) \right] - \sum_{j=m+1}^{2m} \left[ \sigma\big(\langle \mathbf{w}_j^{(T_0)}, \mathbf{v}\rangle\big) + \sigma\big(\langle \mathbf{w}_j^{(T_0)}, \boldsymbol{\xi}_i\rangle\big) \right] \\
&\le m(\widehat{\Lambda}_1^{(T_0)})^q + m(\Gamma_i^{(T_0)})^q - (\bar{\Lambda}_1^{(T_0)})^q - (\Gamma_i^{(T_0)})^q \\
&\le (\widehat{\Lambda}_1^{(T_0)})^q \{+ \text{ lower order terms}\}.
\end{aligned}
$$

If $y_i = -1$, we have following upper bound for $f_{\mathbf{W}^{(T_0)}}(\mathbf{x}_i)$:

$$
\begin{aligned}
f_{\mathbf{W}^{(T_0)}}(\mathbf{x}_i) &= \sum_{j=1}^{m} \left[ \sigma\big(-\langle \mathbf{w}_j^{(T_0)}, \mathbf{v}\rangle\big) + \sigma\big(\langle \mathbf{w}_j^{(T_0)}, \boldsymbol{\xi}_i\rangle\big) \right] - \sum_{j=m+1}^{2m} \left[ \sigma\big(-\langle \mathbf{w}_j^{(T_0)}, \mathbf{v}\rangle\big) + \sigma\big(\langle \mathbf{w}_j^{(T_0)}, \boldsymbol{\xi}_i\rangle\big) \right] \\
&\le m\big(\bar{\Lambda}_{-1}^{(T_0)}\big)^q + m\big(\Gamma_i^{(T_0)}\big)^q - \big(\widehat{\Lambda}_{-1}^{(T_0)}\big)^q - \big(\Gamma_i^{(T_0)}\big)^q \\
&\le -\big(\widehat{\Lambda}_{-1}^{(T_0)}\big)^q \{+ \text{ lower order terms}\},
\end{aligned}
$$

and following lower bound for $f_{\mathbf{W}^{(T_0)}}(\mathbf{x}_i)$:

$$
\begin{aligned}
f_{\mathbf{W}^{(T_0)}}(\mathbf{x}_i) &= \sum_{j=1}^{m} \left[ \sigma\big(-\langle \mathbf{w}_j^{(T_0)}, \mathbf{v}\rangle\big) + \sigma\big(\langle \mathbf{w}_j^{(T_0)}, \boldsymbol{\xi}_i\rangle\big) \right] - \sum_{j=m+1}^{2m} \left[ \sigma\big(-\langle \mathbf{w}_j^{(T_0)}, \mathbf{v}\rangle\big) + \sigma\big(\langle \mathbf{w}_j^{(T_0)}, \boldsymbol{\xi}_i\rangle\big) \right] \\
&\ge \big(\bar{\Lambda}_{-1}^{(T_0)}\big)^q + \big(\Gamma_i^{(T_0)}\big)^q - m\big(\widehat{\Lambda}_{-1}^{(T_0)}\big)^q - m\big(\Gamma_i^{(T_0)}\big)^q \\
&\ge -m\big(\bar{\Lambda}_{-1}^{(T_0)}\big)^q \{- \text{ lower order terms}\}.
\end{aligned}
$$

Therefore, for unlabeled data, we have $y_i \cdot f_{\mathbf{W}^{(T_0)}}(\mathbf{x}_i) \in \left[ (1-o(1)) \cdot (\widehat{\Lambda}_{y_i}^{(T_0)})^q, (m+o(1)) \cdot (\widehat{\Lambda}_{y_i}^{(T_0)})^q \right]$ and hence $\text{sign}\big(f_{\mathbf{W}^{(T_0)}}(\mathbf{x}_i)\big) = \text{sign}(y_i)$ holds with high probability. We can also prove for labeled data $(\mathbf{x}_i', y_i')$ that $y_i' \cdot f_{\mathbf{W}^{(T_0)}}(\mathbf{x}_i') \in \left[ (1-o(1)) \cdot (\widehat{\Lambda}_{y_i'}^{(T_0)})^q, (m+o(1)) \cdot (\widehat{\Lambda}_{y_i'}^{(T_0)})^q \right]$, $\text{sign}\big(f_{\mathbf{W}^{(T_0)}}(\mathbf{x}_i')\big) = \text{sign}(y_i')$ in the same way.

Note that $\widehat{y}_i$ takes $y_i$ with probability $p$, $-y_i$ with probability $p$ and $n_l = o(n_u)$, the first statement in this lemma follows obviously.

To prove the other two statement, we need to give an upper bound for the norm of $\mathbf{w}_j$. According to the update rule of $\mathbf{w}_j^{(t)}$, we have

$$
\begin{aligned}
\mathbf{w}_j^{(t+1)} &= (1 - \eta\lambda) \cdot \mathbf{w}_j^{(t)} + \frac{q\eta u_j}{n_l + n_u} \cdot \left( \sum_{i=1}^{n_u} c_i \widehat{y}_i \big([\langle \mathbf{w}_j^{(t)}, y_i \cdot \mathbf{v}\rangle]_+^{q-1} \cdot y_i \cdot \mathbf{v} + [\langle \mathbf{w}_j^{(t)}, \boldsymbol{\xi}_i\rangle]_+^{q-1} \cdot \boldsymbol{\xi}_i \big) \right. \\
&\quad \left. + \sum_{i=1}^{n_l} b_i y_i' \big([\langle \mathbf{w}_j^{(t)}, y_i' \cdot \mathbf{v}\rangle]_+^{q-1} \cdot y_i' \cdot \mathbf{v} + [\langle \mathbf{w}_j^{(t)}, \boldsymbol{\xi}_i'\rangle]_+^{q-1} \cdot \boldsymbol{\xi}_i' \big) \right),
\end{aligned}
$$

leading to

$$
\begin{aligned}
\|\mathbf{w}_j^{(t+1)}\|_2 &\le (1-\eta\lambda) \cdot \|\mathbf{w}_j^{(t)}\|_2 + \frac{q\eta}{n_l + n_u} \cdot \left( \sum_{i=1}^{n_u} \big([\langle \mathbf{w}_j^{(t)}, y_i \cdot \mathbf{v}\rangle]_+^{q-1} \cdot \|\mathbf{v}\|_2 + [\langle \mathbf{w}_j^{(t)}, \boldsymbol{\xi}_i\rangle]_+^{q-1} \cdot \|\boldsymbol{\xi}_i\|_2 \big) \right. \\
&\quad \left. + \sum_{i=1}^{n_l} \big([\langle \mathbf{w}_j^{(t)}, y_i' \cdot \mathbf{v}\rangle]_+^{q-1} \cdot \|\mathbf{v}\|_2 + [\langle \mathbf{w}_j^{(t)}, \boldsymbol{\xi}_i'\rangle]_+^{q-1} \cdot \|\boldsymbol{\xi}_i'\|_2 \big) \right) \\
&\le (1-\eta\lambda) \cdot \|\mathbf{w}_j^{(t)}\|_2 + \frac{q\eta}{n_l + n_u} \cdot \left( (n_l + n_u) \cdot \|\mathbf{v}\|_2 \cdot \big( \max_{r \in \{\pm 1\}} \{\widehat{\Lambda}_r^{(t)}, \bar{\Lambda}_r^{(t)}\} \big)^{q-1} \right. \\
&\quad \left. + \Big( \sum_{i \in [n_u]} \|\boldsymbol{\xi}_i\|_2 + \sum_{i \in [n_l]} \|\boldsymbol{\xi}_i'\|_2 \Big) \cdot \big(\Gamma^{(t)}\big)^{q-1} \right) \\
&\le \|\mathbf{w}_j^{(t)}\|_2 + \eta \cdot \left( \Theta(d^{\frac{1}{2}}) \cdot \widetilde{\Theta}(1) + \Theta(d^{\frac{1}{2}+\epsilon}) \cdot \widetilde{O}(d^{(q-1)(-\frac{1}{4}+\epsilon)}) \right) \\
&= \|\mathbf{w}_j^{(t)}\|_2 + \eta \cdot \widetilde{\Theta}(d^{\frac{1}{2}}),
\end{aligned}
$$

$$(A.43)$$

where the first inequality is by triangle inequality; the second inequality is due to the definition of $\widehat{\Lambda}_r^{(t)}, \bar{\Lambda}_r^{(t)}, \Gamma^{(t)}$, the last inequality is due to Lemma 5.3.

According to Lemma A.14, we know that $T_r \cdot \eta = \widetilde{\Theta}(d^{-\frac{3}{4}}), r \in \{\pm 1\}$ and $T_0 \cdot \eta = \max_{r \in \{\pm 1\}} \{T_r \cdot \eta\} = \widetilde{\Theta}(d^{-\frac{3}{4}})$. Note that $\mathbf{w}_j^{(0)} \sim \mathcal{N}(\mathbf{0}, \sigma_0^2 \mathbf{I}_d), \sigma_0 = \Theta(d^{-\frac{3}{4}})$ and hence $\|\mathbf{w}_j^{(0)}\|_2 = \widetilde{\Theta}(d^{-\frac{1}{4}})$, we know that

$$\|\mathbf{w}_j^{(T_0)}\|_2 \leq \|\mathbf{w}_j^{(0)}\|_2 + \eta \cdot T_0 \cdot \widetilde{\Theta}(d^{-\frac{1}{4}}) = \widetilde{\Theta}(d^{-\frac{1}{4}}) + \widetilde{\Theta}(d^{-\frac{1}{4}}) = \widetilde{\Theta}(d^{-\frac{1}{4}}).$$

Therefore, for any $(\mathbf{x}, y)$ sampled from distribution $\mathcal{D}$ where $\mathbf{x} = [y \cdot \mathbf{v}^\top, \boldsymbol{\xi}^\top]^\top$ and $\boldsymbol{\xi} \sim \mathcal{N}(0, \sigma_p^2)$, we have

$$\langle \mathbf{w}_j^{(T_0)}, \boldsymbol{\xi} \rangle \sim \mathcal{N}(0, \sigma_p^2 \|\mathbf{w}_j^{(T_0)}\|_2^2), |\langle \mathbf{w}_j^{(T_0)}, \boldsymbol{\xi} \rangle| = \Theta(\sigma_p \|\mathbf{w}_j^{(T_0)}\|_2) = \widetilde{O}(d^{-\frac{1}{4}+\epsilon}). \quad (A.44)$$

And this indicates that $\langle \mathbf{w}_j^{(T_0)}, \boldsymbol{\xi} \rangle$ will still be dominated by $\langle \mathbf{w}_j^{(T_0)}, \mathbf{v} \rangle$, therefore it holds for newly sampled $(\mathbf{x}, y)$ that

$$y \cdot f_{\mathbf{W}^{(T_0)}}(\mathbf{x}) \in \left[ \left(1 - o(1)\right) \cdot (\widehat{\Lambda}_{y_i}^{(T_0)})^q, \left(m + o(1)\right) \cdot (\widehat{\Lambda}_{y_i}^{(T_0)})^q \right],$$

which means that

$$\mathbb{P}_{(\mathbf{x},y) \sim \mathcal{D}}[y \cdot f_{\mathbf{W}^{(T_0)}}(\mathbf{x}) \leq 0] = o(1).$$

This verifies the third statement that test error is nearly zero.

For the second statement, note that

$$\begin{aligned}
&\mathbb{P}_{(\mathbf{x},y) \sim \mathcal{D}, \widehat{y} \sim y \cdot \mathcal{B}(p)}[\widehat{y} \cdot f_{\mathbf{W}^{(T_0)}}(\mathbf{x}) \leq 0] \\
&= \mathbb{P}_{(\mathbf{x},y) \sim \mathcal{D}}[\widehat{y} \cdot f_{\mathbf{W}^{(T_0)}}(\mathbf{x}) \leq 0 | \widehat{y} = y] \cdot \mathbb{P}_{\widehat{y} \sim y \cdot \mathcal{B}(p)}(\widehat{y} = y) \\
&\quad + \mathbb{P}_{(\mathbf{x},y) \sim \mathcal{D}}[\widehat{y} \cdot f_{\mathbf{W}^{(T_0)}}(\mathbf{x}) \leq 0 | \widehat{y} = -y] \cdot \mathbb{P}_{\widehat{y} \sim y \cdot \mathcal{B}(p)}(\widehat{y} = -y) \\
&= p \cdot \mathbb{P}_{(\mathbf{x},y) \sim \mathcal{D}}[y \cdot f_{\mathbf{W}^{(T_0)}}(\mathbf{x}) \leq 0] + (1 - p) \cdot \mathbb{P}_{(\mathbf{x},y) \sim \mathcal{D}}[y \cdot f_{\mathbf{W}^{(T_0)}}(\mathbf{x}) \geq 0] \\
&= p \cdot o(1) + (1 - p) \cdot (1 - o(1)) \\
&= 1 - p \pm o(1),
\end{aligned}$$

which verifies the second statement. $\qquad\square$

## A.8 DOWNSTREAM TASK

For downstream tasks, we use early stopped classifiers, which are stopped when on-diagonal feature $\widehat{\Lambda}_r^{(t)}$ are learned while off-diagonal feature $\bar{\Lambda}_r^{(t)}$ and noise $\Gamma^{(t)}$ are not memorized. Assume we have learned $K$ early stopped classifiers $f_{\mathbf{W}_1^{(T_0^1)}}(\mathbf{x}), \cdots, f_{\mathbf{W}_K^{(T_0^K)}}(\mathbf{x})$ by using $n_u$ pseudo-labeled data generated by pseudo-labeler $f_1^{\mathrm{w}}, \cdots, f_K^{\mathrm{w}}$ and $n_l$ labeled data.

Then, we want to design a classifier on the learned representation $f_{\mathbf{W}_1^{(T_0^1)}}(\mathbf{x}), \cdots, f_{\mathbf{W}_K^{(T_0^K)}}(\mathbf{x})$ to fit $y$. Here we consider training a downstream linear model

$$g_{\mathbf{a}}(\mathbf{x}) = \sum_{k=1}^{K} a_k f_{\mathbf{W}_k^{(T_0^k)}}(\mathbf{x}),$$

where $a_k \in \mathbb{R}$ denotes the weight as the $k$-th pre-trained model. Given labeled training data $S' = \{(\mathbf{x}_i', y_i')\}_{i=1}^{n_l}$, we want to optimize the empirical loss function

$$L_{S'}(\mathbf{a}) = \frac{1}{n_l} \sum_{i=1}^{n_l} \ell(y_i' \cdot g_{\mathbf{a}}(\mathbf{x}_i')),$$

where $\ell(z) = \log(1 + \exp(-z))$ denotes the cross entropy loss. We initialize $\mathbf{a}$ as zero and optimize empirical loss function by gradient descent, i.e.

$$\mathbf{a}^{(t+1)} = \mathbf{a}^{(t)} - \eta \cdot \nabla_{\mathbf{a}} L_{S'}(\mathbf{a}^{(t)}), \mathbf{a}^{(0)} = \mathbf{0}.$$

In order to estimate the training error and test error for downstream task, we first introduce following lemma about the increasing rate of $\|\mathbf{a}^{(t)}\|_1$.

**Lemma A.17** (Logarithmic increasing rate). For any learning rate $\eta > 0$, $a_k^{(t)}$ will always increase for any $k \in [K]$ and hence $\left\|\mathbf{a}^{(t)}\right\|_1 = \sum_{k=1}^{K} a_k^{(t)}$. And it holds that $\left\|\mathbf{a}^{(t)}\right\|_1 = \Theta(\log(t))$.

In order to give the increasing rate of $\left\|\mathbf{a}^{(t)}\right\|_1$, we introduce and prove the following lemma:

**Lemma A.18.** Consider following sequence $\{x_t\}_{t=1}^{\infty}$ with

$$x_{t+1} = x_t + C \cdot a^{-x_t}, x_0 = 0,$$

where $a > 1$ and $C > 0$ are constants, and it follows that

$$\log_a \left( \ln a \cdot C \cdot t + 1 \right) \leq x_t \leq \log_a \left( \ln a \cdot C \cdot t + 1 \right) + C,$$

and

$$x_{t+1} - x_t \leq \frac{C}{C \cdot \ln a \cdot t + 1}.$$

*Proof of Lemma A.18.* Note that

$$x_{i+1} - x_i = C \cdot a^{-x_i} \iff a^{x_i}(x_{i+1} - x_i) = C,$$

by adding up above equation from $i = 0$ to $i = t - 1$, we have

$$\sum_{i=0}^{t-1} a^{x_i}(x_{i+1} - x_i) = C \cdot t \tag{A.45}$$

$$\implies \int_{x_0}^{x_t} a^x dx \geq C \cdot t$$

$$\implies \frac{a^{x_t} - a^{x_0}}{\ln a} \geq C \cdot t$$

$$\implies a^{x_t} \geq C \cdot \ln a \cdot t + 1$$

$$\implies \begin{cases} x_t \geq \log_a \left( C \cdot \ln a \cdot t + 1 \right), \\ x_{t+1} - x_t = C \cdot a^{-x_t} \leq \frac{C}{C \cdot \ln a \cdot t + 1}, \end{cases}$$

where the first arrow is due to $a^x$ is monotone increasing.

On the other hand,

$$a^{x_{i+1}} = a^{x_i + C \cdot a^{-x_i}} = a^{x_i} \cdot a^{C \cdot a^{-x_i}} \leq a^{x_i} \cdot a^{C/(C \cdot \ln a \cdot i + 1)} \leq a^{x_i} \cdot a^C,$$

which implies

$$\sum_{i=0}^{t-1} a^{x_{i+1}} \cdot (x_{i+1} - x_t) \leq a^C \sum_{i=0}^{t-1} a^{x_i} \cdot (x_{i+1} - x_i)$$

$$\implies \sum_{i=0}^{t-1} a^{x_{i+1}} \cdot (x_{i+1} - x_i) \leq a^C \cdot Ct$$

$$\implies \int_{x_0}^{x_t} a^x dx \leq a^C \cdot Ct,$$

where the first arrow is due to (A.45) and the last arrow is due to $a^x$ is monotone increasing.
This leads to

$$x_t \leq \log_a \left( \ln a \cdot C \cdot a^C \cdot n + 1 \right)$$

$$\leq \log_a \left( \ln a \cdot C \cdot a^C \cdot n + a^C \right)$$

$$= \log_a \left( \ln a \cdot C \cdot t + 1 \right) + C$$

Therefore, we have

$$\log_a \left( \ln a \cdot C \cdot t + 1 \right) \leq x_t \leq \log_a \left( \ln a \cdot C \cdot t + 1 \right) + C,$$

and

$$x_{t+1} - x_t \leq \frac{C}{\ln a \cdot C \cdot t + 1}.$$

$\square$

Now we are ready to prove Lemma A.17.

*Proof of Lemma A.17.* Note that we take downstream task linear model $g_{\mathbf{a}}(\mathbf{x})$ as

$$
\begin{aligned}
g_{\mathbf{a}}(\mathbf{x}) &= \sum_{k=1}^{d} a_k \Bigg\{ \sum_{j=1}^{m} \Big[ \sigma\big(\langle \mathbf{w}_{k,j}^{(T_0^k)}, y \cdot \mathbf{v} \rangle\big) + \sigma\big(\langle \mathbf{w}_{k,j}^{(T_0^k)}, \boldsymbol{\xi} \rangle\big) \Big] \\
&\qquad\qquad - \sum_{j=m+1}^{2m} \Big[ \sigma\big(\langle \mathbf{w}_{k,j}^{(T_0^k)}, y \cdot \mathbf{v} \rangle\big) + \sigma\big(\langle \mathbf{w}_{k,j}^{(T_0^k)}, \boldsymbol{\xi} \rangle\big) \Big] \Bigg\} \\
&= \sum_{k=1}^{d} a_k f_{\mathbf{W}_k^{(T_0^k)}}(\mathbf{x}).
\end{aligned}
$$

Then, we have following update rule for model parameter $\mathbf{a}$:

$$
a_k^{(t+1)} = a_k^{(t)} - \eta \cdot \frac{1}{n_1} \sum_{i=1}^{n_1} \ell'\big(y_i' \cdot g_{\mathbf{a}^{(t)}}(\mathbf{x}_i')\big) \cdot y_i' f_{\mathbf{W}_k^{(T_0^k)}}(\mathbf{x}_i'),
$$

where we initialize $a_k^{(0)}$ as zero for all $k \in [K]$.

Next, we prove following statement by using induction method: when $t \geq 1$,

- $a_k^{(t)}, \forall k \in [K]$ is non-negative and increasing.

- $\big\| \mathbf{a}^{(t)} \big\|_1 = \sum_{i=1}^{K} a_k^{(t)}$.

- $a_k^{(t+1)} = a_k^{(t)} + \eta \cdot \widetilde{\Theta}(1) \cdot \Big( \exp\big( -\|\mathbf{a}^{(1)}\|_1 \cdot \widetilde{\Theta}(1) \big) \Big), \forall k \in [K]$.

Note that $a_k^{(0)} = 0$ for all $k \in [d]$ and therefore $g_{\mathbf{a}^{(0)}}(\mathbf{x}_i') = 0$, $\ell'\big(y_i' \cdot g_{\mathbf{a}^{(0)}}(\mathbf{x}_i')\big) = \ell'(0) = -1/2$,

$$
\begin{aligned}
a_k^{(1)} &= a_k^{(0)} - \eta \cdot \frac{1}{n_1} \sum_{i=1}^{n_1} \ell'\big(y_i' \cdot g_{\mathbf{a}^{(0)}}(\mathbf{x}_i')\big) \cdot y_i' f_{\mathbf{W}_k^{(T_0^k)}}(\mathbf{x}_i') \\
&= a_k^{(0)} + \eta \cdot \frac{1}{2n_1} \sum_{i=1}^{n_1} y_i' f_{\mathbf{W}_k^{(T_0^k)}}(\mathbf{x}_i') = \eta \cdot \frac{1}{2n_1} \sum_{i=1}^{n_1} y_i' f_{\mathbf{W}_k^{(T_0^k)}}(\mathbf{x}_i') \text{ for all } k \in [K].
\end{aligned}
$$

Note that the accuracy of the $k$-th pseudo-labeler $p_k > 1/2$, accoring to the proof of Lemma A.16, we have

$$
\begin{aligned}
f_{\mathbf{W}_k^{(T_0^k)}}(\mathbf{x}_i') &= \sum_{j=1}^{m} \Big[ \sigma\big(\langle \mathbf{w}_{k,j}^{(T_0^k)}, y_i' \cdot \mathbf{v} \rangle\big) + \sigma\big(\langle \mathbf{w}_{k,j}^{(T_0^k)}, \boldsymbol{\xi}_i' \rangle\big) \Big] \\
&\qquad\qquad - \sum_{j=m+1}^{2m} \Big[ \sigma\big(\langle \mathbf{w}_{k,j}^{(T_0^k)}, y_i' \cdot \mathbf{v} \rangle\big) + \sigma\big(\langle \mathbf{w}_{k,j}^{(T_0^k)}, \boldsymbol{\xi}_i' \rangle\big) \Big] \\
&= y_i' \cdot \widetilde{\Theta}\big((\widehat{\Lambda}_{y_i'}^{(T_0^k)})^q\big),
\end{aligned}
$$

for all $k \in [K]$. Therefore

$$
a_k^{(1)} = \eta \cdot \frac{1}{2n_1} \sum_{i=1}^{n_1} y_i' f_{\mathbf{W}_k^{(T_0^k)}}(\mathbf{x}_i') \geq \frac{\eta}{2} \cdot \widetilde{\Theta}\big((\widehat{\Lambda}_{y_i'}^{(T_0^k)})^q\big) > 0, \forall k \in [K].
$$

It follows that

$$
\big\| \mathbf{a}^{(t)} \big\|_1 = \sum_{i=1}^{K} |a_k^{(t)}| = \sum_{i=1}^{K} a_k^{(t)}.
$$

Note that

$$
\begin{aligned}
y_i' \cdot g_{\mathbf{a}^{(1)}}(\mathbf{x}_i') &= y_i' \cdot \sum_{k=1}^{K} a_k^{(1)} f_{\mathbf{W}_k^{(T_0^k)}}(\mathbf{x}_i') \\
&= \sum_{k=1}^{K} a_k^{(1)} \cdot \left( y_i' \cdot f_{\mathbf{W}_k^{(T_0^k)}}(\mathbf{x}_i') \right) \\
&= \sum_{k=1}^{K} a_k^{(1)} \cdot \widetilde{\Theta}\big( (\widehat{\Lambda}_{y_i'}^{(T_0^k)})^q \big) \\
&= \sum_{k=1}^{K} a_k^{(1)} \cdot \widetilde{\Theta}(1) \\
&= \|\mathbf{a}^{(1)}\|_1 \cdot \widetilde{\Theta}(1).
\end{aligned}
\tag{A.46}
$$

This leads to

$$
\begin{aligned}
\ell'\big( y_i' \cdot g_{\mathbf{a}^{(1)}}(\mathbf{x}_i') \big) &= -\frac{\exp\left( -y_i' \cdot g_{\mathbf{a}^{(1)}}(\mathbf{x}_i') \right)}{1 + \exp\left( -y_i' \cdot g_{\mathbf{a}^{(1)}}(\mathbf{x}_i') \right)} \\
&= -c \cdot \left( \exp\left( -y_i' \cdot g_{\mathbf{a}^{(1)}}(\mathbf{x}_i') \right) \right) \\
&= -c \cdot \left( \exp\left( -\|\mathbf{a}^{(1)}\|_1 \cdot \widetilde{\Theta}(1) \right) \right),
\end{aligned}
$$

where the second equality is due to $y_i' \cdot g_{\mathbf{a}^{(1)}}(\mathbf{x}_i') > 0$, $\exp\left( -y_i' \cdot g_{\mathbf{a}^{(1)}}(\mathbf{x}_i') \right) < 1$ and $c \in (1/2, 1)$; the last equality is due to (A.46). It follows that

$$
\begin{aligned}
a_k^{(2)} &= a_k^{(1)} - \eta \cdot \frac{1}{n_1} \sum_{i=1}^{n_1} \ell'\big( y_i' \cdot g_{\mathbf{a}^{(1)}}(\mathbf{x}_i') \big) \cdot y_i' f_{\mathbf{W}_k^{(T_0)}}(\mathbf{x}_i') \\
&= a_k^{(1)} + \eta \cdot c \cdot \widetilde{\Theta}(1) \cdot \left( \exp\left( -\|\mathbf{a}^{(1)}\|_1 \cdot \widetilde{\Theta}(1) \right) \right), \forall k \in [K]
\end{aligned}
$$

where $c \in (1/2, 1)$. By then, we have already proved the induction hypothesis of $t = 1$.

Next, assume the induction hypotheses hold for $t$. For $t + 1$, we have

$$
a_k^{(t+1)} = a_k^{(t)} - \eta \cdot \frac{1}{n_1} \sum_{i=1}^{n_1} \underbrace{\ell'\big( y_i' \cdot g_{\mathbf{a}^{(t)}}(\mathbf{x}_i') \big)}_{<0} \cdot \underbrace{y_i' f_{\mathbf{W}_k^{(T_0^k)}}(\mathbf{x}_i')}_{>0} > a_k^{(t)} > 0.
$$

And it follows that

$$
\|\mathbf{a}^{(t+1)}\|_1 = \sum_{i=1}^{K} a_k^{(t+1)} \quad \text{and} \quad y_i' \cdot g_{\mathbf{a}^{(t+1)}}(\mathbf{x}_i') = \|\mathbf{a}^{(t+1)}\|_1 \cdot \widetilde{\Theta}(1),
\tag{A.47}
$$

leading to

$$
\ell'\big( y_i' \cdot g_{\mathbf{a}^{(t+1)}}(\mathbf{x}_i') \big) = -c \cdot \left( \exp\left( -\|\mathbf{a}^{(t+1)}\|_1 \cdot \widetilde{\Theta}(1) \right) \right), c \in (1/2, 1),
$$

and

$$
a_k^{(t+2)} = a_k^{(t+1)} + \eta \cdot \widetilde{\Theta}(1) \cdot \left( \exp\left( -\|\mathbf{a}^{(t+1)}\|_1 \cdot \widetilde{\Theta}(1) \right) \right), \forall k \in [K].
$$

This indicates that if induction hypotheses hold for $t$, then they holds for $t + 1$.

Adding up $k \in [K]$, we can obtain

$$
\|\mathbf{a}^{(t+1)}\|_1 = \|\mathbf{a}^{(t)}\|_1 + \eta \cdot \widetilde{\Theta}(1) \cdot \exp\left( -\widetilde{\Theta}(1) \cdot \|\mathbf{a}^{(t)}\|_1 \right)
\tag{A.48}
$$

According to Lemma A.18, we know that $\|\mathbf{a}^{(t)}\|_1 = \log t / \widetilde{\Theta}(1)\{\pm \text{ lower order terms w.r.t. t}\}$. $\quad\square$

The following lemma gives the convergence guarantee of downstream task:

**Lemma A.19.** (Convergence Guarantee) For any learning rate $\eta > 0$,

$$\|\nabla_{\mathbf{a}} L_{S'}(\mathbf{a}^{(t)})\|_1 \leq \frac{\widetilde{\Theta}(1)}{\eta \cdot \widetilde{\Theta}(1) \cdot t + 1} \text{ and } \nabla_{\mathbf{a}}^2 L_S(\mathbf{a}) \succeq 0 \text{ for any } \mathbf{a} \in \mathbb{R}^d,$$

which means within polynomial steps, gradient descent is guaranteed to find a point with small gradient.

*Proof of Lemma A.19.* Note that

$$
\begin{aligned}
\|\nabla_{\mathbf{a}} L_{S'}(\mathbf{a}^{(t)})\|_1 &= \sum_{k=1}^{K} |\partial_{a_k} L_{S'}(\mathbf{a}^{(t)})| \\
&= -\sum_{k=1}^{K} \partial_{a_k} L_{S'}(\mathbf{a}^{(t)}) \\
&= \sum_{k=1}^{K} \frac{a_k^{(t+1)} - a_k^{(t)}}{\eta} \\
&= \frac{\|\mathbf{a}^{(t+1)}\|_1 - \|\mathbf{a}^{(t)}\|_1}{\eta},
\end{aligned}
$$

then according to Lemma A.18 and (A.48), we know

$$\|\mathbf{a}^{(t+1)}\|_1 - \|\mathbf{a}^{(t)}\|_1 \leq \frac{\eta \cdot \widetilde{\Theta}(1)}{\eta \cdot \widetilde{\Theta}(1) \cdot t + 1}. \tag{A.49}$$

And it follows that

$$\|\nabla_{\mathbf{a}} L_{S'}(\mathbf{a}^{(t)})\|_1 \leq \frac{\widetilde{\Theta}(1)}{\eta \cdot \widetilde{\Theta}(1) \cdot t + 1},$$

which shows that within polynomial steps, gradient descent is guaranteed to find a point with small gradient.

Note that

$$\partial_{a_k} L_{S'}(\mathbf{a}) = \frac{1}{n_1} \sum_{i=1}^{n_1} \ell'\left(y_i' \cdot g_{\mathbf{a}^{(t)}}(\mathbf{x}_i')\right) \cdot y_i' f_{\mathbf{W}_k^{(T_0^k)}}(\mathbf{x}_i'),$$

$$\partial_{a_k} \partial_{a_j} L_{S'}(\mathbf{a}) = \frac{1}{n_1} \sum_{i=1}^{n_1} \ell''\left(y_i' \cdot g_{\mathbf{a}^{(t)}}(\mathbf{x}_i')\right) \cdot \left(f_{\mathbf{W}_k^{(T_0^k)}}(\mathbf{x}_i) \cdot f_{\mathbf{W}_j^{(T_0^j)}}(\mathbf{x}_i)\right) \text{ for all } k, j \in [K],$$

Denote $\left[f_{\mathbf{W}_1^{(T_0^1)}}(\mathbf{x}_i'), \cdots, f_{\mathbf{W}_K^{(T_0^K)}}(\mathbf{x}_i')\right]^{\top}$ as $\mathbf{f}_{\mathbf{W}^*}(\mathbf{x}_i')$, then

$$\nabla_{\mathbf{a}}^2 L_S(\mathbf{a}) = \frac{1}{n_1} \sum_{i=1}^{n_1} \ell''\left(y_i' \cdot g_{\mathbf{a}^{(t)}}(\mathbf{x}_i')\right) \cdot \left(\mathbf{f}_{\mathbf{W}^*}(\mathbf{x}_i') \cdot \mathbf{f}_{\mathbf{W}^*}(\mathbf{x}_i')^{\top}\right).$$

Note that $\mathbf{f}_{\mathbf{W}^*}(\mathbf{x}_i') \cdot \mathbf{f}_{\mathbf{W}^*}(\mathbf{x}_i')^{\top}$ is a non-negative definite matrix, $\ell''(z) = \exp(-z)/(1 + \exp(-z))^2 > 0$ and the fact that sum of non-negative definite matrices is still a non-negative definite matrix, it follows that $\nabla_{\mathbf{a}}^2 L_S(\mathbf{a}) \succeq 0$. □

**Theorem A.20** (Restatement of Theorem 4.3). Under semi-supervised learning setting, for downstream task, suppose $K$ early stopped classifiers $\{f_{\mathbf{W}_k^*}\}_{k=1}^K$ are obtained after the pre-training of $K$ CNN models finished, and after $T_{\text{dt}} = \Theta(d^{0.1}/\eta)$ iterations with learning rate $\eta = \Theta(1)$, then we can find a linear model $\mathbf{a}^{(T_{\text{dt}})}$, which satisfies: Both test error and loss are nearly 0, i.e. $\mathbb{P}_{(\mathbf{x},y)\sim\mathcal{D}}[y \cdot g_{\mathbf{a}^{(T_{\text{dt}})}}(\mathbf{x}) \leq 0] = o(1), L_{\mathcal{D}}(\ell(y \cdot g_{\mathbf{a}^{(T_{\text{dt}})}}(\mathbf{x}))) = o(1).$

*Proof of Theorem A.20.* For test error, we have

$$\mathbb{P}_{(\mathbf{x},y)\sim\mathcal{D}}[y \cdot g_{a^{(T_{\mathrm{dt}})}}(\mathbf{x}) \leq 0] = \mathbb{P}_{(\mathbf{x},y)\sim\mathcal{D}}\left[\sum_{k=1}^{K} a_k^{(T_{\mathrm{dt}})} \cdot \left(y \cdot f_{\mathbf{W}_k^*}(\mathbf{x})\right) \leq 0\right]$$

$$= \mathbb{P}_{(\mathbf{x},y)\sim\mathcal{D}}\left[\sum_{k=1}^{K} a_k^{(T_{\mathrm{dt}})} \cdot \widetilde{\Theta}(1) \leq 0\right] = o(1)$$

where the last equality is due to $a_k^{(T_{\mathrm{dt}})} > 0$ according to Lemma A.17.

For test loss, we have

$$L_{\mathcal{D}}(\ell(y \cdot g_{\mathbf{a}^{(T_{\mathrm{dt}})}}(\mathbf{x}))) = \mathbb{E}_{(\mathbf{x},y)\sim\mathcal{D}}[\ell(y \cdot g_{\mathbf{a}^{(T_{\mathrm{dt}})}}(\mathbf{x}))],$$

i.e., we estimate for newly generated data $(\mathbf{x}, y)$ the magnitude of $\ell(y \cdot g_{\mathbf{a}^{(t)}}(\mathbf{x}))$. In order to do so, we will first estimate $\ell(y_i' \cdot g_{\mathbf{a}^{(t)}}(\mathbf{x}_i))$. Then, we will show that $\ell(y \cdot g_{\mathbf{a}^{(t)}}(\mathbf{x}))$ and $\ell(y_i' \cdot g_{\mathbf{a}^{(t)}}(\mathbf{x}_i))$ nearly equal to each other.

According to the update rule of $a_k^{(t)}$, we have

$$a_k^{(t+1)} = a_k^{(t)} - \eta \cdot \frac{1}{n_{\mathrm{l}}} \sum_{i=1}^{n_{\mathrm{l}}} \ell'\left(y_i' \cdot g_{\mathbf{a}^{(t)}}(\mathbf{x}_i')\right) \cdot y_i' f_{\mathbf{W}_k^{(T_0^k)}}(\mathbf{x}_i').$$

Adding up the above equation for $k \in [K]$, we obtain

$$\|\mathbf{a}^{(t+1)}\|_1 = \|\mathbf{a}^{(t)}\|_1 - \eta \cdot \frac{1}{n_{\mathrm{l}}} \sum_{i=1}^{n_{\mathrm{l}}} \ell'\left(y_i' \cdot g_{\mathbf{a}^{(t)}}(\mathbf{x}_i')\right) \cdot y_i' \sum_{k=1}^{K} f_{\mathbf{W}_k^{(T_0^k)}}(\mathbf{x}_i').$$

And according to (A.49), we have

$$\|\mathbf{a}^{(t+1)}\|_1 - \|\mathbf{a}^{(t)}\|_1 \leq \frac{\eta \cdot \widetilde{\Theta}(1)}{\eta \cdot \widetilde{\Theta}(1) \cdot t + 1},$$

therefore it follows that

$$-\frac{1}{n_{\mathrm{l}}} \sum_{i=1}^{n_{\mathrm{l}}} \ell'\left(y_i' \cdot g_{\mathbf{a}^{(t)}}(\mathbf{x}_i')\right) \cdot y_i' \sum_{k=1}^{K} f_{\mathbf{W}_k^{(T_0^k)}}(\mathbf{x}_i') \leq \frac{\widetilde{\Theta}(1)}{\eta \cdot \widetilde{\Theta}(1) \cdot t + 1}.$$

Note that $K = \Theta(1)$ and for all $k \in [K]$ we have $y_i' \cdot f_{\mathbf{W}_k^{(T_0^k)}}(\mathbf{x}_i') = \widetilde{\Theta}(1)$, it follows that

$$-\frac{1}{n_{\mathrm{l}}} \sum_{i=1}^{n_{\mathrm{l}}} \ell'\left(y_i' \cdot g_{\mathbf{a}^{(t)}}(\mathbf{x}_i')\right) \leq \frac{\widetilde{\Theta}(1)}{\eta \cdot \widetilde{\Theta}(1) \cdot t + 1}.$$

Note that $n_{\mathrm{l}} = \widetilde{\Theta}(1)$ and according to Lemma A.8, there exists a positive sample $(\mathbf{x}_{i_1}, y_{i_1})$ and a negative sample $(\mathbf{x}_{i_2}, y_{i_2})$ with the property that

$$-\ell'\left(y_{i_1}' \cdot g_{\mathbf{a}^{(t)}}(\mathbf{x}_{i_1}')\right) \leq \frac{\widetilde{\Theta}(1)}{\eta \cdot \widetilde{\Theta}(1) \cdot t + 1}, \quad -\ell'\left(y_{i_2}' \cdot g_{\mathbf{a}^{(t)}}(\mathbf{x}_{i_2}')\right) \leq \frac{\widetilde{\Theta}(1)}{\eta \cdot \widetilde{\Theta}(1) \cdot t + 1}.$$

Note that $\ell(z) = \log(1 + \exp(-z))$ and $\ell'(z) = -\exp(-z)/(1 + \exp(-z))$, we know that for $z > 0$,

$$-\ell'(z) = c \cdot \exp(-z),$$
$$\ell(z) < \exp(-z) = -\ell'(z)/c, c \in (1/2, 1).$$

It follows that

$$\ell\left(y_{i_1}' \cdot g_{\mathbf{a}^{(t)}}(\mathbf{x}_{i_1}')\right) \leq \frac{\widetilde{\Theta}(1)}{\eta \cdot \widetilde{\Theta}(1) \cdot t + 1}, \quad \ell\left(y_{i_2}' \cdot g_{\mathbf{a}^{(t)}}(\mathbf{x}_{i_2}')\right) \leq \frac{\widetilde{\Theta}(1)}{\eta \cdot \widetilde{\Theta}(1) \cdot t + 1}.$$

Note that $\ell(z)$ is 1-Lipschitz, we have

$$
\begin{aligned}
\left|\ell\big(y \cdot g_{\mathbf{a}^{(t)}}(\mathbf{x})\big) - \ell\big(y'_{i_1} \cdot g_{\mathbf{a}^{(t)}}(\mathbf{x}'_{i_1})\big)\right| &\leq \left|y \cdot g_{\mathbf{a}^{(t)}}(\mathbf{x}) - y'_{i_1} \cdot g_{\mathbf{a}^{(t)}}(\mathbf{x}'_{i_1})\right|, \\
\left|\ell\big(y \cdot g_{\mathbf{a}^{(t)}}(\mathbf{x})\big) - \ell\big(y'_{i_2} \cdot g_{\mathbf{a}^{(t)}}(\mathbf{x}'_{i_2})\big)\right| &\leq \left|y \cdot g_{\mathbf{a}^{(t)}}(\mathbf{x}) - y'_{i_2} \cdot g_{\mathbf{a}^{(t)}}(\mathbf{x}'_{i_2})\right|.
\end{aligned}
\tag{A.50}
$$

If $y = 1$, we have

$$
\begin{aligned}
\left|y \cdot g_{\mathbf{a}^{(t)}}(\mathbf{x}) - y'_{i_1} \cdot g_{\mathbf{a}^{(t)}}(\mathbf{x}'_{i_1})\right| &= \left|g_{\mathbf{a}^{(t)}}(\mathbf{x}) - g_{\mathbf{a}^{(t)}}(\mathbf{x}'_{i_1})\right| \\
&= \left|\sum_{k=1}^{K} a_k^{(t)} f_{\mathbf{W}_k^{(T_0^k)}}(\mathbf{x}) - \sum_{k=1}^{K} a_k^{(t)} f_{\mathbf{W}_k^{(T_0^k)}}(\mathbf{x}'_{i_1})\right| \\
&= \left|\sum_{k=1}^{K} a_k^{(t)}\Big(f_{\mathbf{W}_k^{(T_0^k)}}(\mathbf{x}) - f_{\mathbf{W}_k^{(T_0^k)}}(\mathbf{x}'_{i_1})\Big)\right|,
\end{aligned}
\tag{A.51}
$$

and

$$
\begin{aligned}
f_{\mathbf{W}_k^{(T_0^k)}}(\mathbf{x}) - f_{\mathbf{W}_k^{(T_0^k)}}(\mathbf{x}'_{i_1}) &= \sum_{j=1}^{m}\left[\sigma\big(\langle \mathbf{w}_j^{(T_0^k)}, \mathbf{v}\rangle\big) + \sigma\big(\langle \mathbf{w}_j^{(T_0^k)}, \boldsymbol{\xi}\rangle\big)\right] \\
&\quad - \sum_{j=m+1}^{2m}\left[\sigma\big(\langle \mathbf{w}_j^{(T_0^k)}, \mathbf{v}\rangle\big) + \sigma\big(\langle \mathbf{w}_j^{(T_0^k)}, \boldsymbol{\xi}\rangle\big)\right] \\
&\quad - \sum_{j=1}^{m}\left[\sigma\big(\langle \mathbf{w}_j^{(T_0^k)}, \mathbf{v}\rangle\big) + \sigma\big(\langle \mathbf{w}_j^{(T_0^k)}, \boldsymbol{\xi}'_{i_1}\rangle\big)\right] \\
&\quad + \sum_{j=m+1}^{2m}\left[\sigma\big(\langle \mathbf{w}_j^{(T_0^k)}, \mathbf{v}\rangle\big) + \sigma\big(\langle \mathbf{w}_j^{(T_0^k)}, \boldsymbol{\xi}'_{i_1}\rangle\big)\right] \\
&= \sum_{j=1}^{m}\left[\sigma\big(\langle \mathbf{w}_j^{(T_0^k)}, \boldsymbol{\xi}\rangle\big) - \sigma\big(\langle \mathbf{w}_j^{(T_0^k)}, \boldsymbol{\xi}'_{i_1}\rangle\big)\right] \\
&\quad + \sum_{j=m+1}^{2m}\left[\sigma\big(\langle \mathbf{w}_j^{(T_0^k)}, \boldsymbol{\xi}'_{i_1}\rangle\big) - \sigma\big(\langle \mathbf{w}_j^{(T_0^k)}, \boldsymbol{\xi}\rangle\big)\right] \\
&= \widetilde{O}(d^{-\frac{1}{4}+\epsilon}),
\end{aligned}
\tag{A.52}
$$

where the last equality is due to (A.44) and Lemma 5.3.

Plugging (A.52) into (A.51), we have

$$
\left|y \cdot g_{\mathbf{a}^{(t)}}(\mathbf{x}) - y'_{i_1} \cdot g_{\mathbf{a}^{(t)}}(\mathbf{x}'_{i_1})\right| = \widetilde{O}(d^{-\frac{1}{4}+\epsilon}) \cdot \|\mathbf{a}^{(t)}\|_1.
\tag{A.53}
$$

If $y = -1$, we can prove in a similar way that

$$
\left|y \cdot g_{\mathbf{a}^{(t)}}(\mathbf{x}) - y'_{i_2} \cdot g_{\mathbf{a}^{(t)}}(\mathbf{x}'_{i_2})\right| = \widetilde{O}(d^{-\frac{1}{4}+\epsilon}) \cdot \|\mathbf{a}^{(t)}\|_1.
\tag{A.54}
$$

Plugging (A.53) and (A.54) into (A.50), we have

$$
\ell\big(y \cdot g_{\mathbf{a}^{(t)}}(\mathbf{x})\big) \leq \max\left\{y'_{i_1} \cdot g_{\mathbf{a}^{(t)}}(\mathbf{x}'_{i_1}), y'_{i_2} \cdot g_{\mathbf{a}^{(t)}}(\mathbf{x}'_{i_2})\right\} + \widetilde{O}(d^{-\frac{1}{4}+\epsilon}) \cdot \|\mathbf{a}^{(t)}\|_1
$$

According to Lemma A.18 and (A.48), we have $\|\mathbf{a}^{(t)}\|_1 = \log t/\widetilde{\Theta}(1)\{\pm \text{ lower order terms w.r.t. } t\}$, therefore

$$
\ell\big(y \cdot g_{\mathbf{a}^{(t)}}(\mathbf{x})\big) \leq \frac{\widetilde{\Theta}(1)}{\eta \cdot \widetilde{\Theta}(1) \cdot t + 1} + \widetilde{O}(d^{-\frac{1}{4}+\epsilon}) \cdot \log t \{\pm \text{ lower order terms w.r.t. } t\}
$$

Taking $\eta = \Theta(1)$ and $T_{dt} = \Theta(d^\alpha/\eta)$ where $\alpha > 0$ is a sufficiently small constant, we know that

$$
\begin{aligned}
&L_{\mathcal{D}}(\ell(y \cdot g_{\mathbf{a}^{(T_{dt})}}(\mathbf{x}))) \\
&= \mathbb{E}_{(\mathbf{x},y)\sim\mathcal{D}}[\ell(y \cdot g_{\mathbf{a}^{(T_{dt})}}(\mathbf{x}))] \\
&\leq \frac{\widetilde{\Theta}(1)}{\eta \cdot \widetilde{\Theta}(1) \cdot T_{dt} + 1} + \widetilde{O}(d^{-\frac{1}{4}+\epsilon}) \cdot \log T_{dt} \{\pm \text{ lower order terms w.r.t. } T_{dt}\} + o(1) \\
&= o(1),
\end{aligned}
$$

which completes the proof. $\qquad\square$

## B PROOF OF SUPERVISED LEARNING SETTING

Here we prove Theorem 4.4. First, we give following lemma to facilitate the proof.

**Lemma B.1** (Gradient Calculation). The gradient of loss function $L_S(\mathbf{W})$ with respect to weight parameter $\mathbf{w}_j$ is

$$\nabla_{\mathbf{w}_j} L_{S'}(\mathbf{W}) = -\frac{qu_j}{n_1} \cdot \sum_{i=1}^{n_1} b_i y_i' \big( [\langle \mathbf{w}_j, y_i' \cdot \mathbf{v} \rangle]_+^{q-1} \cdot y_i' \cdot \mathbf{v} + [\langle \mathbf{w}_j, \boldsymbol{\xi}_i' \rangle]_+^{q-1} \cdot \boldsymbol{\xi}_i' \big),$$

where $u_j := \big( \mathbb{1}_{[1 \le j \le m]} - \mathbb{1}_{[m+1 \le j \le 2m]} \big)$ and $-\ell'\big(y_i' \cdot f_{\mathbf{W}}(\mathbf{x}_i')\big) = \exp\left[-y_i' \cdot f_{\mathbf{W}}(\mathbf{x}_i')\right]/(1 + \exp\left[-y_i' \cdot f_{\mathbf{W}}(\mathbf{x}_i')\right])$ is denoted as $b_i$.

*Proof of Lemma B.1.* When $1 \le j \le m$,

$$\begin{aligned}
\nabla_{\mathbf{w}_j} \ell\big(y_i' \cdot f_{\mathbf{W}}(\mathbf{x}_i')\big) &= \ell'\big(y_i' \cdot f_{\mathbf{W}}(\mathbf{x}_i')\big) \cdot y_i' \cdot \nabla_{\mathbf{w}_j} f_{\mathbf{W}}(\mathbf{x}_i') \\
&= -b_i \cdot y_i' \cdot \nabla_{\mathbf{w}_j} f_{\mathbf{W}}(\mathbf{x}_i') \\
&= -b_i y_i' \cdot \big( \sigma'\big(\langle \mathbf{w}_j, y_i' \cdot \mathbf{v} \rangle\big) \cdot y_i' \cdot \mathbf{v} + \sigma'\big(\langle \mathbf{w}_j, \boldsymbol{\xi}_i' \rangle\big) \cdot \boldsymbol{\xi}_i' \big) \\
&= -q b_i y_i' \big( [\langle \mathbf{w}_j, y_i' \cdot \mathbf{v} \rangle]_+^{q-1} \cdot y_i' \cdot \mathbf{v} + [\langle \mathbf{w}_j, \boldsymbol{\xi}_i' \rangle]_+^{q-1} \cdot \boldsymbol{\xi}_i' \big)
\end{aligned}$$

and when $m + 1 \le j \le 2m$,

$$\nabla_{\mathbf{w}_j} \ell\big(y_i' \cdot f_{\mathbf{W}}(\mathbf{x}_i')\big) = q b_i y_i' \big( [\langle \mathbf{w}_j, y_i' \cdot \mathbf{v} \rangle]_+^{q-1} \cdot y_i' \cdot \mathbf{v} + [\langle \mathbf{w}_j, \boldsymbol{\xi}_i' \rangle]_+^{q-1} \cdot \boldsymbol{\xi}_i' \big)$$

Combining above two cases, we have

$$\begin{aligned}
\nabla_{\mathbf{w}_j} \ell\big(y_i' \cdot f_{\mathbf{W}}(\mathbf{x}_i')\big) &= -q \big( \mathbb{1}_{[1 \le j \le m]} - \mathbb{1}_{[m+1 \le j \le 2m]} \big) b_i y_i' \big( [\langle \mathbf{w}_j, y_i \cdot \mathbf{v} \rangle]_+^{q-1} \cdot y_i' \cdot \mathbf{v} + [\langle \mathbf{w}_j, \boldsymbol{\xi}_i' \rangle]_+^{q-1} \cdot \boldsymbol{\xi}_i' \big) \\
&= -q u_j b_i y_i' \big( [\langle \mathbf{w}_j, y_i' \cdot \mathbf{v} \rangle]_+^{q-1} \cdot y_i' \cdot \mathbf{v} + [\langle \mathbf{w}_j, \boldsymbol{\xi}_i' \rangle]_+^{q-1} \cdot \boldsymbol{\xi}_i' \big)
\end{aligned}$$

and therefore

$$\begin{aligned}
\nabla_{\mathbf{w}_j} L_{S'}(\mathbf{W}) &= \frac{1}{n_1} \sum_{i=1}^{n_1} \nabla_{\mathbf{w}_j} L_i(\mathbf{W}) = \frac{1}{n_1} \sum_{i=1}^{n_1} \nabla_{\mathbf{w}_j} \ell\big(y_i' \cdot f_{\mathbf{W}}(\mathbf{x}_i')\big) \\
&= -\frac{qu_j}{n_1} \sum_{i=1}^{n_1} b_i y_i' \big( [\langle \mathbf{w}_j, y_i' \cdot \mathbf{v} \rangle]_+^{q-1} \cdot y_i' \cdot \mathbf{v} + [\langle \mathbf{w}_j, \boldsymbol{\xi}_i' \rangle]_+^{q-1} \cdot \boldsymbol{\xi}_i' \big).
\end{aligned}$$

$\square$

*Proof of Theorem 4.4.* Recall the definition of $f_{\mathbf{W}}$ in (3.1) that

$$f_{\mathbf{W}}(\mathbf{x}) = \sum_{j=1}^{m} \Big[ \sigma\big(\langle \mathbf{w}_j, y \cdot \mathbf{v} \rangle\big) + \sigma\big(\langle \mathbf{w}_j, \boldsymbol{\xi} \rangle\big) \Big] - \sum_{j=m+1}^{2m} \Big[ \sigma\big(\langle \mathbf{w}_j, y \cdot \mathbf{v} \rangle\big) + \sigma\big(\langle \mathbf{w}_j, \boldsymbol{\xi} \rangle\big) \Big].$$

Define $\widetilde{\mathbf{w}}_j := m^{1/q} \cdot \mathbf{w}_j$, we have

$$\begin{aligned}
f_{\mathbf{W}}(\mathbf{x}) &= \sum_{j=1}^{m} \Big[ \sigma\big(\langle m^{-1/q} \cdot \widetilde{\mathbf{w}}_j, y \cdot \mathbf{v} \rangle\big) + \sigma\big(\langle m^{-1/q} \cdot \widetilde{\mathbf{w}}_j, \boldsymbol{\xi} \rangle\big) \Big] \\
&\quad - \sum_{j=m+1}^{2m} \Big[ \sigma\big(\langle m^{-1/q} \cdot \widetilde{\mathbf{w}}_j, y \cdot \mathbf{v} \rangle\big) + \sigma\big(\langle m^{-1/q} \cdot \widetilde{\mathbf{w}}_j, \boldsymbol{\xi} \rangle\big) \Big] \\
&= \frac{1}{m} \sum_{j=1}^{m} \Big[ \sigma\big(\langle \widetilde{\mathbf{w}}_j, y \cdot \mathbf{v} \rangle\big) + \sigma\big(\langle \widetilde{\mathbf{w}}_j, \boldsymbol{\xi} \rangle\big) \Big] - \frac{1}{m} \sum_{j=m+1}^{2m} \Big[ \sigma\big(\langle \widetilde{\mathbf{w}}_j, y \cdot \mathbf{v} \rangle\big) + \sigma\big(\langle \widetilde{\mathbf{w}}_j, \boldsymbol{\xi} \rangle\big) \Big] \\
&:= f_{\widetilde{\mathbf{W}}}(\mathbf{x}).
\end{aligned}$$

Since the standard deviation of Gaussian initialization of $\mathbf{w}_j$ is $\sigma_0$ and note that $\widetilde{\mathbf{w}}_j := m^{1/q} \cdot \mathbf{w}_j$, the standard deviation of Gaussian initialization of $\widetilde{\mathbf{w}}_j$ is $m^{1/q}\sigma_0 := \widetilde{\sigma}_0$.

On the other hand, note that the update rule of $\mathbf{w}_j^{(t)}$ is $\mathbf{w}_j^{(t+1)} = \mathbf{w}_j^{(t)} - \eta \cdot \nabla_{\mathbf{w}_j} L_{S'}(\mathbf{W}^{(t)})$, and in Lemma B.1, we have

$$\nabla_{\mathbf{w}_j} L_{S'}(\mathbf{W}) = -\frac{q u_j}{n_1} \cdot \sum_{i=1}^{n_1} b_i y_i' \big( [\langle \mathbf{w}_j, y_i' \cdot \mathbf{v} \rangle]_+^{q-1} \cdot y_i' \cdot \mathbf{v} + [\langle \mathbf{w}_j, \boldsymbol{\xi}_i' \rangle]_+^{q-1} \cdot \boldsymbol{\xi}_i'\big).$$

It follows that

$$\mathbf{w}_j^{(t+1)} = \mathbf{w}_j^{(t)} + \frac{q \eta u_j}{n_1} \cdot \sum_{i=1}^{n_1} b_i^{(t)} y_i' \big( [\langle \mathbf{w}_j^{(t)}, y_i' \cdot \mathbf{v} \rangle]_+^{q-1} \cdot y_i' \cdot \mathbf{v} + [\langle \mathbf{w}_j^{(t)}, \boldsymbol{\xi}_i' \rangle]_+^{q-1} \cdot \boldsymbol{\xi}_i'\big). \qquad \text{(B.1)}$$

By plugging $\mathbf{w}_j = m^{-1/q} \cdot \widetilde{\mathbf{w}}_j$ into (B.1), we have

$$\widetilde{\mathbf{w}}_j^{(t+1)} = \widetilde{\mathbf{w}}_j^{(t)} + \frac{q \eta m^{-\frac{1}{q}} u_j}{n_1} \cdot \sum_{i=1}^{n_1} b_i^{(t)} y_i' \big( [\langle \widetilde{\mathbf{w}}_j^{(t)}, y_i' \cdot \mathbf{v} \rangle]_+^{q-1} \cdot y_i' \cdot \mathbf{v} + [\langle \widetilde{\mathbf{w}}_j^{(t)}, \boldsymbol{\xi}_i' \rangle]_+^{q-1} \cdot \boldsymbol{\xi}_i'\big)$$

Assume $\widetilde{\eta} = m^{-\frac{1}{q}} \eta$, we have $\widetilde{\mathbf{w}}_j^{(t+1)} = \widetilde{\mathbf{w}}_j^{(t)} - \widetilde{\eta} \cdot \nabla_{\widetilde{\mathbf{w}}_j} L_{S'}(\widetilde{\mathbf{W}}^{(t)})$. Therefore, our data model and training algorithm is equivalent to the model and algorithm below:

$$f_{\widetilde{\mathbf{W}}^{+1}}(\mathbf{x}) = \frac{1}{m} \sum_{j=1}^{m} \Big[ \sigma\big(\langle \widetilde{\mathbf{w}}_j, y \cdot \mathbf{v} \rangle\big) + \sigma\big(\langle \widetilde{\mathbf{w}}_j, \boldsymbol{\xi} \rangle\big)\Big],$$

$$f_{\widetilde{\mathbf{W}}^{-1}}(\mathbf{x}) = \frac{1}{m} \sum_{j=m+1}^{2m} \Big[ \sigma\big(\langle \widetilde{\mathbf{w}}_j, y \cdot \mathbf{v} \rangle\big) + \sigma\big(\langle \widetilde{\mathbf{w}}_j, \boldsymbol{\xi} \rangle\big)\Big],$$

$$f_{\widetilde{\mathbf{W}}}(\mathbf{x}) = f_{\widetilde{\mathbf{W}}^{+1}}(\mathbf{x}) - f_{\widetilde{\mathbf{W}}^{-1}}(\mathbf{x}),$$

and we use gradient decent with learning rate $\widetilde{\eta}$ and cross-entropy loss to optimize such a data model, i.e.

$$\widetilde{\mathbf{w}}_0^{(t)} \sim \mathcal{N}(\mathbf{0}, \widetilde{\sigma}_0^2 \boldsymbol{I}_d), \widetilde{\mathbf{w}}_j^{(t+1)} = \widetilde{\mathbf{w}}_j^{(t)} - \widetilde{\eta} \cdot \nabla_{\widetilde{\mathbf{w}}_j} L_{S'}(\widetilde{\mathbf{W}}^{(t)}), L_{S'}(\widetilde{\mathbf{W}}^{(t)}) = \sum_{i=1}^{n_1} \ell(y_i' \cdot f_{\widetilde{\mathbf{W}}}(\mathbf{x}_i')),$$

where $\ell(z) = \log(1 + \exp(-z)), \widetilde{\sigma}_0 = m^{1/q} \sigma_0$. Note that the new model meets the one used in Cao et al. (2022). To leverage their result, we introduce condition 4.3 from Cao et al. (2022) and verify that the new model meets the new condition.

**Condition B.2** (Condition 4.2 in Cao et al. (2022)). Dimension $d$ is sufficiently large that $d = \widetilde{\Omega}(m^{2\vee[4/(q-2)]} n^{4\vee[(2q-2)/(q-2)]})$. Training sample size $n$ and neural network width $m$ satisfy $n, m = \Omega(\text{polylog}(d))$. Learning rate $\eta$ satisfies $\eta \leq \widetilde{O}(\min\{\|\mathbf{v}\|_2^{-2}, \sigma_p^{-2} d^{-1}\})$. The standard deviation of Gaussian initialization $\sigma_0$ is approximately chosen such that $\widetilde{O}(n d^{-\frac{1}{2}}) \cdot \min\{(\sigma_p \sqrt{d})^{-1}, \|\mathbf{v}\|_2^{-1}\} \leq \sigma_0 \leq \widetilde{O}(m^{-2/(q-2)} n^{-[1/(q-2)]\vee 1}) \cdot \min\{(\sigma_p \sqrt{d})^{-1}, \|\mathbf{v}\|_2^{-1}\}$.

**Theorem B.3** (Theorem 4.4 in Cao et al. (2022)). For any $\epsilon > 0$, let $T = \widetilde{\Theta}(\eta^{-1} m \cdot n (\sigma_p \sqrt{d})^{-q} \cdot \sigma_0^{-(q-2)} + \eta^{-1} \epsilon^{-1} n m^3 d^{-1} \sigma_p^{-2})$. Under Condition B.2, if $n^{-1} \cdot \text{SNR}^{-q} = \widetilde{\Omega}(1), \text{SNR} = \|\mathbf{v}\|_2 / \sigma_p \sqrt{d}$, then with probability at least $1 - d^{-1}$, there exists $0 \leq t \leq T$ such that:

1. The training loss converges to $\delta$, i.e., $L_S(\mathbf{W}^{(t)}) \leq \delta$.

2. The trained CNN has a constant order test loss: $L_{\mathcal{D}}(\mathbf{W}^{(t)}) = \Theta(1)$.

Note that in our setting, $m = \Theta(\text{polylog}(d)), n_1 = \widetilde{\Theta}(1), \|\mathbf{v}\|_2 = \Theta(d^{\frac{1}{2}}), \widetilde{\sigma}_0 = m^{1/q} \sigma_0, \sigma_0 = \Theta(d^{-\frac{3}{4}}) \sigma_p = \Theta(d^{0.01}), \widetilde{\eta} = m^{-\frac{1}{q}} \eta$ and $\eta = O(d^{-1-2\epsilon})$, it's not difficult to verify that Condition B.2 holds. Besides, $\text{SNR} = d^{-0.01}, n^{-1} \cdot \text{SNR}^{-q} = \widetilde{\Theta}(d^{q\epsilon}) = \widetilde{\Omega}(1)$. Therefore, the conclusion of Theorem B.3 holds for

$$\begin{aligned} T &= \widetilde{\Theta}(\widetilde{\eta}^{-1} m \cdot n (\sigma_p \sqrt{d})^{-q} \cdot \sigma_0^{-(q-2)} + \widetilde{\eta}^{-1} \epsilon^{-1} n m^3 d^{-1} \sigma_p^{-2}) \\ &= \widetilde{\Theta}(\widetilde{\eta}^{-1} \cdot (d^{1/2+\epsilon})^{-q} \cdot (d^{-3/4})^{-(q-2)} + \widetilde{\eta}^{-1} \epsilon^{-1} d^{-1} d^{-2\epsilon}) \\ &= \widetilde{\Theta}(\widetilde{\eta}^{-1} \cdot d^{(1/4-\epsilon)q-3/2} + \widetilde{\eta}^{-1} \epsilon^{-1} d^{-1-2\epsilon}) \\ &= \widetilde{\Theta}(\eta^{-1} \cdot d^{(1/4-\epsilon)q-3/2}). \end{aligned}$$

$\square$

## C  AUXILIARY LEMMAS

For the estimation of $\bar{\Lambda}^{(0)}$ and $\widehat{\Lambda}^{(0)}$, we introduce the following lemma.

**Lemma C.1** (Borell-TIS inequality)**.** Let $X$ be a centered Gaussian on $\mathbb{R}^m$ and set $\sigma_X^2 := \max_{i \in [m]} \mathbb{E}(X_i^2)$. Then for each $t > 0$,

$$\mathbb{P}\left(\left| \max_{i \in [m]} X_i - \mathbb{E}\left( \max_{i \in [m]} X_i \right) \right| > t \right) \leq 2e^{-\frac{t^2}{2\sigma_X^2}}.$$

For the expectation of $\widehat{\Lambda}_r^{(0)}$ and $\bar{\Lambda}_r^{(0)}$, we give the following lemma.

**Lemma C.2.** Let $Y = \max_{1 \leq i \leq m} X_i$, where $X_i \sim \mathcal{N}(0, \sigma^2)$ are i.i.d. random variables. Then

$$\frac{1}{\sqrt{\pi \log 2}} \sigma \sqrt{\log m} \leq \mathbb{E}[Y] \leq \sqrt{2}\sigma \sqrt{\log m}.$$

For the estimation of $\|\boldsymbol{\xi}_i\|_2^2$ and $\langle \boldsymbol{\xi}_i, \boldsymbol{\xi}_l \rangle$, we introduce following lemma.

**Lemma C.3** (Lemma B.2 in Cao et al. (2022))**.** Suppose that $\delta > 0$ and $d = \Omega(\log(4n/\delta))$. Then with probability at least $1 - \delta$,

$$\sigma_p^2 d/2 \leq \|\boldsymbol{\xi}_i\|_2^2 \leq 3\sigma_p^2 d/2,$$
$$|\langle \boldsymbol{\xi}_i, \boldsymbol{\xi}_l \rangle| \leq 2\sigma_p^2 \cdot \sqrt{d \log(4n^2/\delta)},$$

for all $i, l \in [n]$, $i \neq l$.

Besides, we introduce following lemma about tensor power method.

**Lemma C.4.** Consider an increasing sequence $x_t \geq 0$ defined as $x_{t+1} = x_t + \eta \cdot C_t x_t^{q-1}$, and $C_1 \leq C_t \leq C_2$ for all $t > 0$, then we have for $A > x_0$, every $\delta > 0$, and every $\eta > 0$:

$$\sum_{t \geq 0, x_t \leq A} \eta \leq \frac{\delta}{(1 - (1+\delta)^{-(q-2)})x_0 C_1} + \eta \cdot \frac{C_2}{C_1}(1+\delta)^{q-1}\left(1 + \frac{\log(A/x_0)}{\log(1+\delta)}\right),$$

$$\sum_{t \geq 0, x_t \leq A} \eta \geq \frac{\delta\left(1 - (x_0/A)^{q-2}\right)}{(1+\delta)^{q-1}\left(1 - (1+\delta)^{-(q-2)}\right)x_0 C_2} - \eta \cdot (1+\delta)^{-(q-1)}\left(1 + \frac{\log(A/x_0)}{\log(1+\delta)}\right).$$

*Proof of Lemma C.4.* For every $g = 0, 1, 2, \cdots$, let $\tau_g$ be the first iteration such that $x_t \geq (1+\delta)^g x_0$. Let $b$ be the smallest integer such that $(1 + \delta)^b x_0 \geq A$. By the definition of $\tau_g$, we have $x_t \in [(1 + \delta)^g x_0, (1 + \delta)^{g+1} x_0)$ for all $t \in [\tau_g, \tau_{g+1})$ and $x_{\tau_{g+1}} \geq (1+\delta)^{g+1} x_0$, $x_{\tau_g - 1} < (1+\delta)^g x_0$, leading to

$$\sum_{t \in [\tau_g, \tau_{g+1})} \eta \cdot C_t[(1+\delta)^g x_0]^{q-1} \leq x_{\tau_{g+1}} - x_{\tau_g} = \sum_{t \in [\tau_g, \tau_{g+1})} (x_{t+1} - x_t)$$

$$= \sum_{t \in [\tau_g, \tau_{g+1})} \eta \cdot C_t x_t^{q-1} \leq \sum_{t \in [\tau_g, \tau_{g+1})} \eta \cdot C_t[(1+\delta)^{g+1} x_0]^{q-1},$$

following lower bound for $x_{\tau_{g+1}} - x_{\tau_g}$:

$$x_{\tau_{g+1}} - x_{\tau_g} = x_{\tau_{g+1}} - x_{\tau_g - 1} - \eta \cdot C_{\tau_g - 1} x_{\tau_g - 1}^{q-1}$$
$$\geq (1+\delta)^{g+1} x_0 - (1+\delta)^g x_0 - \eta \cdot C_{\tau_g - 1}[(1+\delta)^g x_0]^{q-1}$$
$$= \delta(1+\delta)^g x_0 - \eta \cdot C_{\tau_g - 1}(1+\delta)^{(q-1)g} x_0^{q-1},$$

and following upper bound for $x_{\tau_{g+1}} - x_{\tau_g}$:

$$x_{\tau_{g+1}} - x_{\tau_g} = x_{\tau_{g+1} - 1} + \eta \cdot C_{\tau_{g+1} - 1} x_{\tau_{g+1} - 1}^{q-1} - x_{\tau_g}$$
$$\leq (1+\delta)^{g+1} x_0 + \eta \cdot C_{\tau_{g+1} - 1}[(1+\delta)^{(g+1)} x_0]^{q-1} - (1+\delta)^g x_0$$
$$= \delta(1+\delta)^g x_0 + \eta \cdot C_{\tau_{g+1} - 1}(1+\delta)^{(q-1)(g+1)} x_0^{q-1}.$$

Therefore,

$$\sum_{t\in[\tau_g,\tau_{g+1})} \eta\cdot C_t[(1+\delta)^g x_0]^{q-1} \le \delta(1+\delta)^g x_0 + \eta\cdot C_{\tau_{g+1}-1}(1+\delta)^{(q-1)(g+1)}x_0^{q-1},$$

$$\sum_{t\in[\tau_g,\tau_{g+1})} \eta\cdot C_t[(1+\delta)^{g+1} x_0]^{q-1} \ge \delta(1+\delta)^g x_0 - \eta\cdot C_{\tau_g-1}(1+\delta)^{(q-1)g}x_0^{q-1}.$$

These imply that

$$\sum_{t\in[\tau_g,\tau_{g+1})} \eta\cdot C_t \le \frac{\delta}{(1+\delta)^{(q-2)g}x_0} + \eta\cdot C_{\tau_{g+1}-1}(1+\delta)^{q-1} \le \frac{\delta}{(1+\delta)^{(q-2)g}x_0} + \eta\cdot C_2(1+\delta)^{q-1},$$

$$\sum_{t\in[\tau_g,\tau_{g+1})} \eta\cdot C_t \ge \frac{\delta}{(1+\delta)^{(q-2)g+(q-1)}x_0} - \eta\cdot C_{\tau_g-1}(1+\delta)^{-(q-1)}$$

$$\ge \frac{\delta}{(1+\delta)^{(q-2)g+(q-1)}x_0} - \eta\cdot C_2(1+\delta)^{-(q-1)}.$$

Recall $b$ is the smallest integer such that $(1+\delta)^b x_0 \ge A$, so we can calculate that

$$\sum_{t\ge 0, x_t\le A} \eta\cdot C_t \le \sum_{g=0}^{b-1} \frac{\delta}{(1+\delta)^{(q-2)g}x_0} + \eta\cdot C_2(1+\delta)^{q-1}b$$

$$= \frac{\delta\big(1-(1+\delta)^{-(q-2)b}\big)}{\big(1-(1+\delta)^{-(q-2)}\big)x_0} + \eta\cdot C_2(1+\delta)^{q-1}b$$

$$\le \frac{\delta}{\big(1-(1+\delta)^{-(q-2)}\big)x_0} + \eta\cdot C_2(1+\delta)^{q-1}b,$$

and

$$\sum_{t\ge 0, x_t\le A} \eta\cdot C_t \ge \sum_{g=0}^{b-1} \frac{\delta}{(1+\delta)^{(q-2)g+(q-1)}x_0} - \eta\cdot C_2(1+\delta)^{-(q-1)}b$$

$$= \frac{\delta\big(1-(1+\delta)^{-(q-2)b}\big)}{(1+\delta)^{q-1}\big(1-(1+\delta)^{-(q-2)}\big)x_0} - \eta\cdot C_2(1+\delta)^{-(q-1)}b$$

$$\ge \frac{\delta\big(1-(x_0/A)^{q-2}\big)}{(1+\delta)^{q-1}\big(1-(1+\delta)^{-(q-2)}\big)x_0} - \eta\cdot C_2(1+\delta)^{-(q-1)}b,$$

where the last inequality is due to $(1+\delta)^b x_0 \ge A$.

Note that $(1+\delta)^{b-1}x_0 < A$, i.e. $b \le 1 + \frac{\log(A/x_0)}{\log(1+\delta)}$, therefore

$$\sum_{t\ge 0, x_t\le A} \eta\cdot C_t \le \frac{\delta}{(1-(1+\delta)^{-(q-2)})x_0} + \eta\cdot C_2(1+\delta)^{q-1}\left(1+\frac{\log(A/x_0)}{\log(1+\delta)}\right),$$

$$\sum_{t\ge 0, x_t\le A} \eta\cdot C_t \ge \frac{\delta(1-x_0/A)}{(1+\delta)^{q-1}(1-(1+\delta)^{-(q-2)})x_0} - \eta\cdot C_2(1+\delta)^{-(q-1)}\left(1+\frac{\log(A/x_0)}{\log(1+\delta)}\right),$$

Note that $C_1 \le C_t \le C_2$, we have

$$\sum_{t\ge 0, x_t\le A} \eta \le \frac{\delta}{(1-(1+\delta)^{-(q-2)})x_0 C_1} + \eta\cdot\frac{C_2}{C_1}(1+\delta)^{q-1}\left(1+\frac{\log(A/x_0)}{\log(1+\delta)}\right),$$

$$\sum_{t\ge 0, x_t\le A} \eta \ge \frac{\delta\big(1-(x_0/A)^{q-2}\big)}{(1+\delta)^{q-1}\big(1-(1+\delta)^{-(q-2)}\big)x_0 C_2} - \eta\cdot(1+\delta)^{-(q-1)}\left(1+\frac{\log(A/x_0)}{\log(1+\delta)}\right).$$

$\square$

