# OpenReview forum: "How Does Semi-supervised Learning with Pseudo-labelers Work? A Case Study"
_ICLR.cc/2023/Conference — ICLR 2023 poster_

### Official Review · Reviewer_r8ze · 2022-10-25

**Confidence:** 3
**Correctness:** 3
**Technical Novelty And Significance:** 2
**Empirical Novelty And Significance:** Not applicable
**Recommendation:** 6

**Clarity, Quality, Novelty And Reproducibility:**

**Clarity** the paper has some clarity issues due to some choices of names (fine-tuning / self-supervised learning / CNN ...) as discussed above. The rest is clear.

**Novelty**: the main theorem seems novel

**Quality**: I have not checked the proofs in details, but the sketch proof and end results seems reasonable.

**Strength And Weaknesses:**

**UPDATE**: I’m updating my score 5->6 given that the authors addressed some of my initial concerns. The main remaining issues with this theoretical paper is that there are no actionable insights and that the considered setting is far from practice.
- - -

Strength:
- **theory is non-trivial**
- **proof sketch is clear and useful**
- **good review of related work**

Weakness
- (addressed in rebuttal) **Some lack of clarity in SSL nomenclature** multiple times when reading the paper the authors used terms that indicate that one setting was considered when actually it was not. For example:
    - linear probing vs fine-tuning. You often say that the model is "fine-tuned on small amount of labeled data", which suggests that you are finetuning all the weights of your model (eg abstract or figure 2). In other places, you seem to suggest that you only train a linear probe but freeze the pretrained model (eg end of section 3.1). From section 3.2 it seems that you indeed only train a linear probe with weights $a_k$
    - you use the term "pretext task" which in SSL typically means some task that is not directly related to the downstream task (eg predicting frames in a video, predicting the rotation of an image, predicting the order after a jigsaw transformation ...). Yet the paper is really about pseudo labeling in the sense that the pseudo labels are noisy labels of the downstream tasks
   - The model seems to be a 2 layer neural network instead of a CNN? Or is that a CNN with stride $d$?
- (addressed in rebuttal) **Not about self-supervised learning** as I just mentioned the paper is about pretraining to fit pseudo labels that are noisy versions of the downstream labels. This is not about self-supervised learning but semi-supervised learning (the pseudolabeler has to first be trained with some labels). To be clear there is nothing wrong with studying pseudo-labeling in semi-supervised learning but I think that the paper (especially the title) is misleading because of that.
- **toy assumptions** the setting is very toy (2-layer CNN, toy data generating process, uncommon SSL setting ...). That being said I do think that the high-level idea might generalize to more realistic settings.
- (addressed in rebuttal) **title and abstract are too grandiose** the title and abstract make it seem that the authors do much more than they actually prove. Authors need to be clear how constraint and limited the considered setting is.

**Question**
- **Is it really about representation learning?** the title and abstract seem to put much emphasis on "a representation learning perspective" but wouldn't a supervised learning model that is jointly trained on the pseudo labels and the real labels have the same performance gains? (with proper scheduling of the ratio of pseudo and actual labels"

**Summary Of The Paper:**

This is a theoretical paper that aims to prove the benefits of pretext-based self-supervised pretraining before supervised probing. To do so they consider a toy data generative process where the inputs are a concatenation of label-dependent features  (a linear function of the label) and some random features (Gaussian noise such that noise strength is larger signal). They then prove that an overparametrized 2-layer CNN that was pretrained will achieve near-zero downstream test error if the pretraining set is large, while a fully supervised model cannot if the number of labeled data because it will memorize the noise.


**Summary Of The Review:**

Assuming that the authors modify (or explain why not) their use of the terms "fine-tuning" and "self-supervised learning" I think that it could be of interest to some theoretical researchers. I am nevertheless not sure whether ICLR is the right venue to reach such researchers.

---

> ### Author Response · Authors · 2022-11-14
> **Response to Reviewer r8ze**
>
> Thank you for your helpful comments!
>
> ----
>
> **Q1**: Linear probing vs fine-tuning.
>
> **A1**: Thanks for your valuable suggestion. We have revised the wording and changed “finetune” to “linear probe” in the revision. The changes are highlighted in blue.
>
> ----
>
> **Q2**: You use the term "pretext task" which in SSL typically means some task that is not directly related to the downstream task but the paper is really about pseudo labeling in the sense that the pseudo labels are noisy labels of the downstream tasks.
>
> **A2**: In this paper, we consider a special kind of pretext task, which is designed by generating the pseudolabels for unlabeled data with the help of pseudolablers [1,2]. We have revised our paper to emphasize this point. Nevertheless, we want to stress that we can use other pretext tasks instead of using pseudo labelers to help linear probing. When a pseudo labeler is used in the pretext task, the only property we need is that the output of the pseudo labeler is correlated with the true label. In other words, we don’t need the pseudolabels to be “noisy labels of the downstream tasks”. In fact, we can choose other pretext tasks if we take the intrinsic structure of data into consideration. The main theorems of our paper still hold for more general pretext tasks, by modifying the proof accordingly. Since the pseudo-labeller is probably the simplest possible pretext task to satisfy the above property yet still widely used [1,2], we focus on this for the ease of presentation.
>
> [1] Xie et al. Self-training with noisy student improves imagenet classification. CVPR2020
>
> [2] Pham et al. Meta Pseudo labels. CVPR 2021
>
> ----
>
> **Q3**: The model seems to be a 2 layer neural network instead of a CNN? Or is that a CNN with stride d?
>
> **A3**: The model is indeed a CNN model since both feature patch $\mathbf{v}$ and noise patch $\mathbf{\xi}$ share the same weight in the neural network, i.e. $\sigma(\langle\mathbf{w}_j,\mathbf{v}\rangle)$ and $\sigma(\langle\mathbf{w}_j,\mathbf{\xi}\rangle)$. Like you said, it’s a CNN with stride d.
>
> ----
>
> **Q4**: The setting is very toy (2-layer CNN, toy data generating process, uncommon SSL setting ...). That being said I do think that the high-level idea might generalize to more realistic settings.
>
> **A4**: We agree with the reviewer that the setting is simple. However, even for this simple setting, there is no existing work that has been able to show any results of this kind in this direction. We believe that self-supervised learning with pseudo labeler using two-layer CNNs and simple data models is a good starting point for the understanding of self-supervised learning with more complicated neural networks and more general data distributions. Without a full characterization of what can be learned for the simple setting, it seems unlikely that we will find satisfying explanations for why self-supervised learning is so successful. As our work is the first theoretical result for pretext task-based SSL with neural networks in the literature, we think we have made significant progress on this problem.
>
> In addition, you are right that the high-level idea can be generalized to more realistic settings, e.g., smaller stride, linearly non-separable data, etc. While these extensions are possible, the corresponding analysis will be much more complicated. Therefore, we leave these extensions as future works. We have added a discussion in the future work section.
>
> ----
>
> **Q5**: Title and abstract are too grandiose.
>
> **A5**: We have reworked the title and abstract to tone down our claim. The changes are highlighted in blue.
>
> ----
>
> **Q6**: Is it really about representation learning? The title and abstract seem to put much emphasis on "a representation learning perspective" but wouldn't a supervised learning model that is jointly trained on the pseudo labels and the real labels have the same performance gains? (with proper scheduling of the ratio of pseudo and actual labels)
>
> **A6**: Here, by representation learning, we mean the learner can learn the feature vector $\mathbf{v}$ by pretraining on the pretext task. We would like to clarify that our analysis can handle the setting where $n_l = 0$, which is indeed self-supervised learning. The unlabeled data plays a central role in our analysis, while the additional labeled data only provides additional information about the feature and helps the feature learning (Lemma A.11, A.12). In fact, even without labeled data (setting $n_l = 0$), our pre-training step still works if you check the proof of Lemma A.11, A.12, A.13. In light of this, our pretraining on the pretext task is quite general, which covers self-supervised learning and self-supervised learning with additional labeled data (i.e., semi-supervised learning). We removed “a representation learning perspective” from the title to avoid overemphasizing it.

---

> ### Comment · Reviewer_r8ze · 2022-11-17
> **Thank you for your answers but I'm still concerned about overclaiming and your use of SSL**
>
> Thank you for your answer, I've read it carefully as well as other reviews.
>
> You partly answered my concerns (especially A1 and A6). My only remaining concern is about:
> - **Overclaiming**: although your updated manuscript is a little better than the first one, I still think that the setting is much more restricted than what the abstract/introduction suggests. For example, the sentence "We prove that, under certain data and neural network models," makes it seem that you this is a realistic setting when in reality it is a toy setting (esp. the data generating process). I think you should add "toy" in the abstract and introduction. Another possibility that you might prefer is to say that you prove that there exists some setting in which SSL is better than supervised, as existence makes it clear that this setting might not be realistic.
> - **Semi supervised instead of Self-supervised learning": I still think that the setting you consider is not really self supervised learning but rather semi supervised learning (despite your answer A2). For example, both papers you cited in the rebuttal concerning pseudolablers only talk about **semi**-supervised learning. Although I agree with your answer that "pseudolabellers are widely used" this is to my knowledge never used in self-supervised learning. Do you have any references that show the contrary? Typical pretext tasks in self supervised learning are about reconstruction the input (for example [Lee et al. 2021](https://openreview.net/pdf?id=Yx1OzVU_SRi) ). Given that the setting you consider is never used in self-supervised learning but common in semi-supervised learning I think you should frame it as such (also suggested by Reviewer eLuC).
>
> I am happy to increase my score if both of the above are addressed.

---

> > ### Author Response · Authors · 2022-11-18
> > **We have revised the paper again following your suggestion**
> >
> > Dear Reviewer r8ze,
> >
> > Thank you for your suggestion and for willing to increase the score. We have revised our paper exactly following your detailed suggestion. In detail, we have emphasized that the data generation model is a toy model in the abstract, introduction, and problem setup sections. We have also renamed our learning pipeline as semi-supervised learning with pseudo-labelers. This is reflected in the paper title, abstract, introduction, related work, etc. Thank you.
> >
> > Best,
> > Authors

---

> > > ### Comment · Reviewer_r8ze · 2022-11-20
> > > **Score updated**
> > >
> > > I have read the changes and increased my score.

---

> > > > ### Author Response · Authors · 2022-11-20
> > > > **Thank you!**
> > > >
> > > > Thank you for raising the score!

---

### Official Review · Reviewer_jwNu · 2022-10-25

**Confidence:** 4
**Correctness:** 4
**Technical Novelty And Significance:** 3
**Empirical Novelty And Significance:** Not applicable
**Recommendation:** 8

**Clarity, Quality, Novelty And Reproducibility:**

Clarity: The paper has good clarification and motivation.

Quality: I checked the most proof in the appendix. They look good to me.

Novelty: The paper has novelty in studying representation learning in self-supervised learning.

Reproducibility: I believe the experiments part can be reproduced.


**Strength And Weaknesses:**

Strength:
1. In my understanding, the key idea is that pseudo labels with a positive correlation with true labels can provide more gradient projection on the effective directions. When the pseudo-label data number is much larger than the true-label data, there is a generalization gap between settings (a) and (b). The main results in Section 4 are non-trivial.
2. The paper has clear motivation and good writing. The proof sketch section is comprehensive and insightful.

Weakness:

My major concern is how to generate pseudo-labels in Condition 4.1. Based on my rough calculation, it needs $\Theta(d)$ true labeled data to train a non-trivial teacher such that test accuracy can be larger than ½ with a constant margin with high probability (correct me if I am wrong). Then, here is an unfair comparison for setting (b) which only use constant true labels. If both setting (a) and (b) has $\Theta(d)$ true labeled data, I think the guaranteed generalization gap may be small or even may vanish. Let me know how to fix it or I may misunderstand here.

Here are other concerns:
1. The paper considers linear data and it is acceptable. However, it is worth considering non-linear data settings such as XOR or low-degree polynomial in [1,2,3].
2. It seems the paper is considering neural networks (NN) rather than convolutional neural networks (CNN).
3. I understand the authors need ReLU^3 in the proof so that the gradient has a better formulation and property. However, it is a good try to consider the ReLU activation function by using some feature purification tricks (Allen-Zhu & Li (2020a)).
4. How large is $K$ in Section 3.4? Is it necessary to have a $K$ here rather than using $K=1$?

[1] Shi, Zhenmei, Junyi Wei, and Yingyu Liang. "A Theoretical Analysis on Feature Learning in Neural Networks: Emergence from Inputs and Advantage over Fixed Features." International Conference on Learning Representations. 2021.

[2] Frei, Spencer, Niladri S. Chatterji, and Peter L. Bartlett. "Random feature amplification: Feature learning and generalization in neural networks." arXiv preprint arXiv:2202.07626 (2022).

[3] Damian, Alexandru, Jason Lee, and Mahdi Soltanolkotabi. "Neural networks can learn representations with gradient descent." Conference on Learning Theory. PMLR, 2022.


**Summary Of The Paper:**

The paper theoretically studies optimization and generalization of self-supervised learning with pseudo labelers setting. The pseudo label and true label are correlated. It considers linear data setting (signal/noise ratio = $O(d^{0.01})$) and two-layer neural networks with ReLU^3 activation function. Comparing two settings (a) train networks on constant labeled data and $O(d)$ pseudo-labeled data (b) train on constant labeled data only, the setting (a) will have $o(1)$ test loss while setting (b) will have $\Theta(1)$ test loss. The proof idea is to manipulate the gradient and try to give a bound on the signal growth rate and noise growth rate.

**Summary Of The Review:**

The paper is well-writing and has a non-trivial conclusion. However, I have a major concern about unfair comparison in settings (a) and (b). I may accept the paper if the author can fix my concern, otherwise, I tend to reject it.

---

> ### Author Response · Authors · 2022-11-14
> **Response to Reviewer jwNu (Part 2)**
>
> **Q4**: I understand the authors need ReLU^3 in the proof so that the gradient has a better formulation and property. However, it is a good try to consider the ReLU activation function by using some feature purification tricks (Allen-Zhu & Li (2020a)).
>
> **A4**: We have revised our paper and now we can deal with $RELU^{q}$, for $q > 2$. We can also deal with smoothed ReLU introduced in Allen et.al., 2020b, which behaves like $RELU^{q}$ when neural input is smaller than a threshold c, and behaves linearly if the input is greater than c.
>
> We can show that during the first stage of the pretext task learning, the input of the neuron is smaller than $c$. Therefore our key Lemma 5.1, 5.2, 5.3, 5.4 will still hold.
>
> By adding preactivation noise $\rho$ to the input, Allen-Zhu & Li (2020a) can smooth the RELU activation function to be $\tilde{RELU} = \mathbb{E}_{\rho}[ReLU(z + \rho)]$. We think our work can also be extended to RELU by adding preactivation noise. We appreciate your suggestion and have added this in the future work direction in the revision.
>
> ----
>
> **Q5**: How large is K in Section 3.4? Is it necessary to have a K here rather than using K=1?
>
> **A5**: K can be chosen to be $1$. We introduce $K$ just to make our result more general (i.e., to deal with possibly more than one pretext tasks).

---

> ### Author Response · Authors · 2022-11-14
> **Response to Reviewer jwNu (Part 1)**
>
> Thank you for your supportive comments!
>
> ----
>
> **Q1**: My major concern is how to generate pseudo-labels in Condition 4.1. Based on my rough calculation, it needs $\Theta(d)$ true labeled data to train a non-trivial teacher such that test accuracy can be larger than $1/2$ with a constant margin with high probability. Then, here is an unfair comparison for setting (b) which only uses constant true labels.
>
> **A1**: We will address your concern from both theoretical and practical perspectives.
> In practice, the pseudo-labeler can be directly inferred from some auxiliary prediction tasks, including region/component filling[1,2], rotation prediction[3], category prediction[4], and patch-base spatial composition prediction[5], which require no labeled data for pretext task.
>
> For the theoretical aspect, if we have to specify a pseudo-labeler, in some cases we need no labeled data if we take the intrinsic structure of data into consideration. In fact, the only property we need for a pseudo-labeler is that it is correlated with the true label rather than a random guess. Regarding this, for example, we can pick a random dimension of the feature patch (assume each dimension of $\mathbf{v}$ is non-zero) and output its sign, this gives us a pseudo-labeler with accuracy $p$ with $|p-1/2|\geq C$. Notice that given a small amount of labeled data, the linear probing parameter $\mathbf{a}$ in the downstream task will correct the pseudo-labeler with $p\leq 1/2-C$ by learning the negative direction, the main theorems still hold by slightly modifying the proof. The only reason we assume $p\geq 1/2+C$ is that pseudo-labeler used in practice are often weak learners and it’s weird to assume $p<1/2$.
>
> [1] Criminisi et al. Region filling and object removal by exemplar-based image inpainting, IEEE Transactions on image processing, 2004
>
> [2] Zhang et al. Colorful image colorization, ECCV, 2016
>
> [3] Gidaris et al. Unsupervised representation learning by predicting image rotations, ICLR, 2018
>
> [4] Dosovitskiy et al. Discriminative unsupervised feature learning with exemplar convolutional neural network. PAMI, 2015
>
> [5] Carlucci et al. Domain generalization by solving jigsaw puzzles, CVPR, 2019
>
> ----
>
> **Q2**: The paper considers linear data and it is acceptable. However, it is worth considering non-linear data settings such as XOR or low-degree polynomials in [1,2,3].
>
> **A2**: Thanks for your suggestion. We have added a discussion on it in the future work section in the revision. Below we define the XOR data model and briefly discuss how our analysis can be modified to show the learning of the XOR data.
>
> For XOR, we can consider data model $(\mathbf{x},y)$, where $\mathbf{x}^{\top}=[\mathbf{v}^{\top},\mathbf{\xi}^{\top}]$,$\mathbf{v}$ is feature patch and $\mathbf{\xi}$ is noise patch. If $y=1$, $\mathbf{v}=\mathbf{v}\_{1}$ or $\mathbf{v}=-\mathbf{v}\_{1}$ with equal probability. If $y=-1$, $\mathbf{v}=\mathbf{v}\_{2}$ or $\mathbf{v}=-\mathbf{v}\_{2}$ with equal probability. Assume $\mathbf{v}\_{1}$ and $\mathbf{v}\_{2}$ are orthogonal and have the same norm order. And let $f_{\mathbf{W}}(\mathbf{x})=\sum_{j=1}^{m}[\sigma(\langle\mathbf{w}\_j,\mathbf{v}\rangle)+\sigma(\langle\mathbf{w}\_j,\mathbf{\xi}\rangle)]-\sum_{j=m+1}^{2m}[\sigma(\langle\mathbf{w}\_j,\mathbf{v}\rangle)+\sigma(\langle\mathbf{w}\_j,\mathbf{\xi}\rangle)]$.
>
> In this case, the first $m$ neuron will adapt to either $\mathbf{v}\_{1}$ or $-\mathbf{v}\_{1}$ according to $\langle\mathbf{w}_j^{(0)},\mathbf{v}\_{1}\rangle>0$ or $\langle\mathbf{w}_j^{(0)},\mathbf{v}\_{1}\rangle<0$. The last $m$ neurons will adapt to either $\mathbf{v}\_{2}$ or $-\mathbf{v}\_{2}$ according to $\langle\mathbf{w}_j^{(0)},\mathbf{v}\_{2}\rangle>0$ or $\langle\mathbf{w}_j^{(0)},\mathbf{v}\_{2}\rangle<0$. By considering three kinds of inner product $\langle\mathbf{w}_j^{(t)},\mathbf{v}\_{1}\rangle,\langle\mathbf{w}_j^{(t)},\mathbf{v}\_{2}\rangle,\langle\mathbf{w}_j^{(t)},\mathbf{\xi}\rangle$ and applying similar analysis, the main theorems still hold.
>
> ----
>
> **Q3**: It seems the paper is considering neural networks (NN) rather than convolutional neural networks (CNN).
>
> **A3**: The model is actually a CNN model since both feature patch $\mathbf{v}$ and noise patch $\mathbf{\xi}$ share the same weight in the neural network, i.e. $\sigma(\langle\mathbf{w}_j,\mathbf{v}\rangle)$ and $\sigma(\langle\mathbf{w}_j,\mathbf{\xi}\rangle)$.

---

> ### Comment · Reviewer_jwNu · 2022-11-17
> **Appreciate authors' response and Raise score**
>
> I have read the rebuttal. It fixes most of my concerns, particularly the one about Condition 4.1. For the XOR part, I believe the current analysis can handle XOR based on this specific symmetric model and specific data distribution. However, if we generalize it a little bit, e.g., considering the model having bias terms or considering a three-dimension parity function rather than a two-dimension XOR, then the situation will be complicated. I know this is out of the scope of this paper. For the CNN vs. NN part, it is strange that the filter dimension is the same as the data dimension, but it is not a big concern on my side.
>
> I also read other reviewers' comments. I agree that the paper has some limitations and has a gap between practice, e.g., ReLU^q rather than ReLU, the fixed second layer, and no overlap between the signal patch and noise patch. From my perspective, some are necessary tricks to simplify the analysis and to preclude some technical challenges in the optimization of the training dynamic. I somehow understand the authors’ effort.
>
> As far as I know, the paper has its novelty. The paper intuitively showed that self-supervised learning with a large amount of unlabeled data, which has a weak correlation with labels, can help feature learning. Moreover, feature learning guarantees the generalization gap between self-supervised learning and supervised learning. Thus, I tend to accept the paper and raise the score.

---

> > ### Author Response · Authors · 2022-11-17
> > **Thank you for your positive feedback and support!**
> >
> > Thank you for your support and raising the score!

---

### Official Review · Reviewer_eLuC · 2022-10-26

**Confidence:** 3
**Correctness:** 2
**Technical Novelty And Significance:** 2
**Empirical Novelty And Significance:** 2
**Recommendation:** 5

**Clarity, Quality, Novelty And Reproducibility:**

The paper is reasonably clear and well-written. The theoretical result is most probably novel, and the experiments most probably reproducible. While the paper is clear, I also believe that it generalizes too broadly its contributions, making claims about self-supervised learning in general, when the results presented are specialized to one specific data generation process, one specific architecture and one specific pretraining method (which I would consider a semi-supervised learning method rather than a self-supervised learning method).

**Strength And Weaknesses:**

**Strengths:**
- The paper is reasonably well-written.
- Most of the hypothesis are clearly presented and the derivations look reasonable (even though I did not go through the full derivations in appendix).
- While the data and architecture setups are synthetic, the paper analyses the full gradient descent dynamic.

**Weaknesses:**
- Starting from the title, the claims made by the paper are very strong compared to the restrictive setting presented. This paper does not answer the question asked in the title: 'How does self-supervised learning work?'. It only shows that, in a restrictive (and somehow convoluted, more on that below) setting, one can design a self-supervised learning method, which has only few similarities with existing competitive self-supervised methods, that succeeds in learning more efficiently than a supervised counterpart. I think the title, abstract, and conclusion should be significantly reworked to reflect the restrictions imposed on the setup.
- The setting presented is very restrictive, and some of the restrictions/design choices diverge quite significantly with usual practical use cases. Without some proper motivations and explanations, it's hard to understand how generalizable the theoretical results provided by the paper are. Some examples of such restrictions/departure from practice:
  - The paper uses a ReLU^3 activation. Most common activation function are linear when pre-activations grow big, and either saturate or decrease linearily when preactivations get low. How crucial is the use of this specific activation function?
  - The generation process of the data is extremely simple, far from any practical setup.
  - SSL methods often rely on 1 pretext task to build a representation, then learn a linear predictor on top of the representation. Here, the paper relies on K pretext tasks to build K features, then aggregate the features linearly. Can we relate this setting to the usual SSL setting?
  - The architecture that is used is very ad-hoc, with features operating directly on the two separated parts of the data, and with only the input layer being trainable for the pretraining phase. The results obtained might be extremely dependent on this ad-hoc architecture. Would adding trainable features on the second layer break the analysis? Would removing the hard split in both data and architecture break the analysis?
  While I fully understand that studying setups closer to practically reasonable setups is much harder and might render the analysis intractable, I still believe that the current analysis does not support the claims made.
- While the paper aims to tackle self-supervised learning, the goal that is used to train the self-supervised model is a mix of pretext goal and supervised goal. This kind of mixing is usually considered semi-supervised more than self-supervised. Framing the paper in the context of semi-supervised learning might be a better fit.
- There are a couple of hard coded numerical values in condition 4.1. that could probably be replaced by constants bounded by some less ad-hoc thresholds. My personal view would be that this would make the analysis more readable and general, but that is mostly a minor point.

----
Post rebuttal comments:
I thank the authors for the efforts they put in the rebuttal. Many of my comments have been addressed and I am therefore raising my rating from 3 to 5. I think all the changes made are net positives. The reason I am not raising my rating further is that I still have my doubts on the practical insights that can be drawn from this analysis, which I already mentioned in my initial review.

A couple of additional remarks/questions:
- You mentioned in the rebuttal that working with K = 1 would recover a more standard SSL setup. If I am understanding things properly, the case K = 1 means that you are learning a single feature (the logits of a single pseudo-label predictor), then probing on this single feature (and thus learning a single weight a_0). I would still not consider this to be a standard SSL setup, where you would rather remove the final layer of your self-supervised network (here the untrained second layer), and replace it with a linear probe, working on the actual 'self-supervised' features. Am I misunderstanding something?
- I thank the authors for mentioning the fact that the analysis also worked with n_l = 0 which is something I missed in my first read. I would argue in favor of adding an additional comment mentionning that the case n_l = 0 corresponds to usual self-supervised training, whille n_l > 0 corresponds to a semi-supervised training regime (which only adds some generality to the paper).


**Summary Of The Paper:**

This paper theoretically analyse how self-supervised pretraining can achieve better performance than full supervised training in a synthetic setting, with fixed data, a fixed architecture, and a fixed self-supervised training scheme. More precisely, it looks at how, when a precise data generating process is assumed and a precise architecture is used, combining a form of self-supervised learning with supervised learning achieves zero test loss, while supervised learning achieves a non zero test loss.


**Summary Of The Review:**

The paper at hand provides a clear and probably correct theoretical analysis of a synthetic self(/semi)-supervised learning setup. Based on this analysis, the paper makes broad claims about self-supervised learning. I believe those claims are too broad, and should be properly tamed down to account for the narrowness of the setting considered.
Aside from this claim problem, I am not sure the theoretical results presented here can translate into practical insights, given how far the setup considered is from practical setups of interests.

Given those points, I don't recommend accepting the paper for now, but I am willing to discuss my current rating with both authors and other reviewers.

---

> ### Author Response · Authors · 2022-11-14
> **Response to Reviewer eLuC (Part 1)**
>
> Thank you for your detailed and constructive comments!
>
> ----
>
> **Q1**: The claims made by the paper are very strong compared to the restrictive setting presented. The title, abstract, and conclusion should be significantly reworked to reflect the restrictions imposed on the setup.
>
> **A1**: Thank you for your suggestion. We have reworked the title, abstract, and conclusion to make our claim more specific and accurate. The changes are highlighted in blue.
>
> ----
>
> **Q2**: How crucial is the use of $ReLU^3$ activation function?
>
> **A2**: We can handle activation functions of the form $RELU^{q}$ for $q>2$, and we have revised it accordingly in the revision. Our analysis is also applicable to the smoothed ReLU activation function introduced in Allen et al., 2020b, where $\tilde{RELU}(z) = \frac{1}{qc^{q-1}}RELU^{q}(z), z \leq c, \tilde{RELU}(z) = z - (1 - 1/q)c, z >c$. The smoothed ReLU activation function behaves like $RELU^{q}$ when the input is smaller than c, and behaves linearly if the input is greater than c. We can show that during the first stage of the pretext task learning, the input of the neuron is smaller than $c$. Therefore, our key Lemmas 5.1, 5.2, 5.3, 5.4 will still hold.
>
> ----
>
> **Q3**: The generation process of the data is extremely simple, far from any practical setup.
>
> **A3**: We agree with the reviewer that the data generation model is simple. However, even for this simple data model, there is no existing work that has been able to show any results of this kind in this direction. We believe that self-supervised learning with two-layer CNNs and simple data models is a good starting point for the understanding of self-supervised learning with more complicated neural networks and more general data distributions. Without a full characterization of what can be learned for the simple setting, it seems unlikely that we will find satisfying explanations for why self-supervised learning is so successful. As our work is the first theoretical result for pretext task-based SSL with neural networks in the literature, we think we have made significant progress on this problem.
>
> ----
>
> **Q4**: The paper relies on K pretext tasks to build K features, while SSL methods often rely on 1 pretext task to build a representation.
>
> **A4**: We would like to clarify that $K=1$ is a special case of our setting, and therefore our result covers $K=1$. So considering $K$ pretext tasks should be a strength (we consider the more general setting with possibly more than one pretext task) instead of a weakness of our work.
>
> ----
>
> **Q5**: The results obtained might be extremely dependent on this ad-hoc architecture. Would adding trainable features on the second layer break the analysis? Would removing the hard split in both data and architecture break the analysis?
>
> **A5**: Thanks for the questions.
>
> $\bullet$ It is possible to consider trainable second-layer parameters. We will need a more involved analysis jointly considering the signs of the second layer parameters and the quantities $\hat{\Lambda}\_{1}^{(t)}$, $\hat{\Lambda}\_{-1}^{(t)}$, $\bar{\Lambda}\_{1}^{(t)}$, $\bar{\Lambda}\_{-1}^{(t)}$, $\Gamma\_{i}^{(t)}$, $\Gamma\_{i}’^{(t)}$ defined in Section 5.
>
> $\bullet$ It is also possible for us to consider overlapping convolutions on the data inputs. The motivation for us to require a hard split between the two patches of data is to ensure that the feature learning and noise memorization dynamics can be studied approximately separately during the first stage of training. This can be extended to the setting where the convolution stride is large enough (but not as large as the filter size), because the feature learning and noise memorization dynamics can still be separately studied as long as the two data patches are only mildly overlapped.
>
> Although the above extensions are possible, the corresponding analyses will be more complicated. To our knowledge, the setting studied in the current paper is already quite significant and challenging compared with existing theoretical studies on self-supervised learning, as existing works mostly focus on linear models. We decide to leave the extensions suggested by the reviewer as a future work.  We have added a discussion on these future work directions in the conclusion section.

---

> ### Author Response · Authors · 2022-11-14
> **Response to Reviewer eLuC (Part 2)**
>
> **Q6**: While the paper aims to tackle self-supervised learning, the goal that is used to train the self-supervised model is a mix of pretext goal and supervised goal. This kind of mixing is usually considered semi-supervised more than self-supervised.
>
> **A6**: First of all, we would like to emphasize that our analysis can handle the setting with $n_l = 0$ (when there is no labeled data in the pretext task). The unlabeled data plays a central role in our analysis, while the additional labeled only provides more information about the feature and helps the feature learning (Lemmas A.11, A.12). In fact, without labeled data (setting $n_l = 0$), our pre-training step still works if you check the proof of Lemmas A.11, A.12, A.13. In light of this, our pretraining on the pretext task is quite general, and covers both self-supervised learning and self-supervised learning with additional labeled data (i.e., semi-supervised learning).
>
> ----
>
> **Q7**: There are a couple of hard-coded numerical values in condition 4.1. that could probably be replaced by constants bounded by some less ad-hoc thresholds.
>
> **A7**: Thank you for your suggestion. In our proof, we assume $\sigma_p = d^\epsilon$. Only in the presentation of Condition 4.1, we choose $\epsilon = 0.01$. We have revised Condition 4.1 to keep \epsilon to avoid hard-coded constants, and updated the statement of Theorem 4.4 and the related proofs in the appendix accordingly. As long as $\epsilon<1/8$, the analysis is still correct and the main theorems still hold.

---

> ### Author Response · Authors · 2022-11-18
> **Response to Additional Comments**
>
> **Q1**: You mentioned in the rebuttal that working with K = 1 would recover a more standard SSL setup. If I am understanding things properly, the case K = 1 means that you are learning a single feature (the logits of a single pseudo-label predictor), then probing on this single feature (and thus learning a single weight a_0). I would still not consider this to be a standard SSL setup, where you would rather remove the final layer of your self-supervised network (here the untrained second layer), and replace it with a linear probe, working on the actual 'self-supervised' features. Am I misunderstanding something?
>
> **A1**: Thank you for your comment. As strongly suggested by Reviewer r8ze, we have renamed our learning pipeline as semi-supervised learning with pseudo-labelers. The semi-supervised learning with pseudo-labelers studied in this paper can be seen as a special case of pretext task-based self-supervised learning where the pretext task is generated by pseudo-labelers. These pseudo-labers can either be obtained by training with labeled data (commonly used in semi-supervised learning) or directly generated based on the attribute found in the data (commonly used in pretext-based self-supervised learning). For more general pretext task-based self-supervised learning, the target features to be predicted in the pretext task are typically multi-dimensional. Regarding this, we work with $K\geq 1$. We hope this can fully resolve your concern.
>
> ----
>
> **Q2**: I thank the authors for mentioning the fact that the analysis also worked with $n_l = 0$ which is something I missed in my first read. I would argue in favor of adding an additional comment mentioning that the case $n_l = 0$ corresponds to usual self-supervised training, while $n_l > 0$ corresponds to a semi-supervised training regime (which only adds some generality to the paper).
>
> **A2**: Thank you for your suggestion. We have added a comment there.

---

### Decision · Program_Chairs · 2023-01-20

**Decision:**

Accept: poster

**Justification For Why Not Higher Score:**

Without clear practical actionable insights, the paper as presented might not be relevant to the wider ML community.

**Justification For Why Not Lower Score:**

The theoretical results are impressive and the paper appears worthy of publication.

**Metareview: Summary, Strengths And Weaknesses:**

The paper provides theoretical understanding to certain self-supervised learning approaches based on pre-training and fine-tuning.
The reviewers all appreciated the impressive theoretical results and clarity of the paper.
The reviewers also agreed the paper's insights may not have immediate practical relevance.

**Note From Pc:**

if the above contains the word "oral" or "spotlight" please see: "oral" presentation means -> notable-top-5% and "spotlight" means -> notable-top-25%. As stated in our emails, we are disassociating presentation type from AC recommendations